# CLUBench: A Clustering Benchmark

## Abstract

Clustering is a fundamental problem in data science with a long-standing research history. Over the past few decades, numerous clustering algorithms have been developed. However, a systematic and experimental evaluation of these algorithms remains lacking and is urgently needed. To address this gap, we introduce CLUBench, a comprehensive clustering benchmark comprising 23 algorithms of diverse principles evaluated on 131 datasets across tabular, text and image data types. Our extensive experiments (174,485) yield statistically meaningful insights into the performance of various clustering methods, such as the impact of hyperparameter tuning, similarity between algorithms, and the impact of data type and dimension. Notably, we observe low-rank characteristics in cross-model performance matrices, which facilitates an efficient strategy for rapid algorithm evaluation and selection in practical applications. Additionally, we provide an easy-to-use toolbox by encapsulating the source codes from the official code repository into a unified framework, accompanied by detailed instructions. With CLUBench, researchers and practitioners can efficiently select appropriate algorithms or datasets for evaluating new datasets or proposed methods. All benchmark datasets and the toolbox are fully open-sourced and available at `https://anonymous.4open.science/r/CLUBench-ICLR2026/`.

## 1 Introduction

We are living in a world full of data, which serves as an approximate reflection of the physical reality. One of the basic but vital means of mining these data is to classify or group them into a set of categories for further analysis and exploration. Therefore, data clustering is ubiquitous in the real world and cluster analysis becomes a fundamental technique used in various fields, including pattern recognition, information retrieval, bioinformatics, data compression, etc. Since the 1960s, there have been systematic studies (Forgy, 1965; McQueen, 1967) concerning the clustering problem. Over the years, numerous clustering algorithms have been developed based on different observations or assumptions. Following the de facto standard taxonomy (Xu & Tian, 2015; Yin et al., 2024), the conventional clustering algorithms can be organized into five categories. The first category is partition-based clustering with classic algorithms like K-means (McQueen, 1967), K-medoids (Park & Jun, 2009) and CLARANS (Ng & Han, 2002). The second category is hierarchical clustering (Zhang et al., 1996) which constructs the hierarchical relationship among data. The third category is density-based clustering (Ester et al., 1996; Ankerst et al., 1999; Comaniciu & Meer, 2002). The fourth category is grid-based clustering (Wang et al., 1997), in which the original data space is changed into a grid structure with a definite size for clustering. The last is model-based clustering, where each cluster is assigned a particular model (e.g. GMM (Rasmussen, 1999)) and the core idea is to find the best fit between clusters and models. With the progress of deep learning techniques and especially deep unsupervised learning, many deep architectural (neural networks) clustering (DC) methods (Xie et al., 2016; Guo et al., 2017; Ji et al., 2017; Huang et al., 2020; Li et al., 2021; Cai et al., 2022; Li et al., 2023; Metaxas et al., 2023; Li et al., 2025) have been proposed in the past few years. These DC methods exhibit marked superiority when dealing with complex and high-dimensional data compared with conventional clustering algorithms.

The profusion of cluster analysis techniques, on one hand, equips us with diverse tools. On the other hand, the selection and application of such abundant means also causes confusion. In response, a few surveys and reviews (Jain et al., 1999; Xu & Wunsch, 2005; Berkhin, 2006; Xu & Tian, 2015; Min et al., 2018; Aljalbout et al., 2018; Nutakki et al., 2018; Liu et al., 2022) about clustering

techniques have been introduced to organize and compare these algorithms. More recently, four clustering reviews (Yin et al., 2024; Wei et al., 2024; Zhou et al., 2024; Ren et al., 2024) are released, which further demonstrate the abundance and diversity of cluster analysis techniques again. Although the reviews provide considerable taxonomy, summarization, and comparison for these clustering techniques, they still lack a comprehensively quantified evaluation and analysis. Surprisingly, the benchmark in the literature concerning clustering techniques is quite limited and incomplete. Although there have been several attempts, such as (Javed et al., 2020; Zhou et al., 2024; Wei et al., 2024), these works, at least, have the following three limitations:

1. The benchmark datasets and clustering algorithms evaluated are quite limited (number and type), which can easily lead to a biased evaluation and conclusion.
2. The evaluation considers either conventional clustering algorithms or deep learning based clustering methods, lacking a joint and unified comparison.
3. There is a lack of a convenient and unified toolbox for the clustering methods.

In this benchmark, we attempt to address these limitations. Based on the existing work (Jeon et al., 2025) and publicly available dataset archive, we collect 131 datasets from various real-world fields and data types that cover tabular, text, and image. We evaluate these datasets on 23 clustering algorithms and provide complete experimental results, meaningful comparisons and analysis. In addition, we integrate the source codes, particularly the deep learning codes that depend on different deep learning frameworks, such as TensorFlow, PyTorch, and Caffe, into a unified framework, providing a consistent and convenient interface. Furthermore, our toolbox is easy to extend to new datasets and clustering methods. The main contributions of this benchmark are as follows.

- This benchmark provides a comprehensive evaluation for 23 clustering algorithms on 131 datasets from various fields. To a unified comparison for conventional and deep learning based clustering methods, this benchmark considers both original data samples and feature representation, particularly on image data.
- Based on these extensive results, this benchmark conduct the overall performance analysis and grouped performance comparison for investigating the potential preferences of algorithms.
- This benchmark analyzes performance similarity among algorithms and among datasets, as well as the low-rank structure of the performance matrices, to enable reliable performance prediction and rapid model selection.
- This benchmark provides a toolbox to foster the application and research of clustering techniques.

## 2 RELATED WORK

### 2.1 CLUSTERING ALGORITHMS

Basically, we can divide the clustering algorithms into two categories: conventional clustering algorithms and deep clustering algorithms, depending on whether neural networks are used.

**Conventional Algorithms.** Conventional clustering methods typically operate on original data features through distance calculation, similarity measurement, or density estimation. They can be categorized into five groups: partition-based, hierarchical, density-based, grid-based, and model-based. Representative examples include K-means (McQueen, 1967) for partitioning, BIRCH (Zhang et al., 1996) for hierarchical clustering, DBSCAN (Ester et al., 1996) and OPTICS (Ankerst et al., 1999) for density-based clustering, STING (Wang et al., 1997) for grid-based clustering, and Gaussian Mixture Models (GMMs) (Rasmussen, 1999) for model-based clustering. Beyond these, similarity-based approaches such as affinity propagation (AP) (Frey & Dueck, 2007), spectral clustering (SepClu) (Shi & Malik, 2000; Ng et al., 2001), and subspace clustering methods (Elhamifar & Vidal, 2013; Chen et al., 2020; Fan, 2021) further expand the landscape by leveraging pairwise relations or self-expressive representations. Besides these basic algorithms, there have been many extensions, such as kernel K-means (Liu, 2022), low-rank representation (Liu et al., 2012), multi-view subspace clustering (Kang et al., 2020), federated spectral clustering (Qiao et al., 2023), etc.

**Deep Clustering Algorithms.** In recent years, deep clustering (Xie et al., 2016; Dilokthanakul et al., 2016; Guo et al., 2017; Ji et al., 2017; Caron et al., 2018; Han et al., 2019; Asano et al., 2019; Huang et al., 2020; Cai et al., 2022; Metaxas et al., 2023; Zhang et al., 2024; Li et al., 2025) has

emerged to address the challenges of large-scale and complex structured datasets. A seminal approach is deep embedded clustering (DEC) (Xie et al., 2016), which jointly learns feature representations and cluster assignments, later extended by IDEC (Guo et al., 2017) with a reconstruction objective. Subsequent works explore diverse directions: self-expressive models combined with spectral clustering Ji et al. (2017), confidence-driven assignment (Huang et al., 2020), contrastive learning with data augmentation (Li et al., 2021), deep subspace clustering (Cai et al., 2022), diversity-aware objectives (Metaxas et al., 2023), and stability-based supervision (Li et al., 2025). These methods demonstrate the rapid evolution of deep clustering, yet systematic evaluation remains limited. There are more deep clustering algorithms and we will not detail them in this paper due to space limitations.

## 2.2 EXISTING CLUSTERING BENCHMARKS AND REVIEWS WITH EXPERIMENTAL EVALUATION

As mentioned above, numerous conventional and deep clustering algorithms have been developed over the past few decades. Therefore, it is crucial to evaluate these methods on diverse real-world datasets and provide the community with an easy-to-use toolbox for implementation. The prior studies (Jain et al., 1999; Xu & Wunsch, 2005; Berkhin, 2006; Omran et al., 2007; Von Luxburg et al., 2010; Murtagh & Contreras, 2012; Xu & Tian, 2015; Min et al., 2018; Aljalbout et al., 2018; Nutakki et al., 2018; Javed et al., 2020; Liu et al., 2022; Yin et al., 2024; Ren et al., 2024; Zhou et al., 2024; Wei et al., 2024) have reviewed clustering algorithms. For example, Jain et al. (1999) surveys partition-based and hierarchical methods, discussing their theoretical foundations and applications, while Murtagh & Contreras (2012) focuses on hierarchical clustering and offers implementations in multiple software environments. Javed et al. (2020) benchmarks conventional methods on 112 time series datasets, whereas Zhou et al. (2024), Ren et al. (2024) and Wei et al. (2024) provide overviews of deep clustering algorithms, with Zhou et al. (2024) and Wei et al. (2024) including experimental evaluations on diverse datasets. Our CLUBench differs from these works in several key aspects. First, existing studies typically focus on specific data types or algorithms. Given the rapid development of clustering methods, a more comprehensive benchmark that covers various methods and data sets is necessary. Second, most prior benchmarks evaluate only a limited number of datasets and provide relatively simple analyses, overlooking insights that could guide further research and applications. In contrast, CLUBench offers extensive evaluations with detailed performance analysis. Finally, many existing methods are not user-friendly due to their complex implementations. CLUBench provides a unified toolbox where each algorithm can be executed with just 3 lines of code. A detailed comparison between CLUBench and other works are provided in Table 1.

| Benchmark | Coverage | | | Algorithm Type (-based) | | | | | | Data Type | | | Resource Integration | |
|---|---|---|---|---|---|---|---|---|---|---|---|---|---|---|
| | Datasets | Algo. | Metrics | Partition | Hierarchy | Density | Model | Subspace | DL | Tabular | Image | Sequence | Data | Toolbox |
| Javed et al. (2020) | 112 | 8 | 1 | ✓ | ✓ | ✓ | ✗ | ✗ | ✗ | ✗ | ✗ | ✓ | ✗ | ✓[1] |
| Zhou et al. (2024)* | 4 | 2 | 1 | ✗ | ✗ | ✗ | ✗ | ✗ | ✓ | ✗ | ✓ | ✗ | ✓ | ✓[2] |
| Wei et al. (2024)* | 12 | 26 | 3 | ✗ | ✗ | ✗ | ✗ | ✗ | ✓ | ✓ | ✓ | ✓ | ✗ | ✗ |
| CLUBench (ours) | 131 | 23 | 3 | ✓ | ✓ | ✓ | ✓ | ✓ | ✓ | ✓ | ✓ | ✓ | ✓ | ✓ |

[1] https://github.com/ali-javed/clusteringBenchmark
[2] https://github.com/zhoushengisnoob/OpenDeepClustering

Table 1: Comparison among CLUBench and existing related works. * marks the clustering review, where only the datasets and methods evaluated in the experiments are counted.

# 3 CLUBENCH

Cluster analysis aims to organize samples into distinct groups based on the inherent patterns and similarities within the data. The precise definition of the clustering problem for different clustering algorithms is not completely agreed upon, such as soft clustering, hard clustering, and hierarchical clustering. In this section, we give a simple mathematical description by considering only the inputs and final outputs of algorithms, based on the description in previous work (Hansen & Jaumard, 1997). More importantly, we provide a guidance map in Section 3.2, which aims to offer readers a quick overview of the topics addressed and discussed in this benchmark, and guide them to relevant sections.

## 3.1 PROBLEM DESCRIPTION

Given a set $\mathcal{X} = \{\mathbf{x}_1, \mathbf{x}_2, \cdots, \mathbf{x}_n\}$ with $n$ samples and each $\mathbf{x}_i \in \mathbb{R}^m$. In CLUBench, a clustering algorithm $f$ aims to seek a $K$-partition of $\mathcal{X}$, $C = \{C_1, C_2, \cdots, C_K\}(K \leq n)$, such that

1. $C_i \neq \emptyset, i = 1, \ldots, K$;
2. $\bigcup_{i=1}^{K} C_i = \mathcal{X}$;
3. $C_i \cap C_j = \emptyset, i, j = 1, \ldots, K$ and $i \neq j$.

where 3 indicates all algorithms belong to hard clustering. $K$ must be determined prior to learning for some algorithms, whereas for others it is determined during the learning process.

## 3.2 MAP OF CLUBENCH

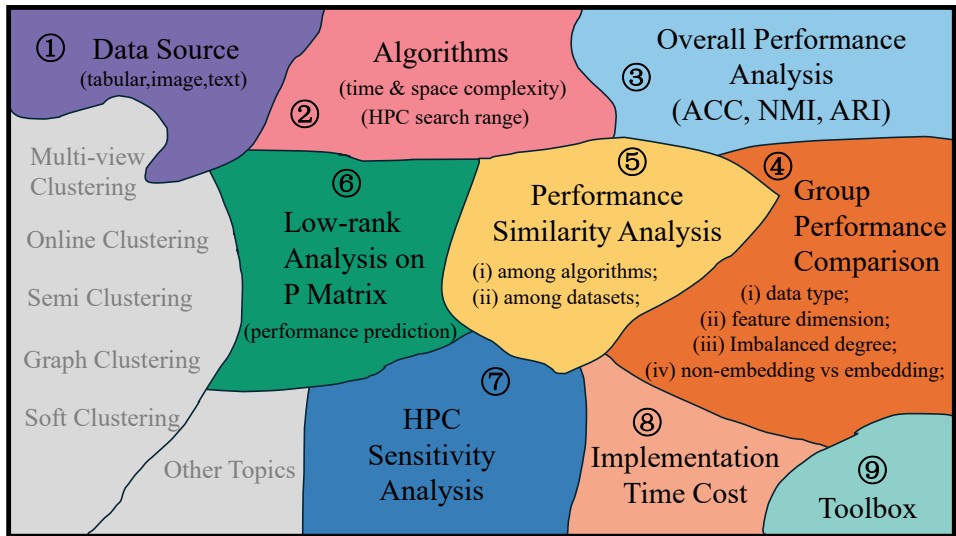

Figure 1: CLUBench MAP. Uncovered topics in CLUBench: Multi-view Clustering (Kang et al., 2020; Fang et al., 2023; Chen et al., 2023), Online Clustering (Beringer & Hüllermeier, 2006; Barbakh & Fyfe, 2008; Li et al., 2022), Semi-Clustering Basu et al. (2004); Bair (2013); Cai et al. (2023), Graph Clustering (clustering object:graph) (Schaeffer, 2007; Tian et al., 2014; Liu et al., 2023; Cai et al., 2024), Soft Clustering (Kumar & Futschik, 2007; Peters et al., 2013; Ferraro & Giordani, 2020) and possible Other Topics.

1. **Data Source**: Based on previous works (Wei et al., 2024; Zhou et al., 2024; Jeon et al., 2025) and publicly available dataset archive[1]. We gather and clean 131 benchmark datasets for clustering evaluation. The data type covers tabular, image, sequence (text) and bioinformatics data. The detailed statistics of these datasets are provided in Appendix B.1.

2. **Algorithms**: In CLUBench, we collect 23 clustering methods with (i) conventional classic algorithms including KMeans (McQueen, 1967), AggClu (Agglomerative Clustering) (Johnson, 1967), BIRCH (Zhang et al., 1996), DBSCAN (Ester et al., 1996) and SpeClu (Spectral Clustering) (Ng et al., 2001), GMM Rasmussen (1999), OPTICS (Ankerst et al., 1999), MeanShift (Comaniciu & Meer, 2002), k-PC (Agarwal & Mustafa, 2004), Affinity (Frey & Dueck, 2007), AutoSC (Fan, 2021); (ii) subspace-based algorithms like SSC (Elhamifar & Vidal, 2013), $S^3$COMP-C (Chen et al., 2020) and k-FSC (Fan, 2021); (iii) deep learning based clustering methods like DEC (Xie et al., 2016), IDEC (Guo et al., 2017), DSCN (Ji et al., 2017), PICA (Huang et al., 2020), ConClu(Contrastive Clustering) (Li et al., 2021), EDESC (Cai et al., 2022); (iv) the latest SOTA methods like DMICC (Li et al., 2023), DIVC (Metaxas et al., 2023) and LFSS (Li et al., 2025). More information about time and space complexity and HPC search range of the clustering algorithms evaluated is provided in Appendix B.2. Due to the extensive scope of this study, the current version of CLUBench does not include all prominent clustering algorithms, especially deep learning-based methods. We are actively integrating more methods like LRR (Liu et al., 2012), SENet (Zhang et al., 2021), $P^2$OT Zhang et al., 2024 and so on. In addition, since most of our datasets are single-view, multi-view clustering methods are not considered in our benchmark.

3. **Overall Performance Analysis**: First of all, we analyze the average performance (measured by ACC, Normalized Mutual Information (NMI), and Adjusted Rand Index (ARI)) of cluster-

---

[1]https://www.openml.org/search?type=data&status=active

ing algorithms across all 131 datasets. The detailed results are provided in Section 4.1. The complete performance results of each algorithm on each dataset are provided in Appendix D

4. **Group Performance Comparison**: To investigate the potential preferences of the clustering algorithms, we group the datasets according to four criteria: (i) data type (image, text, tabular, bioinformatics); (ii) feature dimensionality (low, middle, high); (iii) the degree of cluster imbalance; and (iv) the use of non-embedded versus embedded image data. A comparative analysis is then conducted based on these groupings in Section 4.2.

5. **Performance Similarity Analysis**: Based on the obtained performance results, a unique performance vector can be constructed for each clustering algorithm and dataset. Using these vectors, Section 4.3 analyzes the performance-based similarities among algorithms and among datasets, respectively.

6. **Low-rank Analysis on Performance Matrix**: In CLUBench, we search clustering performance for each algorithm across multiple Hyperparameter Configurations (HPC). For instance, if there are $h$ HPCs for each clustering algorithm, a performance matrix $\mathbf{P} \in \mathbb{R}^{131 \times (23h)}$ can be obtained. Under this condition, Section 4.4, analyzes the low-rank structure of $\mathbf{P}$ and constructs a matrix completion task to assess the effectiveness of performance prediction based on $\mathbf{P}$,

7. **HPC Sensitivity Analysis**: Section 4.5 compares the average performance difference between the best and worst HPC.

8. **Implementation Time Cost**: Appendix E details the average computational time (over five runs) for all algorithms.

9. **Toolbox**: To facilitate the application and research of such abundant clustering methods, we provide an easy-to-use toolbox encapsulated in Python and make it compatible with scikit-learn [2], keeping a simple usage logic. An example of calling DSCN is as follows:

```
CM = DSCN(**hpc) # hpc: dict of hyperparameter configurations.
CM.fit_predict(X) # X: data, (n_samples, dim) = X.shape.
CM.labels # predicted labels.
acc, nmi, ari, time = CM.evaluation(Y) # Y: true labels.
```

# 4 EXPERIMENT RESULTS AND ANALYSIS

## 4.1 OVERALL PERFORMANCE ANALYSIS

In this section, we compare the clustering performance (measured by ACC, NMI and ARI) across all datasets. Table 2 reports the average performance of a subset of algorithms on default and best HPC (in our search range). The complete results are provided in Appendix C.1. It is worth noting that we select best performance for each dataset and then obtain best average performance in Table 2.

| ConV. Algorithms | KMeans | AggClu | DBSCAN | BIRCH | GMM | SpeClu | SSC | AutoSC | k-FSC |
|---|---|---|---|---|---|---|---|---|---|
| ACC(default) | 0.593 | 0.501 | 0.424 | 0.592 | 0.579 | 0.588 | 0.518 | 0.607 | 0.496 |
| ACC(best) | 0.596 (+0.003) | 0.631 (+0.130) | 0.570 (+0.146) | 0.619 (+0.027) | 0.626 (+0.047) | **0.688** (+0.100) | 0.561 (+0.043) | - | 0.579 (+0.083) |
| NMI(default) | 0.336 | 0.178 | 0.028 | 0.330 | 0.315 | 0.318 | 0.200 | 0.327 | 0.200 |
| NMI(best) | 0.339 (+0.003) | 0.366 (+0.188) | 0.320 (+0.292) | 0.363 (+0.033) | 0.360 (+0.045) | **0.422** (+0.104) | 0.232 (+0.032) | - | 0.250 (+0.050) |
| ARI(default) | 0.293 | 0.124 | 0.019 | 0.272 | 0.261 | 0.249 | 0.150 | 0.282 | 0.156 |
| ARI(best) | 0.295 (+0.002) | 0.323 (+0.199) | 0.256 (+0.237) | 0.316 (+0.044) | 0.318 (+0.057) | **0.380** (+0.131) | 0.190 (+0.040) | - | 0.219 (+0.063) |
| **Deep Algorithms** | DEC | IDEC | DSCN | PICA | ConClu | EDESC | DMICC | DIVC | LFSS |
| ACC(default) | 0.560 | 0.550 | 0.550 | 0.540 | 0.519 | 0.557 | 0.543 | 0.541 | 0.529 |
| ACC(best) | 0.577 (+0.017) | 0.603 (+0.053) | 0.600 (+0.050) | 0.582 (+0.042) | 0.587 (+0.068) | **0.622** (+0.065) | 0.593 (+0.050) | 0.575 (+0.034) | 0.579 (+0.050) |
| NMI(default) | 0.290 | 0.251 | 0.240 | 0.257 | 0.257 | 0.307 | 0.272 | 0.257 | 0.252 |
| NMI(best) | 0.309 (+0.019) | 0.309 (+0.058) | 0.310 (+0.070) | 0.296 (+0.039) | 0.321 (+0.064) | **0.367** (+0.060) | 0.317 (+0.045) | 0.286 (+0.029) | 0.305 (+0.053) |
| ARI(default) | 0.248 | 0.210 | 0.171 | 0.220 | 0.202 | 0.257 | 0.232 | 0.219 | 0.212 |
| ARI(best) | 0.270 (+0.022) | 0.278 (+0.068) | 0.248 (+0.077) | 0.259 (+0.039) | 0.283 (+0.081) | **0.333** (+0.076) | 0.287 (+0.055) | 0.251 (+0.032) | 0.275 (+0.063) |

Table 2: The average clustering performance (ACC, NMI, ARI) across all datasets.

In addition, Figure 2 and Figure3 visualize the statistical performance difference and overall performance distribution difference based on ACC, NMI and ARI, respectively. Under these results, we have the following observations:

- Spectral clustering (SpeClu), as a conventional algorithm, significantly outperforms other algorithms on the average performance under the best hyperparameter configurations (HPCs),

---

[2] https://scikit-learn.org/stable/index.html

demonstrating its effectiveness on different data types. However, under default HPC settings, AutoSC achieves superior performance, since it is an automated machine learning algorithm.

- Moreover, most deep clustering methods, though effective on complex data such as images, do not show a significant advantage with respect to average performance compared to the conventional methods in our benchmark datasets. We attribute this to two factors: (1) tabular or data embedding lack spatial structure and augmentation strategies, making deep models designed for images less transferable; and (2) tabular features often directly capture semantic differences, so even simple metrics (e.g., Euclidean distance) remain highly effective, leaving limited room for neural representation learning to provide additional benefits.

- The performance gap between the best-performing HPC and the default HPC reflects the hyperparameter sensitivity of each method. Large gaps observed from AggClu, DBSCAN, and SpeClu indicate that these methods are particularly sensitive to hyperparameter changes.

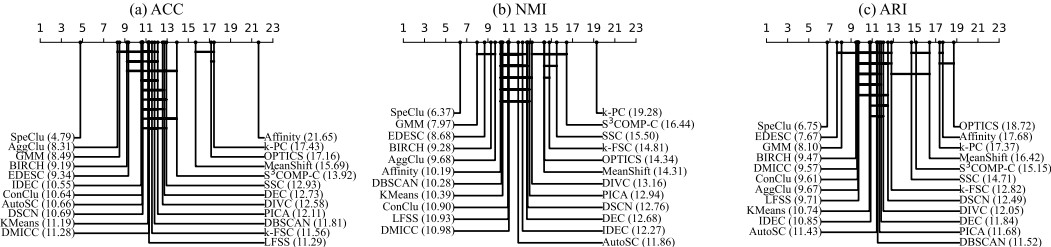

Figure 2: The CD diagram of best performance on all datasets by paired t-test. The '(value)'s are the average ranks.

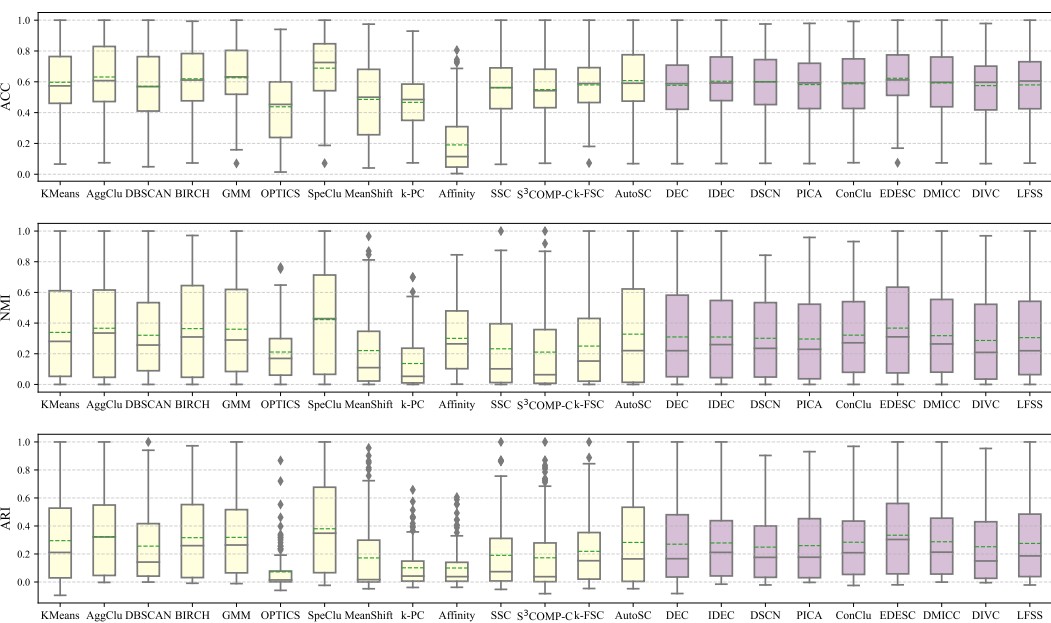

Figure 3: The statistics comparison of the performance on all datasets. The boxplots with light-yellow and purple correspond to the conventional and deep clustering algorithms, respectively. The green dash line denotes the average performance.

## 4.2 GROUP PERFORMANCE COMPARISON

### 4.2.1 ANALYSIS FROM THREE DIFFERENT PERSPECTIVES

Beyond the overall performance analysis, we group the datasets from three perspectives to uncover more specific insights: (i) data type (image, text, tabular, bioinformatics); (ii) feature dimensionality (low ($m \leq 100$), middle ($100 < m \leq 500$), high ($m > 500$); and (iii) the degree of cluster imbalance.

In Figure 4, we visualize the average ACC rank of each algorithms on different groups. Although SpeClu consistently outperforms most methods in many scenarios, we also observe that certain methods are particularly effective in specific groups. For instance, AggClu demonstrates superior performance on highly imbalanced datasets, GMM is more effective on low-imbalance datasets, and EDESC outperforms other methods on high-dimensional datasets. More analysis on NMI and ARI performance are shown in Appendix C.4.

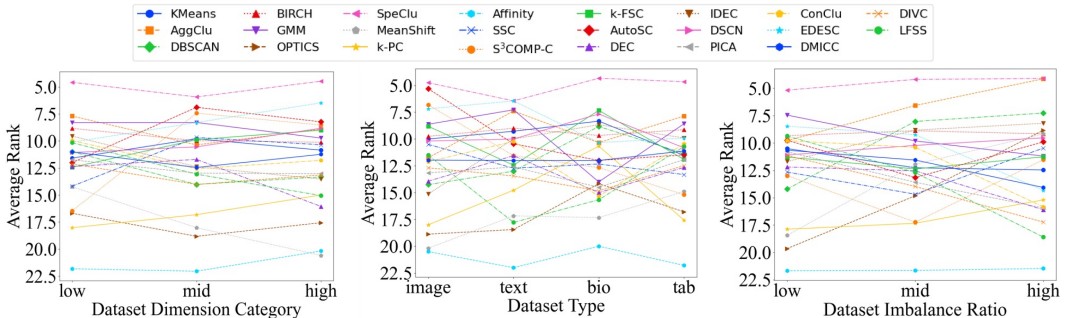

Figure 4: ACC Performance sensitivity to dataset categories across: dataset dimensionality, dataset type, and class imbalance ratio.

### 4.2.2 COMPARISON OF ORIGINAL AND EMBEDDED FEATURES OF IMAGE DATA

Deep learning-based clustering methods generally follow one of two paradigms. Some employ an end-to-end approach, where original image samples are fed directly into CNN-based models. Others use a two-stage process, where data embeddings are first generated and then provided to an MLP-based model for clustering. To enable an extensive comparison between the two paradigms, we rewrite an MLP-based code version for each CNN-based method based on their official codebase. Here, we compare the performance difference of the two paradigms on four widely used clustering benchmark datasets: STL-10, CIFAR10, CIFAR100, and COIL20. Embeddings were extracted using CLIP Radford et al. (2021). The related results are provided in Table 3. Across nearly all methods, using CLIP features leads to substantial performance improvement. In addition, we also notice that some methods, like ConClu and LFSS, are better suited for original image data (COIL20). We also compare the performance difference between conventional and deep clustering methods with respect to the embedding data. The related results are provided in Appendix C.2.

| Method | STL-10 Original ACC | NMI | ARI | Emb ACC | NMI | ARI | CIFAR10 Original ACC | NMI | ARI | Emb ACC | NMI | ARI | CIFAR100 Original ACC | NMI | ARI | Emb ACC | NMI | ARI | COIL20 Original ACC | NMI | ARI | Emb ACC | NMI | ARI |
|---|---|---|---|---|---|---|---|---|---|---|---|---|---|---|---|---|---|---|---|---|---|---|---|---|
| DEC | NA | NA | NA | 96.70 | 94.14 | 93.63 | NA | NA | NA | 74.89 | 70.98 | 61.49 | NA | NA | NA | 42.97 | 60.80 | 27.40 | NA | NA | NA | 74.87 | 89.98 | 71.48 |
| IDEC | NA | NA | NA | 94.30 | 90.81 | 88.59 | NA | NA | NA | 55.40 | 52.57 | 38.21 | NA | NA | NA | 43.36 | 60.79 | 27.53 | NA | NA | NA | 72.56 | 89.35 | 70.95 |
| DSCN | 24.74 | 16.41 | 7.93 | 79.24 | 73.64 | 67.58 | 19.29 | 5.47 | 2.71 | 73.12 | 63.71 | 55.03 | 6.18 | 16.50 | 0.50 | 32.36 | 58.00 | 23.69 | 45.35 | 61.10 | 20.90 | 61.78 | 78.85 | 50.78 |
| PICA | 44.20 | 34.40 | 22.90 | 96.24 | 91.79 | 91.96 | 38.20 | 25.70 | 16.80 | 50.18 | 41.33 | 33.85 | 9.50 | 23.20 | 3.00 | 16.46 | 38.02 | 7.79 | 76.30 | 86.90 | 72.70 | 86.73 | 91.08 | 82.36 |
| ConClu | 47.30 | 36.60 | 26.30 | 82.76 | 78.89 | 72.38 | 44.10 | 31.30 | 21.80 | 62.35 | 56.04 | 46.11 | 11.99 | 27.60 | 3.75 | 12.38 | 32.57 | 4.68 | 82.70 | 81.40 | 71.70 | 55.55 | 68.19 | 43.62 |
| EDESC | NA | NA | NA | 97.47 | 95.07 | 94.87 | NA | NA | NA | 80.59 | 75.56 | 69.64 | NA | NA | NA | 44.26 | 61.27 | 29.55 | NA | NA | NA | 81.02 | 92.77 | 77.97 |
| DMICC | 30.10 | 24.30 | 13.50 | 85.04 | 80.83 | 76.30 | 30.40 | 20.50 | 11.70 | 50.90 | 46.05 | 32.13 | 6.80 | 19.70 | 14.60 | 11.46 | 31.88 | 4.67 | 64.50 | 75.20 | 55.60 | 74.20 | 83.67 | 66.98 |
| DIVC | 25.00 | 13.60 | 6.90 | 95.19 | 90.47 | 90.06 | 25.50 | 13.40 | 7.10 | 61.13 | 50.71 | 41.94 | 8.77 | 22.80 | 2.20 | 15.70 | 37.16 | 7.26 | 80.60 | 86.00 | 73.90 | 86.45 | 90.06 | 80.60 |
| LFSS | 38.90 | 30.80 | 18.90 | 68.36 | 52.07 | 47.16 | 39.30 | 27.10 | 17.80 | 41.41 | 26.19 | 19.40 | 13.40 | 28.50 | 4.70 | 32.73 | 44.94 | 17.73 | 83.60 | 86.50 | 76.50 | 75.55 | 84.90 | 69.74 |

Table 3: Clustering results on four image datasets using raw images (Original) vs. CLIP embeddings (Emb). The best and second-best per column are highlighted in red and orange, respectively. The 'NA' indicates the result is not available.

### 4.3 PERFORMANCE SIMILARITY ANALYSIS

We construct a performance vector **p** for each algorithm and dataset. Specifically, each algorithm is characterized by a vector $\mathbf{p} \in \mathbb{R}^a$ representing its performance across all datasets with $a = 131$ in our benchmark, while each dataset is characterized by a vector $\mathbf{p} \in \mathbb{R}^b$ representing the performance of all algorithms on it with $b = 23$. Aggregating these vectors forms the performance matrices $\mathbf{P}_{acc}, \mathbf{P}_{nmi}, \mathbf{P}_{ari} \in \mathbb{R}^{a \times b}$.

### 4.3.1 AMONG ALGORITHMS

In this section, We obtain an overall performance matrix for all methods as $\hat{\mathbf{P}} = [\mathbf{P}_{acc}^T, \mathbf{P}_{nmi}^T, \mathbf{P}_{ari}^T] \in \mathbb{R}^{b \times 3a}$. Each $\hat{\mathbf{P}}_{i,:}$ represents a performance vector for a specific method. Dimensionality reduction techniques, such as t-SNE Maaten & Hinton (2008), can then be used to visualize these performance vectors and observe performance similarities between methods. Based on Figure 5, we have the following observations. First, most deep learning algorithms cluster together, while conventional methods form two distinct clusters, indicating strong performance similarities within each group. Second, methods with similar designs are located close to each other in the visualization. For instance, subspace-based methods such as SSC, k-PC, k-FSC, and S³COMP-C cluster together. IDEC, an extension of DEC, is positioned near DEC, and DIVC, an

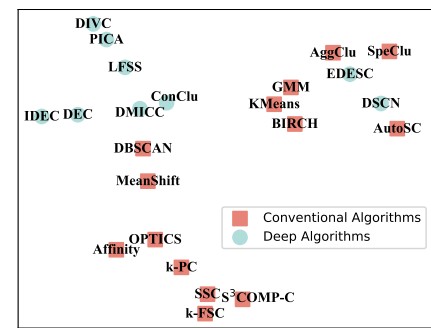

Figure 5: The t-SNE visualization results on algorithm performance vectors.

extension of PICA, is located close to PICA. Moreover, several deep algorithms that leverage data augmentation to construct positive and negative samples for contrastive learning, such as DIVC, PICA, LFSS, DMICC, and ConClu, form another coherent cluster. Lastly, such performance similarities provide valuable guidance for clustering research and practical applications on unseen datasets: selecting representative methods from each cluster can be far less time-consuming while remaining as effective as evaluating all.

### 4.3.2 AMONG DATASETS

Following the similar procedure mentioned above, we can obtain an overall performance matrix for all datasets as $\tilde{\mathbf{P}} = [\mathbf{P}_{acc}, \mathbf{P}_{nmi}, \mathbf{P}_{ari}] \in \mathbb{R}^{a \times 3b}$. Each $\tilde{\mathbf{P}}_{i,:}$ represents a performance vector for a specific dataset. Then t-SNE is utilized to obtain the visualization for each dataset. Here we divide the datasets into groups from 3 aspects: (1) data dimensionality (low, middle, high); (2) data type (image, text, bioinformatics data and other tabular datasets); (3) the degree of cluster imbalance (low, middle, high). The imbalance ratio is defined as the standard deviation of the cluster proportion distribution.

From the results in Figure 6, we observe the following: (1) With respect to dimensionality and data type, clusters are not clearly formed within groups, especially among high-dimensional datasets, suggesting that performance is not strongly related to dataset dimensionality or type. (2) In terms of cluster imbalance, most highly imbalanced datasets aggregate in a local region, which indicates that the clustering performance is easily affected by high data imbalance.

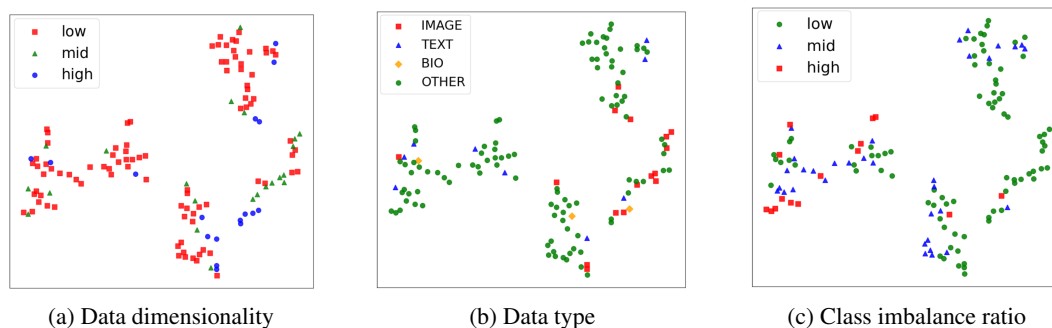

|  (a) Data dimensionality | (b) Data type | (c) Class imbalance ratio |

Figure 6: The t-SNE visualization of datasets based on their performance vectors. The points are labeled according to three perspectives: data dimensionality, data type, and class imbalance ratio.

### 4.4 LOW-RANK ANALYSIS ON PERFORMANCE MATRICES

To analyze the low-rank structure of the performance matrices, we conduct singular value decomposition (SVD) among the performance matrices (ACC, NMI, ARI) where each performance matrix consists of 131 rows (datasets) and 267 columns (clustering algorithms with different HPCs).

Let $\mathbf{P}_{acc}, \mathbf{P}_{nmi}, \mathbf{P}_{ari} \in \mathbb{R}^{a \times c}$ be the corresponding performance matrices and $\sigma_i$ denotes the $i$-th singular value, where $\sigma_1 \geq \sigma_2 \geq \cdots \geq \sigma_n > 0$, $a$ denotes the number of datasets, $c$ denotes the number of all clustering algorithms under different hyperparameter configurations. Figure 7 presents the cumulative contribution ratio of the singular values on $\mathbf{P}_{acc}$, where the cumulative contribution ratio ($ccr$) is defined by $ccr(j) = (\sum_{i=1}^{j} \sigma_i)/(\sum_{i=1}^{n} \sigma_i)$. The low-rank analysis on $\mathbf{P}_{nmi}$ and $\mathbf{P}_{ari}$ is provided in Appendix C.3.

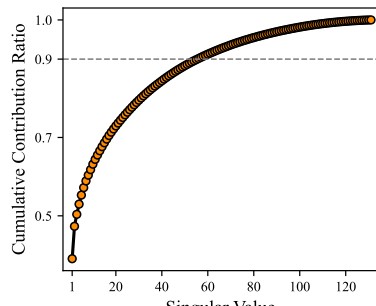

Figure 7: Cumulative contribution ratio of singular values of performance matrix $\mathbf{P}_{acc}$.

As evidenced by Figure 7, the cumulative contribution ratio of the first sixty (60/131) singular values ($ccr(60)$) exceeds 90%. This demonstrates that matrix $\mathbf{P}_{acc}$ possesses low-rank characteristics. This finding has two important implications: (1) clustering performance of novel datasets or clustering methods can be reliably predicted using only a subset of performance measurements, and (2) this property enables an efficient strategy for rapid algorithm evaluation and selection in practical applications.

To further verify the low-rank property of $\mathbf{P}_{acc}$ and the effectiveness of performance prediction, we construct the matrix completion tasks by MCAR (missing completely at random) mechanism with missing rate $\mathbf{mr} \in \{0.5, 0.6, 0.7, 0.8, 0.9\}$. We use matrix factorization and non-convex optimization techniques (Candes & Recht, 2012; Chi, 2018; Fan et al., 2019) to recover the missed entries of the performance matrix $\mathbf{P}_{acc}$. The recovery results are provided in Table 4, indicating that a rapid and reliable performance prediction is available based on the performance matrix $\mathbf{P}_{acc}$.

| mr | 0.5 | 0.6 | 0.7 | 0.8 | 0.9 |
|---|---|---|---|---|---|
| MAPE | 0.1326 (0.0039) | 0.1483 (0.0027) | 0.1666 (0.0024) | 0.1944 (0.0027) | 0.2499 (0.0068) |

Table 4: Recovery performance (MAPE) on the performance matrix $\mathbf{P}_{acc}$ in the setting of MCAR.

### 4.5 Hyperparameter Sensitivity Analysis

Figure 10 (in the Appendix) visualizes the average performance of different methods under their best- and worst-performing hyperparameter configurations. A larger gap between these two results indicates higher sensitivity to hyperparameter variation. From the Figure 10, we observe that methods such as AggClu, OPTICS, DBSCAN and SpeClu are relatively sensitive to hyperparameter changes, whereas methods like KMeans, BIRCH, and DEC are more robust. It is noteworthy, however, that the hyperparameter configuration (HPC) ranges differ across methods, and thus, the results may not provide a comprehensive measure of sensitivity.

## 5 Conclusions and Future Work

In this paper, we introduce CLUBench, a comprehensive clustering benchmark that evaluates 23 algorithms across 131 datasets. Based on extensive evaluation, we first conduct an overall performance analysis, which reveals that spectral clustering achieves markedly superior performance in terms of ACC, NMI, and ARI compared to other baselines. Meanwhile, deep learning-based methods do not demonstrate a general advantage over conventional algorithms. To investigate potential algorithmic preferences, we then categorize datasets according to several different criteria and observe performance variation on different kinds of datasets. Also, We utilize performance matrices to analyze similarities among datasets and algorithms, as well as their low-rank structure for a reliable performance prediction and rapid model selection. Within CLUBench, clustering performance is evaluated for each algorithm across multiple hyperparameter configurations, allowing us to assess sensitivity to hyperparameter choices within the specified search ranges. More importantly, we provide an easy-to-use toolbox to facilitate further research and application of these clustering techniques.

CLUBench can be easily extended to aggregate more clustering algorithms and new datasets. We will further consider more data types like graph and those uncovered topics like multi-view clustering, graph clustering, online clustering and so on in the future.

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

# A    EXPERIMENTAL COVERAGE

The experimental coverage of CLUBench are summarized in Table 5.

| # Datasets | # Algorithms | # HPCs | # Exp. Repeat | # Exp. Total | Metrics |
|---|---|---|---|---|---|
| 131 | 23 | 276 | 5 | 174,885($131 \times 276 \times 5$) | ACC, NMI, ARI, Time (second) |

Table 5: The experimental coverage of CLUBench.

**Datasets.**    Based on previous works (Wei et al., 2024; Zhou et al., 2024; Jeon et al., 2025) and publicly available dataset archive[3]. We gather and clean 131 benchmark datasets for clustering evaluation. The detailed statistics of these datasets are provided in Table 6, 7. The data type covers tabular, image and sequence (text), where the original samples and feature representations of image datasets are both used for two kind of comparisons: 1) the performance difference between conventional clustering algorithms and deep learning based methods; 2) the performance difference between DC methods.

**Algorithms.**    In CLUBench, we have collected 23 clustering methods with (i) conventional classic algorithms like KMeans (McQueen, 1967), AggClu (Agglomerative Clustering) (Johnson, 1967), DB-SCAN (Ester et al., 1996), BIRCH (Zhang et al., 1996), GMM Rasmussen (1999), OPTICS (Ankerst et al., 1999), Spectral Clustering (SpeClu) (Ng et al., 2001), MeanShift (Comaniciu & Meer, 2002), k-PC (Agarwal & Mustafa, 2004), Affinity (Frey & Dueck, 2007),AutoSC (Fan, 2021); (ii) subspace-based algorithms like SSC Elhamifar & Vidal (2013), S$^3$COMP-C (Chen et al., 2020),k-FSC (Fan, 2021); (iii) deep learning based clustering methods like DEC (Xie et al., 2016), IDEC (Guo et al., 2017), DSCN (Ji et al., 2017), PICA (Huang et al., 2020), ConClu(Contrastive Clustering) (Li et al., 2021), EDESC (Cai et al., 2022); (iv) the latest SOTA methods like DMICC (Li et al., 2023), DIVC (Metaxas et al., 2023) and LFSS (Li et al., 2025). For the deep methods with CNN-architecture, we retain the original architecture for raw image data while adapt a new version with MLP-architecture for evaluation on embedded image, tabular and text data.

**Evaluation.**    In CLUBench, we use three evaluation metrics ACC (Clustering Accuracy), NMI (Normalized Mutual Information), and ARI (Adjusted Rand Index) to quantifiably analyze the performance among the clustering methods. In addition, the average computational time (over five runs) of each algorithm is provided in Appendix E. Although not all experiments were conducted on the same hardware platform, these time records still provide valuable practical reference.

# B    DETAILED INFORMATION ON ALGORITHMS AND DATASETS

## B.1    DATASETS

To control the scale of the experiments, we have undersampled the datasets with sample size $n \geq 10,000$ to 10000 samples and removed the extreme clusters (outliers to some extent) with $r_i \leq 0.05$ ($r_i := \frac{\# \ i\text{th cluster}}{\# \ \text{maximal cluster}}$).

We details the statistics of all datasets used in Table 6, Table 7, where we give two ratios $r_{mm} := \frac{\# \ \text{minimal cluster}}{\# \ \text{maximal cluster}}$ and $r_{ma} := \frac{\# \ \text{minimal cluster}}{\# \ \text{all samples}}$ to reflect the biggest difference of sample size between clusters. Moreover, we give an imbalance factor (IR) to measure the imbalance ratio of a dataset. Suppose we have a dataset containing $K$ classes, then we define: $p_i = \frac{\# \ i\text{th class}}{\sum_{j=1}^{K} \# \ j\text{th class}}$. Then the imbalance factor is defined as the standard deviation of the class proportion distribution $\{p_i\}_{i=1}^{K}$.

---

[3]https://www.openml.org/search?type=data&status=active

| Datasets | Type | Samples | Dimension | Classes | $r_{mm}$ | $r_{ma}$ | IR |
|---|---|---|---|---|---|---|---|
| [1] echocardiogram | tabular | 61 | 10 | 2 | 0.386 | 0.279 | 0.221 |
| [2] skillcraft1_master_table_dataset | tabular | 3303 | 18 | 6 | 0.206 | 0.051 | 0.071 |
| [3] breast_cancer_wisconsin_original | tabular | 683 | 9 | 2 | 0.538 | 0.350 | 0.150 |
| [4] smoker_condition | tabular | 1012 | 7 | 2 | 0.656 | 0.396 | 0.104 |
| [5] glass_identification | tabular | 214 | 9 | 6 | 0.118 | 0.042 | 0.127 |
| [6] statlog_image_segmentation | tabular | 2310 | 19 | 7 | 1.000 | 0.143 | 0.000 |
| [7] planning_relax | tabular | 182 | 12 | 2 | 0.400 | 0.286 | 0.214 |
| [8] customer_classification | tabular | 1000 | 11 | 4 | 0.772 | 0.217 | 0.025 |
| [9] pima_indians_diabetes_database | tabular | 768 | 8 | 2 | 0.536 | 0.349 | 0.151 |
| [10] mobile_price_classification | tabular | 2000 | 20 | 4 | 1.000 | 0.250 | 0.000 |
| [11] spambase | tabular | 4601 | 57 | 2 | 0.650 | 0.394 | 0.106 |
| [12] rice_seed_gonen_jasmine | tabular | 9999 | 10 | 2 | 0.821 | 0.451 | 0.049 |
| [13] heart_attack_analysis_prediction_dataset | tabular | 303 | 13 | 2 | 0.836 | 0.455 | 0.045 |
| [14] user_knowledge_modeling | tabular | 258 | 5 | 4 | 0.273 | 0.093 | 0.098 |
| [15] world12d | tabular | 150 | 12 | 5 | 0.190 | 0.053 | 0.088 |
| [16] pumpkin_seeds | tabular | 2500 | 12 | 2 | 0.923 | 0.480 | 0.020 |
| [17] iris | tabular | 150 | 4 | 3 | 1.000 | 0.333 | 0.000 |
| [18] wine | tabular | 178 | 13 | 3 | 0.676 | 0.270 | 0.053 |
| [19] letter_recognition | tabular | 9992 | 16 | 26 | 0.904 | 0.037 | 0.001 |
| [20] mammographic_mass | tabular | 830 | 5 | 2 | 0.944 | 0.486 | 0.014 |
| [21] breast_tissue | tabular | 106 | 9 | 6 | 0.636 | 0.132 | 0.028 |
| [22] hepatitis | tabular | 80 | 19 | 2 | 0.194 | 0.163 | 0.338 |
| [23] predicting_pulsar_star | tabular | 9273 | 8 | 2 | 0.101 | 0.092 | 0.408 |
| [24] breast_cancer_wisconsin_prognostic | tabular | 569 | 30 | 2 | 0.594 | 0.373 | 0.127 |
| [25] wireless_indoor_localization | tabular | 2000 | 7 | 4 | 1.000 | 0.250 | 0.000 |
| [26] date_fruit | tabular | 898 | 34 | 7 | 0.319 | 0.072 | 0.062 |
| [27] zoo | tabular | 101 | 16 | 7 | 0.098 | 0.040 | 0.118 |
| [28] htru2 | tabular | 9999 | 8 | 2 | 0.101 | 0.092 | 0.408 |
| [29] ionosphere | tabular | 351 | 34 | 2 | 0.560 | 0.359 | 0.141 |
| [30] music_genre_classification | tabular | 1000 | 26 | 10 | 1.000 | 0.100 | 0.000 |
| [31] spectf_heart | tabular | 80 | 44 | 2 | 1.000 | 0.500 | 0.000 |
| [32] rice_dataset_cammeo_and_osmancik | tabular | 3810 | 7 | 2 | 0.748 | 0.428 | 0.072 |
| [33] ph_recognition | tabular | 653 | 3 | 15 | 0.864 | 0.058 | 0.002 |
| [34] banknote_authentication | tabular | 1372 | 4 | 2 | 0.801 | 0.445 | 0.055 |
| [35] wine_quality | tabular | 4873 | 11 | 5 | 0.074 | 0.033 | 0.160 |
| [36] cardiovascular_study | tabular | 2927 | 15 | 2 | 0.179 | 0.152 | 0.348 |
| [37] statlog_german_credit | tabular | 1000 | 24 | 2 | 0.429 | 0.300 | 0.200 |
| [38] boston | tabular | 154 | 13 | 3 | 0.371 | 0.169 | 0.121 |
| [39] seismic_bumps | tabular | 646 | 24 | 2 | 0.071 | 0.067 | 0.433 |
| [40] dry_bean | tabular | 9997 | 16 | 7 | 0.147 | 0.038 | 0.065 |
| [41] credit_risk_classification | tabular | 976 | 11 | 2 | 0.239 | 0.193 | 0.307 |
| [42] epileptic_seizure_recognition | tabular | 5750 | 178 | 5 | 1.000 | 0.200 | 0.000 |
| [43] website_phishing | tabular | 1353 | 9 | 3 | 0.147 | 0.076 | 0.188 |
| [44] optical_recognition_of_handwritten_digits | tabular | 3823 | 64 | 10 | 0.967 | 0.098 | 0.001 |
| [45] siberian_weather_stats | tabular | 1407 | 11 | 7 | 0.073 | 0.024 | 0.122 |
| [46] orbit_classification_for_prediction_nasa | tabular | 1722 | 11 | 3 | 0.065 | 0.056 | 0.371 |
| [47] magic_gamma_telescope | tabular | 9999 | 10 | 2 | 0.542 | 0.352 | 0.148 |
| [48] raisin | tabular | 900 | 7 | 2 | 1.000 | 0.500 | 0.000 |
| [49] patient_treatment_classification | tabular | 4412 | 10 | 2 | 0.679 | 0.404 | 0.096 |
| [50] fetal_health_classification | tabular | 2126 | 21 | 3 | 0.106 | 0.083 | 0.316 |
| [51] dermatology | tabular | 358 | 34 | 6 | 0.180 | 0.056 | 0.373 |
| [52] secom | tabular | 1567 | 590 | 2 | 0.071 | 0.066 | 0.000 |
| [53] paris_housing_classification | tabular | 10000 | 17 | 2 | 0.145 | 0.127 | 0.053 |
| [54] seeds | tabular | 210 | 7 | 3 | 1.000 | 0.333 | 0.275 |
| [55] wine_customer | tabular | 178 | 13 | 3 | 0.676 | 0.270 | 0.000 |
| [56] crowdsourced_mapping | tabular | 9997 | 28 | 4 | 0.060 | 0.043 | 0.212 |
| [57] durum_wheat_features | tabular | 9000 | 236 | 3 | 1.000 | 0.333 | 0.093 |
| [58] classification_in_asteroseismology | tabular | 1001 | 3 | 2 | 0.404 | 0.288 | 0.063 |
| [59] birds_bones_and_living_habits | tabular | 413 | 10 | 6 | 0.185 | 0.056 | 0.000 |
| [60] microbes | tabular | 9995 | 24 | 10 | 0.082 | 0.020 | 0.097 |
| [61] image_segmentation | tabular | 210 | 19 | 7 | 1.000 | 0.143 | 0.440 |
| [62] water_quality | tabular | 2011 | 9 | 2 | 0.676 | 0.403 | 0.235 |
| [63] insurance_company_benchmark | tabular | 5822 | 85 | 2 | 0.064 | 0.060 | 0.115 |
| [64] harbermans_survival | tabular | 306 | 3 | 2 | 0.360 | 0.265 | 0.175 |
| [65] yeast | tabular | 1459 | 8 | 8 | 0.065 | 0.021 | 0.132 |

Table 6: Statistics of datasets (1-65).

| Datasets | Type | Samples | Dimension | Classes | $r_{mm}$ | $r_{ma}$ | IR |
|---|---|---|---|---|---|---|---|
| [66] heart_disease | tabular | 297 | 13 | 5 | 0.081 | 0.044 | 0.004 |
| [67] ecoli | tabular | 327 | 7 | 5 | 0.140 | 0.061 | 0.052 |
| [68] extyaleb | tabular | 319 | 30 | 5 | 0.954 | 0.194 | 0.171 |
| [69] breast_cancer_coimbra | tabular | 116 | 9 | 2 | 0.812 | 0.448 | 0.061 |
| [70] student_grade | tabular | 395 | 29 | 2 | 0.491 | 0.329 | 0.234 |
| [71] human_stress_detection | tabular | 2001 | 3 | 3 | 0.634 | 0.250 | 0.004 |
| [72] fraud_detection_bank | tabular | 9999 | 112 | 2 | 0.362 | 0.266 | 0.031 |
| [73] pen_based_recognition_of_handwritten_digits | tabular | 7494 | 16 | 10 | 0.922 | 0.096 | 0.000 |
| [74] diabetic_retinopathy_debrecen | tabular | 1151 | 19 | 2 | 0.884 | 0.469 | 0.026 |
| [75] pistachio | tabular | 2148 | 28 | 2 | 0.744 | 0.426 | 0.262 |
| [76] turkish_music_emotion | tabular | 400 | 50 | 4 | 1.000 | 0.250 | 0.000 |
| [77] parkinsons | tabular | 195 | 22 | 2 | 0.327 | 0.246 | 0.000 |
| [78] weather | tabular | 365 | 192 | 7 | 0.603 | 0.121 | 0.148 |
| [79] blood_transfusion_service_center | tabular | 748 | 4 | 2 | 0.312 | 0.238 | 0.004 |
| [80] mfeat-karhunen | tabular | 2000 | 64 | 10 | 1.000 | 0.100 | 0.039 |
| [81] mfeat-factors | tabular | 2000 | 216 | 10 | 1.000 | 0.100 | 0.116 |
| [82] wall-robot-navigation | tabular | 5456 | 24 | 4 | 0.149 | 0.060 | 0.007 |
| [83] Waveform | tabular | 5000 | 21 | 3 | 0.971 | 0.329 | 0.053 |
| [84] gas-drift | tabular | 10000 | 128 | 6 | 0.546 | 0.118 | 0.005 |
| [85] mfeat-morphological | tabular | 2000 | 6 | 10 | 1.000 | 0.100 | 0.000 |
| [86] JapaneseVowels | tabular | 9961 | 14 | 9 | 0.485 | 0.079 | 0.124 |
| [87] rmftsa_sleepdata | tabular | 1024 | 2 | 4 | 0.233 | 0.092 | 0.337 |
| [88] first-order-theorem-proving | tabular | 6118 | 51 | 6 | 0.190 | 0.079 | 0.062 |
| [89] gina_prior2 | tabular | 3468 | 784 | 10 | 0.822 | 0.091 | 0.153 |
| [90] fabert | tabular | 8237 | 800 | 7 | 0.261 | 0.061 | 0.064 |
| [91] dilbert | tabular | 10000 | 2000 | 5 | 0.934 | 0.191 | 0.000 |
| [92] synthetic_control | tabular | 600 | 60 | 6 | 1.000 | 0.167 | 0.009 |
| [93] Drug Consumption | tabular | 1749 | 12 | 4 | 0.261 | 0.113 | 0.053 |
| [94] shuttle | tabular | 10000 | 9 | 2 | 0.195 | 0.163 | 0.005 |
| [95] tr45.wc | tabular | 676 | 8261 | 9 | 0.113 | 0.027 | 0.000 |
| [96] steel-plates-fault | tabular | 1941 | 33 | 2 | 0.531 | 0.347 | 0.000 |
| [97] fbis.wc | tabular | 2196 | 2000 | 11 | 0.128 | 0.030 | 0.000 |
| [98] mfeat-fourier | tabular | 2000 | 76 | 10 | 1.000 | 0.100 | 0.000 |
| [99] vehicle | tabular | 846 | 18 | 4 | 0.913 | 0.235 | 0.000 |
| [100] micro-mass | tabular | 360 | 1300 | 10 | 1.000 | 0.100 | 0.000 |
| [101] ISOLET | tabular | 7797 | 617 | 26 | 0.993 | 0.038 | 0.000 |
| [102] poker-hand | tabular | 10000 | 10 | 2 | 0.843 | 0.457 | 0.000 |
| [103] tamilnadu-electricity | tabular | 10000 | 2 | 20 | 0.480 | 0.030 | 0.000 |
| [104] mnist64 | image | 1082 | 64 | 6 | 0.967 | 0.164 | 0.000 |
| [105] MNIST_CLIP[+] | image | 9996 | 512 | 10 | 0.801 | 0.090 | 0.000 |
| [106] fashion_mnist | image | 3000 | 784 | 10 | 1.000 | 0.100 | 0.001 |
| [107] FashionMNIST_CLIP[+] | image | 10000 | 512 | 10 | 1.000 | 0.100 | 0.038 |
| [108] cifar10 | image | 3250 | 1024 | 10 | 1.000 | 0.100 | 0.021 |
| [109] CIFAR10_CLIP[+] | image | 10000 | 512 | 10 | 1.000 | 0.100 | 0.153 |
| [110] coil20[*] | image | 1440 | 400 | 20 | 1.000 | 0.050 | 0.062 |
| [111] COIL20_CLIP[+] | image | 1440 | 512 | 20 | 1.000 | 0.050 | 0.000 |
| [112] labeled_faces_in_the_wild | image | 2200 | 5828 | 2 | 1.000 | 0.500 | 0.006 |
| [113] flickr_material_database | image | 997 | 1536 | 10 | 0.990 | 0.099 | 0.053 |
| [114] street_view_house_numbers | image | 732 | 1024 | 10 | 0.341 | 0.064 | 0.000 |
| [115] har | image | 735 | 561 | 6 | 0.702 | 0.135 | 0.006 |
| [116] Indian_pines | image | 8858 | 220 | 5 | 0.121 | 0.055 | 0.000 |
| [117] satellite_image | image | 6435 | 36 | 6 | 0.408 | 0.097 | 0.000 |
| [118] olivetti_faces | image | 400 | 4096 | 40 | 1.000 | 0.025 | 0.000 |
| [119] cnae9 | text | 1080 | 856 | 9 | 1.000 | 0.111 | 0.000 |
| [120] imdb | text | 3250 | 700 | 2 | 1.000 | 0.500 | 0.000 |
| [121] hate_speech | text | 3221 | 100 | 3 | 0.075 | 0.058 | 0.000 |
| [122] sentiment_labeld_sentences | text | 2748 | 200 | 2 | 0.983 | 0.496 | 0.000 |
| [123] sms_spam_collection | text | 835 | 500 | 2 | 0.155 | 0.134 | 0.000 |
| [124] wos | text | 9997 | 4096 | 7 | 0.223 | 0.069 | 0.000 |
| [125] enron | text | 9999 | 4096 | 2 | 0.990 | 0.497 | 0.315 |
| [126] reuters | text | 6576 | 4096 | 3 | 0.562 | 0.243 | 0.004 |
| [127] 20newsgroups | text | 9991 | 4096 | 20 | 0.612 | 0.033 | 0.366 |
| [128] Mouse_retina | tabular (BioInfo) | 8352 | 6198 | 5 | 0.054 | 0.043 | 0.073 |
| [129] Campbell | tabular(BioInfo) | 9993 | 26774 | 14 | 0.052 | 0.024 | 0.003 |
| [130] PCam | image | 4000 | 27648 | 2 | 0.977 | 0.494 | 0.302 |
| [131] Baron Human | tabular (BioInfo) | 8451 | 20125 | 9 | 0.069 | 0.020 | 0.111 |

Table 7: Statistics of datasets (66-131).

## B.2 CLUSTERING ALGORITHMS

### B.2.1 ANALYSIS OF TIME AND SPACE COMPLEXITY

In this section, we detail the time and space complexity of the algorithms evaluated in Table 8. Note that the reported time complexity for iteratively optimized algorithms refers to that of a single iteration. The notations used in Table 8 are defined as follows.

- $n$: data size.
- $m$: feature dimension.
- $k$: number of clusters.
- $\rho$: proportion of nonzero entries.
- $\tilde{m}$: $\max\{m, h\}$ where $h$ is maximal latent dimension of neural networks.
- $\theta$: number of parameters of neural networks.
- $p$: dimension of output from encoder.
- $n_b$: batch size.
- $d$: dimension of subspace.
- $\tilde{p}$: $\max\{m, p\}$.
- $Q$: number of combinations of hyperparameters.

| Methods | Time Complexity | Space Complexity |
|---|---|---|
| KMeans (McQueen, 1967) | $\mathcal{O}(kmn)$ | $\mathcal{O}(nm + km)$ |
| AggClu (Agglomerative Clustering) (Johnson, 1967) | $\mathcal{O}(n^3)$ | $\mathcal{O}(n^2)$ |
| DBSCAN (Ester et al., 1996) | $\mathcal{O}(n \log n)$ | $\mathcal{O}(n^2)$ |
| BIRCH (Zhang et al., 1996) | $\mathcal{O}(mn)$ | $\mathcal{O}(mn)$ |
| GMM Rasmussen (1999) | $\mathcal{O}(nkm^2)$ | $\mathcal{O}(nm + km^2)$ |
| OPTICS (Ankerst et al., 1999) | $\mathcal{O}(n^2)$ | $\mathcal{O}(n)$ |
| SpeClu (Ng et al., 2001) | $\mathcal{O}(mn^2)$ | $\mathcal{O}(n^2)$ |
| MeanShift (Comaniciu & Meer, 2002) | $\mathcal{O}(n^2)$ | $\mathcal{O}(nm)$ |
| k-PC (Agarwal & Mustafa, 2004) | $\mathcal{O}(dmn + kdm^2 + m^2 n)$ | $\mathcal{O}(mn + kmd)$ |
| Affinity (Frey & Dueck, 2007) | $\mathcal{O}(n^2)$ | $\mathcal{O}(n^2)$ |
| SSC (Elhamifar & Vidal, 2013) | $\mathcal{O}(mn^2)$ | $\mathcal{O}(mn + \rho n^2)$ |
| S$^3$COMP-C (Chen et al., 2020) | $\mathcal{O}(m\rho n^3)$ | $\mathcal{O}(mn + \rho n^2)$ |
| k-FSC (Fan, 2021)) | $\mathcal{O}(kdmn)$ | $\mathcal{O}(mn + kmd + kdn)$ |
| AutoSC (Fan, 2021) | $\mathcal{O}(Q(k + m)n^2)$ | $\mathcal{O}(mn + n^2)$ |
| DEC (Xie et al., 2016) | $\mathcal{O}(\tilde{m}^2 n + knp)$ | $\mathcal{O}(\theta + knp + \tilde{m}n)$ |
| IDEC (Guo et al., 2017) | $\mathcal{O}(\tilde{m}^2 n + knp)$ | $\mathcal{O}(\theta + knp + \tilde{m}n)$ |
| DSCN (Ji et al., 2017) | $\mathcal{O}(\tilde{m}^2 n + n^2 p)$ | $\mathcal{O}(\theta + n^2 + \tilde{m}n)$ |
| PICA (Huang et al., 2020) | $\mathcal{O}(k^2 n_b + \tilde{m}^2 n_b)$ | $\mathcal{O}(\theta + \tilde{m}n + k^2)$ |
| ConClu(Contrastive Clustering) (Li et al., 2021) | $\mathcal{O}(\tilde{m}^2 n + n^2 \tilde{m})$ | $\mathcal{O}(\theta + \tilde{m}n_b + n_b^2)$ |
| EDESC (Cai et al., 2022) | $\mathcal{O}(kdpn + \tilde{m}\tilde{p}n)$ | $\mathcal{O}(\theta + \tilde{m}n + kn + kpd)$ |
| DMICC (Li et al., 2023) | $\mathcal{O}(\tilde{m}^2 n + n^2 \tilde{m})$ | $\mathcal{O}(\theta + p^2 + n_b^2 + n_b\tilde{m})$ |
| DIVC (Metaxas et al., 2023) | $\mathcal{O}(k^2 n_b + \tilde{m}^2 n_b)$ | $\mathcal{O}(\theta + \tilde{m}n + k^2)$ |
| LFSS (Li et al., 2025) | $\mathcal{O}(\tilde{m}^2 n + n^2 p)$ | $\mathcal{O}(\theta + n_b^2 + n_b\tilde{m})$ |

Table 8: Time and space complexity.

### B.2.2 THE SEARCH RANGE OF HYPERPARAMETER CONFIGURATION (HPC)

To obtain valid performance on all algorithms evaluated in CLUBench, we adjust the hyperparameter configurations (HPC) and search for the best performance for each clustering algorithm. The detailed search range of HPC is provided in the Table 9. It is worth noting that, in Table 9, each value of 'eps' in DBSCAN denotes a multiplier. In implementation, we use the average distance among samples as an 'eps_base' to control the argument in a reasonable range. For a dataset $\mathcal{X} = \{\mathbf{x}_1, \mathbf{x}_2, \cdots, \mathbf{x}_n\}$, we have

$$\text{eps} \leftarrow \text{eps} \times \text{eps\_base},$$

$$\text{eps\_base} = \frac{1}{n(n-1)} \sum_{i=1}^{n} \sum_{j \neq i}^{n} \text{dist}(\mathbf{x}_i, \mathbf{x}_j),$$

Where $\text{dist}(\cdot, \cdot)$ denotes a distance measure that is consistent with the argument 'metric'. The same process is used for 'max_eps' in OPTICS. Similarly, for 'gamma' in SC (Spectral Clustering), we have

$$\text{gamma} \leftarrow \text{gamma} \times \text{gamma\_base},$$

$$\text{gamma\_base} = \frac{1}{2 \times \text{median}(\{\|\mathbf{x}_i - \mathbf{x}_j\|_2 \mid \mathbf{x}_i, \mathbf{x}_j \in \mathcal{X}, i \neq j\})}.$$

| Algorithm | HPC (Hyperparameter Configuration) | # Total |
|---|---|---|
| KMeans | init $\in$ {kmeans++, random}; n_init=10; max_iter=500 | 2 |
| AggClu | metric $\in$ {euclidean, manhattan, cosine}; linkage $\in$ {average, complete, single} | 9 |
| DBSCAN | eps $\in$ {0.001, 0.005, 0.01, 0.1, 0.2, 0.4, 0.6, 0.8, 1.0, 10.0}; min_sample $\in$ {3, 5, 10}; metric $\in$ {euclidean, manhattan, cosine} | 90 |
| BIRCH | threshold $\in$ {0.3, 0.5, 0.7, 0.9}; branching_factor $\in$ {30, 50, 70} | 12 |
| GMM | covariance_type $\in$ {full, spherical}; init_params $\in$ {kmeans, kmeans++, random} | 6 |
| OPTICS | min_sample $\in$ {3, 5, 7}; max_eps $\in$ {0.01, 0.1, 1.0, 10.0}; metric $\in$ {euclidean, manhattan, cosine}; | 36 |
| SpeClu | affinity=knn; $k$ = {3, 5, 10, 20, 30, 50} affinity=rbf; gamma $\in$ {0.1, 0.5, 1.0, 5.0, 10.0} | 11 |
| MeaShift | bandwidth $\in$ {0.1, 0.3, 0.5, 0.7}; min_bin_freq $\in$ {1, 3} | 8 |
| k-PC | init_type=k-means; d $\in$ {5, 10, 20, 30, 50} | 5 |
| Affinity | damping $\in$ {0.6, 0.7, 0.8, 0.9}; max_iter $\in$ {200, 500}; affinity=euclidean | 8 |
| SSC | lambda = {100.0, 10.0, 1.0, 0.01} | 4 |
| S$^3$COMP-C | delta $\in$ {0.1, 0.3}; lambda $\in$ {0.1, 0.3, 0.5} | 6 |
| k-FSC | $d \in$ {5, 10, 20}; lambda = {0.01, 0.1, 0.3, 0.5} | 12 |
| AutoSC | Auto Hyperparameters Search | NA |
| DEC | lr= $1e^{-3}$; hidden_dims $\in$ {64, 32, 16} | 3 |
| IDEC | lr= $1e^{-4}$; hidden_dims $\in$ {64, 32, 16}; gamma $\in$ {0.01, 0.1} | 6 |
| DSCN | lr= $1e^{-3}$; hidden_dims $\in$ {64, 32, 16}; dim_subspace $\in$ {10, 20} | 6 |
| PICA | lr= $1e^{-3}$; lamda $\in$ {0.1, 0.5, 1} | 3 |
| ConClu | lr= $1e^{-4}$; instance_tempature $\in$ {0.1, 0.5, 1.0}; cluster_tempature $\in$ {0.1, 0.5, 1} | 9 |
| EDESC | lr= $1e^{-3}$; beta $\in$ {0.1, 1, 10}; d $\in$ {1, 5, 10} | 9 |
| DMICC | lr= $1e^{-3}$; lamda1 $\in$ {$1e^{-3}, 1e^{-4}$}; lamda2 $\in$ {$1e^{-3}, 1e^{-4}$} hidden_dims $\in$ {64, 32, 16} | 12 |
| DIVC | lr= $1e^{-3}$; lamda $\in$ {0.1, 0.5, 1} | 3 |
| LFSS | lr= $1e^{-3}$; hidden_dims $\in$ {64, 32, 16}; lamda_da $\in$ {0.1, 0.5}; temp = 0.5 | 6 |

Table 9: The search range of hyperparameter configurations.

# C MORE EXPERIMENTAL ANALYSIS AND RESULTS

## C.1 BEST AVERAGE PERFORMANCE OF EACH ALGORITHMS

In this subsection, we provide complete average performance results (best) of different methods over 131 datasets. The results are shown in Table 10. SpeClu (Spectral Clustering) significantly outperforms other algorithms on the average performance under the best hyperparameter configurations (HPCs), demonstrating its effectiveness on different data types. Most deep clustering methods do not show a significant advantage with respect to average performance compared to the conventional methods in our benchmark datasets.

| Algorithms | KMeans | AggClu | DBSCAN | BIRCH | GMM | OPTICS | SpeClu | MeanShift | k-PC | Affinity | SSC | S$^3$COMP-C |
|---|---|---|---|---|---|---|---|---|---|---|---|---|
| **ACC(default)** | 0.593 | 0.501 | 0.424 | 0.592 | 0.579 | 0.278 | 0.588 | 0.432 | 0.447 | 0.158 | 0.518 | 0.518 |
| **ACC(best)** | 0.596 | 0.631 | 0.570 | 0.619 | 0.626 | 0.437 | **0.688** | 0.485 | 0.466 | 0.190 | 0.561 | 0.549 |
| **NMI(default)** | 0.336 | 0.178 | 0.028 | 0.330 | 0.315 | 0.199 | 0.318 | 0.118 | 0.122 | 0.300 | 0.200 | 0.186 |
| **NMI(best)** | 0.339 | 0.366 | 0.320 | 0.363 | 0.360 | 0.211 | **0.422** | 0.220 | 0.136 | 0.300 | 0.232 | 0.210 |
| **ARI(default)** | 0.293 | 0.124 | 0.019 | 0.272 | 0.261 | 0.017 | 0.249 | 0.080 | 0.084 | 0.095 | 0.150 | 0.142 |
| **ARI(best)** | 0.295 | 0.323 | 0.256 | 0.316 | 0.318 | 0.072 | **0.380** | 0.171 | 0.101 | 0.099 | 0.190 | 0.172 |
| Algorithms | k-FSC | AutoSC | DEC | IDEC | DSCN | PICA | ConClu | EDESC | DMICC | DIVC | LFSS | – |
| **ACC(default)** | 0.496 | 0.607 | 0.560 | 0.550 | 0.550 | 0.540 | 0.519 | 0.557 | 0.543 | 0.541 | 0.529 | – |
| **ACC(best)** | 0.579 | - | 0.577 | 0.603 | 0.600 | 0.582 | 0.587 | 0.622 | 0.593 | 0.575 | 0.579 | – |
| **NMI(default)** | 0.200 | 0.327 | 0.290 | 0.251 | 0.240 | 0.257 | 0.257 | 0.307 | 0.272 | 0.257 | 0.252 | – |
| **NMI(best)** | 0.250 | - | 0.309 | 0.309 | 0.310 | 0.296 | 0.321 | 0.367 | 0.317 | 0.286 | 0.305 | – |
| **ARI(default)** | 0.156 | 0.282 | 0.248 | 0.210 | 0.171 | 0.220 | 0.202 | 0.257 | 0.232 | 0.219 | 0.212 | – |
| **ARI(best)** | 0.219 | - | 0.270 | 0.278 | 0.248 | 0.259 | 0.283 | 0.333 | 0.287 | 0.251 | 0.275 | – |

Table 10: The average clustering performance among all datasets.

## C.2 Comparison between Conventional and Deep algorithms based on Embedded Image Data

Table 11 reports the performance of different methods on CLIP-embedded datasets. Interestingly, conventional algorithms such as K-means and SpeClu (Spectral Clustering) outperform several deep methods, demonstrating that strong conventional approaches can be highly effective when combined with a powerful pre-trained feature extractor like CLIP.

| Algorithms | STL-10 | | | CIFAR10 | | | CIFAR100 | | | COIL20 | | |
|---|---|---|---|---|---|---|---|---|---|---|---|---|
| | ACC | NMI | ARI | ACC | NMI | ARI | ACC | NMI | ARI | ACC | NMI | ARI |
| KMeans | 98.00 | 95.15 | 95.63 | 71.78 | 66.24 | 55.32 | 47.09 | 62.39 | 31.89 | 80.17 | 93.44 | 77.99 |
| AggClu | 83.98 | 88.72 | 80.20 | 58.86 | 61.30 | 49.94 | 32.90 | 56.97 | 25.35 | 87.99 | 96.56 | 84.71 |
| DBSCAN | 36.50 | 48.66 | 21.58 | 14.64 | 10.87 | 0.12 | 7.01 | 16.75 | 0.70 | 84.65 | 94.90 | 82.68 |
| BIRCH | 97.06 | 93.92 | 93.60 | 68.85 | 64.69 | 49.06 | 45.73 | 61.53 | 28.36 | 77.71 | 93.33 | 76.60 |
| GMM | 81.52 | 88.34 | 78.44 | 73.01 | 67.70 | 54.07 | 46.13 | 61.57 | 31.17 | 81.64 | 92.06 | 79.25 |
| OPTICS | 13.30 | 13.58 | 0.12 | 10.43 | 10.13 | 0.02 | 6.22 | 11.44 | 0.03 | 32.22 | 52.30 | 5.59 |
| SpeClu | 98.60 | 96.51 | 96.93 | 76.71 | 70.71 | 48.68 | 45.70 | 60.61 | 16.53 | 93.06 | 95.40 | 90.51 |
| MeanShift | 10.00 | 0.00 | 0.00 | 10.01 | 0.02 | 0.00 | 1.01 | 0.02 | 0.00 | 5.00 | 0.00 | 0.00 |
| Affinity | 10.14 | 58.88 | 9.25 | 5.37 | 45.70 | 3.82 | 20.03 | 60.96 | 13.97 | 48.54 | 84.50 | 55.49 |
| DEC | 96.70 | 94.14 | 93.63 | 74.89 | 70.98 | 61.49 | 42.97 | 60.80 | 27.40 | 74.87 | 89.98 | 71.48 |
| IDEC | 94.30 | 90.81 | 88.59 | 55.40 | 52.57 | 38.21 | 43.36 | 60.79 | 27.53 | 72.56 | 89.35 | 70.95 |
| DSCN | 79.24 | 73.64 | 67.58 | 73.12 | 63.71 | 55.03 | 32.36 | 58.00 | 23.69 | 61.78 | 78.85 | 50.78 |
| PICA | 96.24 | 91.79 | 91.96 | 50.18 | 41.33 | 33.85 | 16.46 | 38.02 | 7.79 | 86.73 | 91.08 | 82.36 |
| ConClu | 82.76 | 78.89 | 72.38 | 62.35 | 56.04 | 46.11 | 12.38 | 32.57 | 4.68 | 55.55 | 68.19 | 43.62 |
| EDESC | 97.47 | 95.07 | 94.87 | 80.59 | 75.56 | 69.64 | 44.26 | 61.27 | 29.55 | 81.02 | 92.77 | 77.97 |
| DMICC | 85.04 | 80.83 | 76.30 | 50.90 | 46.05 | 32.13 | 11.46 | 31.88 | 4.67 | 74.20 | 83.67 | 66.98 |
| DIVC | 95.19 | 90.47 | 90.06 | 61.13 | 50.71 | 41.94 | 15.70 | 37.16 | 7.26 | 86.45 | 90.06 | 80.60 |
| LFSS | 68.36 | 52.07 | 47.16 | 41.41 | 26.19 | 19.40 | 32.73 | 44.94 | 17.73 | 75.55 | 84.90 | 69.74 |

Table 11: Clustering results on four image datasets using CLIP embeddings. Metrics are ACC / NMI / ARI (%). Best per column within each block (traditional vs. deep) is highlighted in red.

## C.3 Low-rank analysis on NMI and ARI

We also analyze the low-rank structure on performance matrices $\mathbf{P}_{nmi}$ and $\mathbf{P}_{ari}$. As evidenced by Figure 8, the cumulative contribution ratio of the first sixty (60/131) singular values ($ccr(60)$) exceeds 90%. This demonstrates that the performance matrices possess low-rank characteristics.

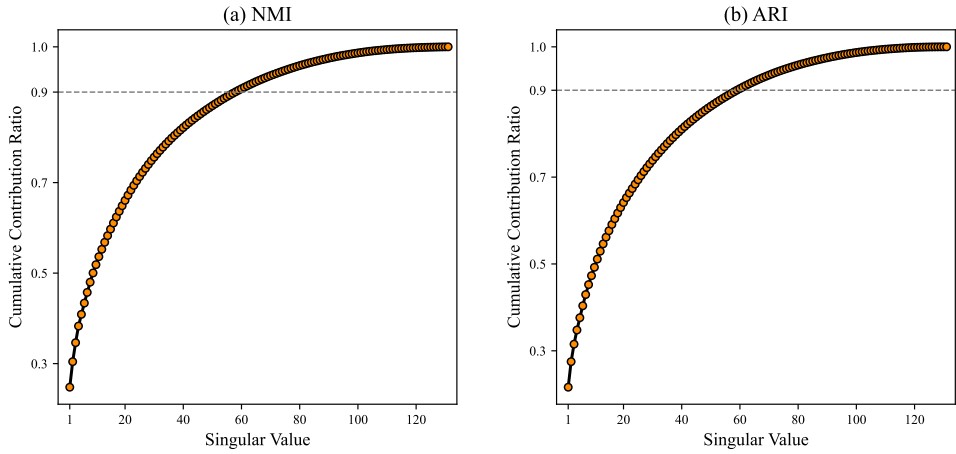

Figure 8: Cumulative contribution ratio of singular values of performance matrix $\mathbf{P}_{nmi}$ and $\mathbf{P}_{ari}$.

C.4 GROUP ANALYSIS FROM THREE DIFFERENT PERSPECTIVES

Beyond the overall performance analysis, we groups the datasets from three perspectives to uncover more specific insights: (i) data type (image, text, tabular, bioinformatics); (ii) feature dimensionality (low ($m \leq 100$), middle ($100 < m \leq 500$), high ($m > 500$); and (iii) the degree (IR B.1) of cluster imbalance (low (IR < 0.1), middle ($0.1 \leq IR \leq 0.3$), high (IR > 0.3) ).

In Figure 9, we visualize the average rank of ACC, NMI and ARI on different groups. Although SpeClu consistently outperforms most methods in many scenarios, we also observe that certain methods are particularly effective in specific groups. For instance, AggClu demonstrates superior performance on highly imbalanced datasets, GMM is more effective on low-imbalance datasets, and EDESC outperforms other methods on high-dimensional datasets. On different data type, SpeClu shows the consistent advantages compared most baselines and AutoSC achieves superior performance on image data but inferior performance on other data types.

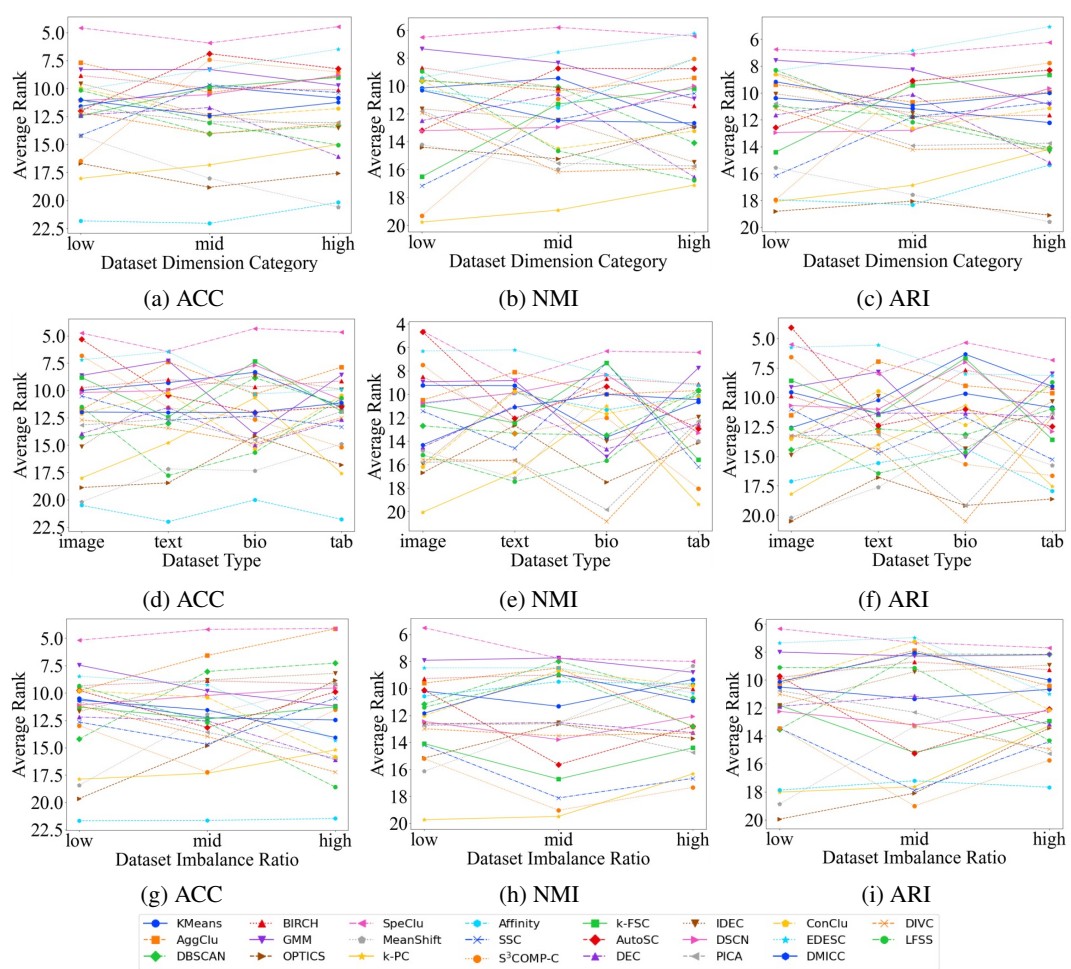

Figure 9: Performance sensitivity to dataset categories across: dataset dimensionality, dataset type, and class imbalance ratio.

## C.5 HYPERPARAMETER SENSITIVITY ANALYSIS

Figure 10 visualizes the average performance of different methods under their best- and worst-performing hyperparameter configurations. A larger gap between these two results indicates higher sensitivity to hyperparameter variation. From the Figure 10, we observe that methods such as AggClu, OPTICS, DBSCAN and SpeClu are relatively sensitive to hyperparameter changes, whereas methods like K-means, BIRCH, and DEC are more robust. It is noteworthy, however, that the hyperparameter configuration (HPC) ranges differ across methods, and thus, the results may not provide a comprehensive measure of sensitivity.

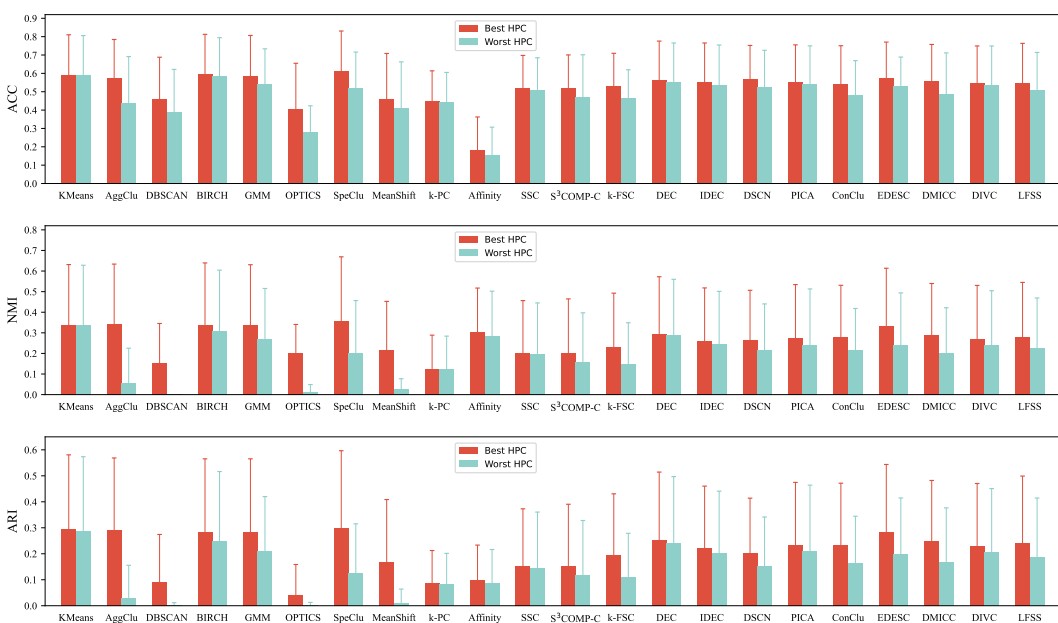

Figure 10: Performance comparison between the overall-best and overall-worst hyperparameter configurations.

## D  COMPLETE BEST PERFORMANCE (ACC, NMI, ARI) ON 131 DATASETS

The detailed best performance of each method on 131 datasets are provided from Table 12 to Table 23.

| Dataset Index | KMeans | AggClu | DBSCAN | BIRCH | GMM | OPTICS | SpeClu | MeanShift | k-PC | Affinity | SSC |
|---|---|---|---|---|---|---|---|---|---|---|---|
| 1 | 0.869 | 0.902 | 0.885 | 0.918 | 0.803 | 0.770 | 0.918 | 0.770 | 0.659 | 0.344 | 0.557 |
| 2 | 0.322 | 0.304 | 0.263 | 0.324 | 0.325 | 0.219 | 0.316 | 0.249 | 0.224 | 0.033 | 0.240 |
| 3 | 0.957 | 0.972 | 0.956 | 0.968 | 0.947 | 0.391 | 0.971 | 0.908 | 0.655 | 0.284 | 0.770 |
| 4 | 0.988 | 0.991 | 0.977 | 0.993 | 0.993 | 0.938 | 0.993 | 0.937 | 0.598 | 0.100 | 0.600 |
| 5 | 0.450 | 0.430 | 0.519 | 0.542 | 0.446 | 0.453 | 0.462 | 0.472 | 0.358 | 0.341 | 0.379 |
| 6 | 0.572 | 0.509 | 0.466 | 0.552 | 0.610 | 0.174 | 0.548 | 0.516 | 0.487 | 0.212 | 0.732 |
| 7 | 0.553 | 0.720 | 0.703 | 0.577 | 0.709 | 0.692 | 0.725 | 0.714 | 0.715 | 0.099 | 0.687 |
| 8 | 0.328 | 0.320 | 0.286 | 0.333 | 0.336 | 0.254 | 0.334 | 0.279 | 0.294 | 0.046 | 0.292 |
| 9 | 0.675 | 0.639 | 0.688 | 0.676 | 0.672 | 0.651 | 0.716 | 0.604 | 0.540 | 0.073 | 0.638 |
| 10 | 0.297 | 0.316 | 0.213 | 0.299 | 0.295 | 0.235 | 0.304 | 0.250 | 0.276 | 0.023 | 0.290 |
| 11 | 0.599 | 0.822 | 0.577 | 0.599 | 0.705 | 0.503 | 0.879 | 0.575 | 0.564 | 0.059 | 0.607 |
| 12 | 0.977 | 0.970 | 0.863 | 0.967 | 0.981 | 0.512 | 0.974 | 0.974 | 0.552 | 0.044 | 0.573 |
| 13 | 0.815 | 0.792 | 0.657 | 0.789 | 0.719 | 0.495 | 0.825 | 0.541 | 0.540 | 0.139 | 0.564 |
| 14 | 0.458 | 0.632 | 0.415 | 0.562 | 0.600 | 0.357 | 0.567 | 0.407 | 0.350 | 0.194 | 0.330 |
| 15 | 0.821 | 0.827 | 0.747 | 0.887 | 0.811 | 0.687 | 0.920 | 0.773 | 0.289 | 0.687 | 0.560 |
| 16 | 0.797 | 0.820 | 0.546 | 0.847 | 0.865 | 0.455 | 0.808 | 0.597 | 0.520 | 0.048 | 0.584 |
| 17 | 0.831 | 0.887 | 0.667 | 0.860 | 0.967 | 0.560 | 0.847 | 0.667 | 0.443 | 0.440 | 0.587 |
| 18 | 0.967 | 0.927 | 0.843 | 0.938 | 0.961 | 0.506 | 0.983 | 0.601 | 0.483 | 0.309 | 0.809 |
| 19 | 0.282 | 0.296 | 0.207 | 0.319 | 0.329 | 0.053 | 0.600 | 0.041 | 0.265 | 0.113 | 0.290 |
| 20 | 0.798 | 0.778 | 0.702 | 0.516 | 0.759 | 0.478 | 0.794 | 0.720 | 0.574 | 0.516 | 0.524 |
| 21 | 0.483 | 0.538 | 0.462 | 0.453 | 0.545 | 0.481 | 0.642 | 0.387 | 0.370 | 0.415 | 0.481 |
| 22 | 0.770 | 0.825 | 0.850 | 0.800 | 0.810 | 0.750 | 0.838 | 0.662 | 0.690 | 0.312 | 0.800 |
| 23 | 0.935 | 0.964 | 0.923 | 0.973 | 0.852 | 0.869 | 0.955 | 0.953 | 0.905 | 0.020 | 0.880 |
| 24 | 0.908 | 0.942 | 0.800 | 0.923 | 0.940 | 0.587 | 0.942 | 0.615 | 0.564 | 0.121 | 0.707 |
| 25 | 0.927 | 0.833 | 0.683 | 0.947 | 0.979 | 0.266 | 0.966 | 0.900 | 0.377 | 0.115 | 0.571 |
| 26 | 0.743 | 0.739 | 0.664 | 0.776 | 0.751 | 0.222 | 0.788 | 0.640 | 0.317 | 0.278 | 0.578 |
| 27 | 0.851 | 0.921 | 0.851 | 0.871 | 0.812 | 0.723 | 0.855 | 0.842 | 0.453 | 0.683 | 0.600 |
| 28 | 0.938 | 0.924 | 0.920 | 0.969 | 0.851 | 0.878 | 0.965 | 0.922 | 0.916 | 0.018 | 0.929 |
| 29 | 0.707 | 0.689 | 0.923 | 0.729 | 0.848 | 0.598 | 0.701 | 0.507 | 0.794 | 0.345 | 0.652 |
| 30 | 0.358 | 0.340 | 0.324 | 0.370 | 0.366 | 0.146 | 0.481 | 0.269 | 0.217 | 0.188 | 0.320 |
| 31 | 0.565 | 0.675 | 0.787 | 0.525 | 0.658 | 0.688 | 0.802 | 0.512 | 0.562 | 0.287 | 0.650 |
| 32 | 0.913 | 0.888 | 0.798 | 0.897 | 0.910 | 0.507 | 0.919 | 0.904 | 0.646 | 0.047 | 0.590 |
| 33 | 0.553 | 0.510 | 0.464 | 0.443 | 0.525 | 0.320 | 0.619 | 0.317 | 0.206 | 0.559 | 0.466 |
| 34 | 0.559 | 0.792 | 0.898 | 0.686 | 0.644 | 0.541 | 0.683 | 0.573 | 0.609 | 0.090 | 0.637 |
| 35 | 0.345 | 0.375 | 0.406 | 0.389 | 0.365 | 0.406 | 0.449 | 0.432 | 0.276 | 0.028 | 0.453 |
| 36 | 0.687 | 0.841 | 0.818 | 0.677 | 0.819 | 0.817 | 0.831 | 0.808 | 0.610 | 0.022 | 0.830 |
| 37 | 0.633 | 0.701 | 0.645 | 0.614 | 0.632 | 0.669 | 0.695 | 0.676 | 0.629 | 0.037 | 0.694 |
| 38 | 0.710 | 0.961 | 0.851 | 0.812 | 0.779 | 0.604 | 0.955 | 0.545 | 0.522 | 0.338 | 0.649 |
| 39 | 0.642 | 0.932 | 0.933 | 0.932 | 0.860 | 0.933 | 0.932 | 0.907 | 0.626 | 0.107 | 0.860 |
| 40 | 0.785 | 0.526 | 0.569 | 0.778 | 0.833 | 0.189 | 0.752 | 0.441 | 0.391 | 0.071 | 0.502 |
| 41 | 0.581 | 0.803 | 0.773 | 0.760 | 0.729 | 0.792 | 0.806 | 0.804 | 0.584 | 0.805 | 0.769 |
| 42 | 0.249 | 0.257 | 0.399 | 0.238 | 0.404 | 0.329 | 0.308 | 0.202 | 0.279 | 0.038 | 0.350 |
| 43 | 0.689 | 0.599 | 0.489 | 0.611 | 0.691 | 0.429 | 0.687 | 0.480 | 0.380 | 0.049 | 0.542 |
| 44 | 0.645 | 0.733 | 0.494 | 0.654 | 0.643 | 0.108 | 0.795 | 0.116 | 0.518 | 0.112 | 0.793 |
| 45 | 0.324 | 0.386 | 0.303 | 0.343 | 0.310 | 0.265 | 0.382 | 0.330 | 0.238 | 0.073 | 0.230 |
| 46 | 0.439 | 0.858 | 0.818 | 0.674 | 0.590 | 0.764 | 0.840 | 0.857 | 0.374 | 0.031 | 0.420 |
| 47 | 0.553 | 0.701 | 0.763 | 0.706 | 0.629 | 0.637 | 0.649 | 0.617 | 0.621 | 0.016 | 0.657 |
| 48 | 0.769 | 0.829 | 0.556 | 0.844 | 0.788 | 0.461 | 0.813 | 0.557 | 0.622 | 0.084 | 0.524 |
| 49 | 0.609 | 0.584 | 0.571 | 0.664 | 0.630 | 0.577 | 0.598 | 0.593 | 0.529 | 0.021 | 0.530 |
| 50 | 0.644 | 0.782 | 0.718 | 0.701 | 0.671 | 0.739 | 0.768 | 0.732 | 0.493 | 0.044 | 0.778 |
| 51 | 0.734 | 0.849 | 0.707 | 0.749 | 0.908 | 0.545 | 0.964 | 0.721 | 0.587 | 0.405 | 0.936 |
| 52 | 0.926 | 0.932 | 0.894 | 0.932 | 0.869 | 0.916 | 0.923 | 0.875 | 0.662 | 0.035 | 0.900 |
| 53 | 0.504 | 0.873 | 0.867 | 0.569 | 0.591 | 0.872 | 0.873 | 0.874 | 0.539 | 0.006 | 0.740 |
| 54 | 0.919 | 0.924 | 0.700 | 0.910 | 0.922 | 0.424 | 0.933 | 0.776 | 0.433 | 0.367 | 0.733 |
| 55 | 0.966 | 0.927 | 0.843 | 0.938 | 0.960 | 0.506 | 0.983 | 0.601 | 0.463 | 0.309 | 0.809 |
| 56 | 0.494 | 0.488 | 0.773 | 0.401 | 0.492 | 0.738 | 0.566 | 0.723 | 0.371 | 0.032 | 0.788 |
| 57 | 0.487 | 0.512 | 0.486 | 0.485 | 0.524 | 0.274 | 0.600 | 0.477 | 0.489 | 0.024 | 0.574 |
| 58 | 0.913 | 0.904 | 0.890 | 0.852 | 0.906 | 0.615 | 0.941 | 0.838 | 0.681 | 0.330 | 0.680 |
| 59 | 0.447 | 0.496 | 0.564 | 0.467 | 0.546 | 0.351 | 0.543 | 0.433 | 0.443 | 0.266 | 0.448 |
| 60 | 0.194 | 0.238 | 0.243 | 0.213 | 0.285 | 0.242 | 0.368 | 0.242 | 0.171 | 0.036 | 0.248 |
| 61 | 0.581 | 0.586 | 0.586 | 0.557 | 0.556 | 0.548 | 0.781 | 0.424 | 0.474 | 0.533 | 0.700 |
| 62 | 0.517 | 0.597 | 0.594 | 0.561 | 0.585 | 0.597 | 0.597 | 0.599 | 0.521 | 0.020 | 0.542 |
| 63 | 0.632 | 0.940 | 0.891 | 0.773 | 0.875 | 0.940 | 0.936 | 0.895 | 0.929 | 0.010 | 0.940 |
| 64 | 0.524 | 0.739 | 0.716 | 0.742 | 0.667 | 0.729 | 0.739 | 0.654 | 0.690 | 0.732 | 0.562 |
| 65 | 0.475 | 0.408 | 0.372 | 0.513 | 0.472 | 0.325 | 0.447 | 0.328 | 0.246 | 0.098 | 0.316 |
| 66 | 0.378 | 0.589 | 0.542 | 0.404 | 0.437 | 0.424 | 0.613 | 0.559 | 0.284 | 0.128 | 0.293 |

Table 12: The ACC on datasets [1]-[66] (Part-1).

| Dataset Index | S³COMP-C | k-FSC | AutoSC | DEC | IDEC | DSCN | PICA | ConClu | EDESC | DMICC | DIVC | LFSS |
|---|---|---|---|---|---|---|---|---|---|---|---|---|
| 1 | 0.659 | 0.587 | 0.836 | 0.928 | 0.925 | 0.774 | 0.836 | 0.869 | 0.830 | 0.866 | 0.803 | 0.918 |
| 2 | 0.246 | 0.258 | 0.252 | 0.372 | 0.326 | 0.257 | 0.296 | 0.354 | 0.316 | 0.320 | 0.265 | 0.283 |
| 3 | 0.583 | 0.704 | 0.635 | 0.969 | 0.618 | 0.927 | 0.970 | 0.974 | 0.975 | 0.970 | 0.971 | 0.969 |
| 4 | 0.569 | 0.614 | 0.991 | 0.991 | 0.990 | 0.904 | 0.967 | 0.992 | 0.956 | 0.974 | 0.912 | 0.975 |
| 5 | 0.357 | 0.450 | 0.439 | 0.488 | 0.490 | 0.464 | 0.436 | 0.505 | 0.448 | 0.464 | 0.449 | 0.472 |
| 6 | 0.573 | 0.619 | 0.625 | 0.590 | 0.618 | 0.420 | 0.658 | 0.526 | 0.544 | 0.609 | 0.659 | 0.616 |
| 7 | 0.682 | 0.696 | 0.692 | 0.576 | 0.579 | 0.723 | 0.521 | 0.555 | 0.570 | 0.576 | 0.537 | 0.577 |
| 8 | 0.288 | 0.293 | 0.282 | 0.317 | 0.311 | 0.320 | 0.327 | 0.356 | 0.343 | 0.331 | 0.331 | 0.337 |
| 9 | 0.560 | 0.664 | 0.572 | 0.697 | 0.716 | 0.611 | 0.654 | 0.701 | 0.642 | 0.691 | 0.663 | 0.677 |
| 10 | 0.278 | 0.278 | 0.274 | 0.278 | 0.322 | 0.306 | 0.328 | 0.341 | 0.298 | 0.340 | 0.317 | 0.331 |
| 11 | 0.574 | 0.660 | 0.579 | 0.551 | 0.548 | 0.600 | 0.606 | 0.720 | 0.876 | 0.699 | 0.606 | 0.601 |
| 12 | 0.554 | 0.736 | 0.985 | 0.974 | 0.877 | 0.975 | 0.979 | 0.985 | 0.988 | 0.967 | 0.978 | 0.984 |
| 13 | 0.545 | 0.625 | 0.706 | 0.709 | 0.721 | 0.738 | 0.751 | 0.825 | 0.751 | 0.791 | 0.769 | 0.752 |
| 14 | 0.346 | 0.337 | 0.492 | 0.426 | 0.481 | 0.535 | 0.491 | 0.430 | 0.478 | 0.480 | 0.455 | 0.512 |
| 15 | 0.575 | 0.673 | 0.747 | 0.787 | 0.788 | 0.677 | 0.795 | 0.867 | 0.744 | 0.817 | 0.799 | 0.873 |
| 16 | 0.570 | 0.699 | 0.619 | 0.664 | 0.877 | 0.823 | 0.787 | 0.860 | 0.784 | 0.802 | 0.810 | 0.866 |
| 17 | 0.460 | 0.613 | 0.813 | 0.916 | 0.881 | 0.599 | 0.821 | 0.853 | 0.819 | 0.844 | 0.825 | 0.880 |
| 18 | 0.565 | 0.837 | 0.961 | 0.917 | 0.912 | 0.826 | 0.931 | 0.955 | 0.928 | 0.893 | 0.930 | 0.899 |
| 19 | 0.251 | 0.303 | 0.328 | 0.203 | 0.578 | 0.375 | 0.257 | 0.178 | 0.309 | 0.206 | 0.260 | 0.316 |
| 20 | 0.569 | 0.559 | 0.511 | 0.790 | 0.798 | 0.723 | 0.786 | 0.806 | 0.788 | 0.802 | 0.787 | 0.805 |
| 21 | 0.477 | 0.485 | 0.613 | 0.504 | 0.496 | 0.406 | 0.623 | 0.594 | 0.543 | 0.615 | 0.617 | 0.632 |
| 22 | 0.800 | 0.667 | 0.738 | 0.705 | 0.765 | 0.823 | 0.650 | 0.750 | 0.730 | 0.710 | 0.693 | 0.637 |
| 23 | 0.542 | 0.908 | 0.876 | 0.799 | 0.947 | 0.702 | 0.711 | 0.792 | 0.844 | 0.824 | 0.699 | 0.699 |
| 24 | 0.619 | 0.609 | 0.910 | 0.877 | 0.947 | 0.934 | 0.900 | 0.916 | 0.928 | 0.915 | 0.876 | 0.875 |
| 25 | 0.496 | 0.721 | 0.825 | 0.777 | 0.802 | 0.737 | 0.962 | 0.966 | 0.824 | 0.906 | 0.957 | 0.905 |
| 26 | 0.504 | 0.700 | 0.631 | 0.677 | 0.659 | 0.666 | 0.652 | 0.732 | 0.711 | 0.710 | 0.631 | 0.668 |
| 27 | 0.390 | 0.657 | 0.564 | 0.691 | 0.764 | 0.574 | 0.798 | 0.713 | 0.828 | 0.814 | 0.749 | 0.693 |
| 28 | 0.633 | 0.909 | 0.845 | 0.785 | 0.945 | 0.795 | 0.699 | 0.742 | 0.846 | 0.846 | 0.652 | 0.754 |
| 29 | 0.521 | 0.885 | 0.527 | 0.904 | 0.893 | 0.621 | 0.723 | 0.741 | 0.791 | 0.720 | 0.730 | 0.769 |
| 30 | 0.318 | 0.357 | 0.279 | 0.362 | 0.761 | 0.362 | 0.320 | 0.336 | 0.348 | 0.356 | 0.332 | 0.340 |
| 31 | 0.560 | 0.647 | 0.750 | 0.638 | 0.642 | 0.585 | 0.770 | 0.800 | 0.695 | 0.748 | 0.740 | 0.762 |
| 32 | 0.549 | 0.604 | 0.909 | 0.815 | 0.761 | 0.881 | 0.893 | 0.909 | 0.915 | 0.900 | 0.901 | 0.912 |
| 33 | 0.473 | 0.486 | 0.560 | 0.565 | 0.546 | 0.470 | 0.635 | 0.565 | 0.468 | 0.573 | 0.616 | 0.629 |
| 34 | 0.545 | 0.638 | 0.577 | 0.630 | 0.592 | 0.660 | 0.783 | 0.789 | 0.704 | 0.675 | 0.656 | 0.880 |
| 35 | 0.382 | 0.411 | 0.451 | 0.301 | 0.623 | 0.411 | 0.442 | 0.389 | 0.428 | 0.348 | 0.392 | 0.303 |
| 36 | 0.735 | 0.741 | 0.837 | 0.554 | 0.643 | 0.844 | 0.671 | 0.634 | 0.608 | 0.777 | 0.604 | 0.651 |
| 37 | 0.691 | 0.577 | 0.692 | 0.663 | 0.697 | 0.676 | 0.593 | 0.613 | 0.592 | 0.596 | 0.545 | 0.589 |
| 38 | 0.548 | 0.727 | 0.968 | 0.740 | 0.719 | 0.658 | 0.792 | 0.818 | 0.887 | 0.804 | 0.794 | 0.851 |
| 39 | 0.868 | 0.917 | 0.918 | 0.932 | 0.932 | 0.923 | 0.562 | 0.641 | 0.730 | 0.722 | 0.559 | 0.684 |
| 40 | 0.428 | 0.477 | 0.581 | 0.664 | 0.590 | 0.783 | 0.632 | 0.663 | 0.631 | 0.656 | 0.604 | 0.702 |
| 41 | 0.801 | 0.662 | 0.760 | 0.613 | 0.635 | 0.769 | 0.557 | 0.622 | 0.612 | 0.601 | 0.577 | 0.663 |
| 42 | 0.336 | 0.427 | 0.429 | 0.388 | 0.352 | 0.253 | 0.265 | 0.406 | 0.268 | 0.416 | 0.247 | 0.423 |
| 43 | 0.524 | 0.524 | 0.492 | 0.679 | 0.669 | 0.663 | 0.566 | 0.730 | 0.738 | 0.676 | 0.597 | 0.618 |
| 44 | 0.919 | 0.859 | 0.777 | 0.565 | 0.502 | 0.337 | 0.685 | 0.468 | 0.687 | 0.591 | 0.647 | 0.643 |
| 45 | 0.263 | 0.332 | 0.244 | 0.294 | 0.308 | 0.327 | 0.245 | 0.318 | 0.330 | 0.257 | 0.303 | 0.274 |
| 46 | 0.437 | 0.724 | 0.591 | 0.657 | 0.811 | 0.488 | 0.634 | 0.547 | 0.580 | 0.524 | 0.584 | 0.480 |
| 47 | 0.530 | 0.589 | 0.537 | 0.568 | 0.723 | 0.557 | 0.577 | 0.576 | 0.609 | 0.643 | 0.639 | 0.620 |
| 48 | 0.524 | 0.564 | 0.844 | 0.790 | 0.736 | 0.755 | 0.861 | 0.877 | 0.867 | 0.835 | 0.867 | 0.866 |
| 49 | 0.528 | 0.596 | 0.521 | 0.622 | 0.640 | 0.612 | 0.600 | 0.603 | 0.595 | 0.614 | 0.609 | 0.677 |
| 50 | 0.680 | 0.607 | 0.774 | 0.520 | 0.616 | 0.734 | 0.601 | 0.474 | 0.577 | 0.596 | 0.636 | 0.509 |
| 51 | 0.879 | 0.915 | 0.955 | 0.821 | 0.803 | 0.820 | 0.798 | 0.704 | 0.830 | 0.836 | 0.692 | 0.849 |
| 52 | 0.931 | 0.511 | 0.930 | 0.807 | 0.881 | 0.928 | 0.667 | 0.843 | 0.698 | 0.817 | 0.591 | 0.625 |
| 53 | 0.830 | 0.821 | 0.630 | 0.588 | 0.838 | 0.630 | 0.633 | 0.568 | 0.606 | 0.761 | 0.723 | 0.605 |
| 54 | 0.496 | 0.690 | 0.895 | 0.863 | 0.826 | 0.755 | 0.913 | 0.876 | 0.855 | 0.892 | 0.910 | 0.933 |
| 55 | 0.571 | 0.837 | 0.961 | 0.901 | 0.925 | 0.866 | 0.915 | 0.972 | 0.911 | 0.899 | 0.927 | 0.949 |
| 56 | 0.711 | 0.485 | 0.407 | 0.460 | 0.838 | 0.718 | 0.511 | 0.473 | 0.490 | 0.486 | 0.490 | 0.447 |
| 57 | 0.353 | 0.580 | 0.346 | 0.721 | 0.587 | 0.597 | 0.812 | 0.787 | 0.669 | 0.807 | 0.749 | 0.809 |
| 58 | 0.631 | 0.678 | 0.904 | 0.924 | 0.926 | 0.861 | 0.818 | 0.896 | 0.909 | 0.916 | 0.828 | 0.904 |
| 59 | 0.451 | 0.426 | 0.458 | 0.473 | 0.481 | 0.484 | 0.453 | 0.523 | 0.534 | 0.465 | 0.459 | 0.479 |
| 60 | 0.190 | 0.199 | 0.208 | 0.210 | 0.587 | 0.236 | 0.197 | 0.238 | 0.252 | 0.198 | 0.190 | 0.209 |
| 61 | 0.463 | 0.641 | 0.605 | 0.538 | 0.551 | 0.541 | 0.643 | 0.467 | 0.568 | 0.583 | 0.656 | 0.610 |
| 62 | 0.551 | 0.565 | 0.527 | 0.528 | 0.550 | 0.597 | 0.564 | 0.531 | 0.559 | 0.547 | 0.560 | 0.571 |
| 63 | 0.940 | 0.655 | 0.934 | 0.564 | 0.808 | 0.929 | 0.842 | 0.814 | 0.753 | 0.761 | 0.767 | 0.568 |
| 64 | 0.535 | 0.735 | 0.523 | 0.649 | 0.671 | 0.708 | 0.608 | 0.624 | 0.637 | 0.626 | 0.607 | 0.578 |
| 65 | 0.282 | 0.344 | 0.407 | 0.417 | 0.442 | 0.449 | 0.364 | 0.339 | 0.461 | 0.358 | 0.366 | 0.371 |
| 66 | 0.335 | 0.539 | 0.337 | 0.381 | 0.388 | 0.434 | 0.347 | 0.515 | 0.498 | 0.428 | 0.358 | 0.360 |

Table 13: The ACC on datasets [1]-[66] (Part-2).

| Dataset Index | KMeans | AggClu | DBSCAN | BIRCH | GMM | OPTICS | SpeClu | MeanShift | k-PC | Affinity | SSC |
|---|---|---|---|---|---|---|---|---|---|---|---|
| 67 | 0.818 | 0.838 | 0.709 | 0.826 | 0.827 | 0.523 | 0.854 | 0.795 | 0.364 | 0.349 | 0.580 |
| 68 | 0.529 | 0.862 | 0.636 | 0.915 | 0.518 | 0.470 | 0.912 | 0.204 | 0.821 | 0.251 | 0.950 |
| 69 | 0.560 | 0.560 | 0.655 | 0.534 | 0.636 | 0.578 | 0.569 | 0.526 | 0.602 | 0.526 | 0.629 |
| 70 | 0.790 | 0.722 | 0.689 | 0.656 | 0.785 | 0.656 | 0.749 | 0.663 | 0.517 | 0.081 | 0.668 |
| 71 | 0.823 | 0.799 | 0.646 | 0.832 | 0.796 | 0.146 | 0.977 | 0.472 | 0.414 | 0.151 | 0.523 |
| 72 | 0.734 | 0.734 | 0.679 | 0.734 | 0.723 | 0.704 | 0.745 | 0.686 | 0.729 | 0.040 | 0.739 |
| 73 | 0.722 | 0.580 | 0.695 | 0.754 | 0.739 | 0.083 | 0.798 | 0.216 | 0.601 | 0.122 | 0.760 |
| 74 | 0.545 | 0.534 | 0.599 | 0.533 | 0.597 | 0.520 | 0.560 | 0.520 | 0.532 | 0.064 | 0.574 |
| 75 | 0.759 | 0.846 | 0.527 | 0.804 | 0.688 | 0.573 | 0.789 | 0.568 | 0.533 | 0.041 | 0.572 |
| 76 | 0.574 | 0.608 | 0.412 | 0.595 | 0.560 | 0.270 | 0.627 | 0.263 | 0.389 | 0.142 | 0.455 |
| 77 | 0.579 | 0.703 | 0.733 | 0.682 | 0.630 | 0.805 | 0.851 | 0.713 | 0.615 | 0.723 | 0.708 |
| 78 | 1.000 | 1.000 | 1.000 | 0.989 | 1.000 | 0.542 | 1.000 | 0.967 | 0.307 | 0.600 | 1.000 |
| 79 | 0.679 | 0.767 | 0.763 | 0.767 | 0.666 | 0.571 | 0.765 | 0.767 | 0.762 | 0.131 | 0.666 |
| 80 | 0.746 | 0.626 | 0.416 | 0.718 | 0.706 | 0.122 | 0.768 | 0.100 | 0.580 | 0.130 | 0.680 |
| 81 | 0.719 | 0.606 | 0.447 | 0.774 | 0.789 | 0.127 | 0.931 | 0.124 | 0.305 | 0.193 | 0.937 |
| 82 | 0.418 | 0.426 | 0.415 | 0.456 | 0.432 | 0.361 | 0.477 | 0.409 | 0.360 | 0.049 | 0.410 |
| 83 | 0.504 | 0.601 | 0.440 | 0.678 | 0.837 | 0.320 | 0.507 | 0.339 | 0.343 | 0.032 | 0.342 |
| 84 | 0.370 | 0.431 | 0.267 | 0.388 | 0.519 | 0.147 | 0.539 | 0.276 | 0.392 | 0.111 | 0.383 |
| 85 | 0.641 | 0.498 | 0.558 | 0.488 | 0.614 | 0.085 | 0.833 | 0.552 | 0.373 | 0.479 | 0.463 |
| 86 | 0.378 | 0.466 | 0.313 | 0.438 | 0.537 | 0.090 | 0.780 | 0.162 | 0.422 | 0.054 | 0.480 |
| 87 | 0.351 | 0.414 | 0.418 | 0.365 | 0.379 | 0.259 | 0.375 | 0.389 | 0.359 | 0.253 | 0.320 |
| 88 | 0.284 | 0.420 | 0.422 | 0.383 | 0.264 | 0.282 | 0.412 | 0.403 | 0.332 | 0.157 | 0.380 |
| 89 | 0.482 | 0.515 | 0.244 | 0.592 | 0.490 | 0.101 | 0.455 | 0.126 | 0.356 | 0.082 | 0.480 |
| 90 | 0.236 | 0.230 | 0.160 | 0.227 | 0.243 | 0.184 | 0.241 | 0.220 | 0.212 | 0.037 | 0.217 |
| 91 | 0.343 | 0.325 | 0.334 | 0.356 | 0.371 | 0.178 | 0.471 | 0.205 | 0.377 | 0.045 | 0.412 |
| 92 | 0.657 | 0.575 | 0.667 | 0.575 | 0.727 | 0.458 | 0.817 | 0.402 | 0.303 | 0.368 | 0.383 |
| 93 | 0.300 | 0.435 | 0.431 | 0.362 | 0.364 | 0.428 | 0.432 | 0.430 | 0.310 | 0.030 | 0.285 |
| 94 | 0.855 | 0.924 | 0.915 | 0.837 | 0.852 | 0.806 | 0.814 | 0.892 | 0.832 | 0.058 | 0.583 |
| 95 | 0.230 | 0.334 | 0.241 | 0.214 | 0.248 | 0.228 | 0.348 | 0.163 | 0.208 | 0.188 | 0.377 |
| 96 | 0.510 | 0.654 | 0.655 | 0.653 | 0.608 | 0.653 | 0.632 | 0.653 | 0.603 | 0.079 | 0.529 |
| 97 | 0.252 | 0.323 | 0.350 | 0.260 | 0.274 | 0.216 | 0.444 | 0.168 | 0.252 | 0.203 | 0.239 |
| 98 | 0.556 | 0.432 | 0.314 | 0.558 | 0.561 | 0.118 | 0.583 | 0.113 | 0.391 | 0.122 | 0.458 |
| 99 | 0.361 | 0.407 | 0.408 | 0.430 | 0.419 | 0.336 | 0.411 | 0.400 | 0.350 | 0.144 | 0.402 |
| 100 | 0.486 | 0.661 | 0.514 | 0.556 | 0.519 | 0.353 | 0.773 | 0.150 | 0.238 | 0.325 | 0.669 |
| 101 | 0.548 | 0.407 | 0.127 | 0.589 | 0.554 | 0.054 | 0.533 | 0.046 | 0.485 | 0.132 | 0.572 |
| 102 | 0.504 | 0.543 | 0.550 | 0.531 | 0.526 | 0.543 | 0.542 | 0.543 | 0.503 | 0.004 | 0.534 |
| 103 | 0.066 | 0.074 | 0.049 | 0.073 | 0.070 | 0.014 | 0.072 | 0.074 | 0.074 | 0.018 | 0.064 |
| 104 | 0.870 | 0.653 | 0.700 | 0.966 | 0.804 | 0.389 | 0.958 | 0.175 | 0.486 | 0.147 | 0.879 |
| 105 | 0.573 | 0.521 | 0.318 | 0.543 | 0.539 | 0.106 | 0.781 | 0.116 | 0.424 | 0.060 | 0.664 |
| 106 | 0.476 | 0.417 | 0.236 | 0.468 | 0.571 | 0.115 | 0.552 | 0.146 | 0.449 | 0.131 | 0.524 |
| 107 | 0.639 | 0.499 | 0.385 | 0.622 | 0.635 | 0.105 | 0.687 | 0.101 | 0.485 | 0.063 | 0.553 |
| 108 | 0.211 | 0.194 | 0.149 | 0.204 | 0.225 | 0.108 | 0.204 | 0.110 | 0.136 | 0.064 | 0.216 |
| 109 | 0.718 | 0.589 | 0.146 | 0.689 | 0.730 | 0.104 | 0.767 | 0.100 | 0.590 | 0.054 | 0.715 |
| 110 | 0.651 | 0.478 | 0.823 | 0.664 | 0.658 | 0.365 | 0.747 | 0.085 | 0.266 | 0.344 | 0.633 |
| 111 | 0.802 | 0.880 | 0.847 | 0.777 | 0.816 | 0.322 | 0.931 | 0.050 | 0.532 | 0.485 | 0.397 |
| 112 | 0.523 | 0.520 | 0.516 | 0.544 | 0.533 | 0.471 | 0.527 | 0.481 | 0.511 | 0.043 | 0.503 |
| 113 | 0.320 | 0.309 | 0.161 | 0.326 | 0.325 | 0.130 | 0.774 | 0.101 | 0.196 | 0.221 | 0.483 |
| 114 | 0.157 | 0.198 | 0.191 | 0.156 | 0.159 | 0.195 | 0.187 | 0.153 | 0.163 | 0.112 | 0.201 |
| 115 | 0.531 | 0.479 | 0.358 | 0.552 | 0.530 | 0.335 | 0.528 | 0.350 | 0.263 | 0.317 | 0.418 |
| 116 | 0.572 | 0.629 | 0.656 | 0.494 | 0.572 | 0.601 | 0.675 | 0.616 | 0.404 | 0.056 | 0.435 |
| 117 | 0.679 | 0.554 | 0.523 | 0.703 | 0.702 | 0.209 | 0.674 | 0.653 | 0.505 | 0.141 | 0.540 |
| 118 | 0.605 | 0.525 | 0.427 | 0.645 | 0.584 | 0.463 | 0.721 | 0.070 | 0.268 | 0.615 | 0.547 |
| 119 | 0.244 | 0.609 | 0.322 | 0.116 | 0.284 | 0.140 | 0.572 | 0.107 | 0.378 | 0.266 | 0.735 |
| 120 | 0.635 | 0.550 | 0.526 | 0.500 | 0.649 | 0.501 | 0.510 | 0.500 | 0.516 | 0.494 | 0.626 |
| 121 | 0.680 | 0.780 | 0.781 | 0.767 | 0.585 | 0.606 | 0.771 | 0.773 | 0.588 | 0.076 | 0.729 |
| 122 | 0.508 | 0.526 | 0.525 | 0.506 | 0.539 | 0.453 | 0.537 | 0.503 | 0.528 | 0.100 | 0.505 |
| 123 | 0.876 | 0.940 | 0.891 | 0.892 | 0.867 | 0.715 | 0.872 | 0.800 | 0.626 | 0.744 | 0.861 |
| 124 | 0.438 | 0.446 | 0.281 | 0.533 | 0.434 | 0.285 | 0.558 | 0.310 | 0.350 | 0.024 | 0.545 |
| 125 | 0.957 | 0.940 | 0.518 | 0.962 | 0.957 | 0.448 | 0.915 | 0.500 | 0.643 | 0.010 | 0.504 |
| 126 | 0.531 | 0.861 | 0.506 | 0.538 | 0.771 | 0.390 | 0.721 | 0.432 | 0.488 | 0.032 | 0.431 |
| 127 | 0.159 | 0.145 | 0.074 | 0.166 | 0.160 | 0.062 | 0.441 | 0.055 | 0.332 | 0.050 | 0.309 |
| 128 | 0.913 | 0.830 | 0.833 | 0.926 | 0.891 | 0.838 | 0.847 | 0.770 | 0.573 | 0.310 | 0.285 |
| 129 | 0.462 | 0.403 | 0.457 | 0.388 | 0.214 | 0.457 | 0.490 | 0.418 | 0.428 | 0.195 | 0.253 |
| 130 | 0.556 | 0.578 | 0.601 | 0.608 | 0.567 | 0.504 | 0.614 | 0.458 | 0.518 | 0.153 | 0.518 |
| 131 | 0.438 | 0.453 | 0.325 | 0.488 | 0.361 | 0.311 | 0.542 | 0.267 | 0.637 | 0.318 | 0.910 |

Table 14: The ACC on datasets [67]-[131] (Part-1).

| Dataset Index | S³COMP-C | k-FSC | AutoSC | DEC | IDEC | DSCN | PICA | ConClu | EDESC | DMICC | DIVC | LFSS |
|---|---|---|---|---|---|---|---|---|---|---|---|---|
| 67 | 0.435 | 0.591 | 0.596 | 0.801 | 0.798 | 0.746 | 0.626 | 0.716 | 0.782 | 0.703 | 0.620 | 0.661 |
| 68 | 0.927 | 0.909 | 0.972 | 0.749 | 0.759 | 0.896 | 0.503 | 0.370 | 0.668 | 0.464 | 0.482 | 0.470 |
| 69 | 0.653 | 0.612 | 0.552 | 0.534 | 0.536 | 0.634 | 0.583 | 0.647 | 0.595 | 0.579 | 0.579 | 0.707 |
| 70 | 0.669 | 0.617 | 0.684 | 0.682 | 0.661 | 0.711 | 0.738 | 0.770 | 0.727 | 0.797 | 0.704 | 0.696 |
| 71 | 0.506 | 0.421 | 0.526 | 0.800 | 0.812 | 0.742 | 0.589 | 0.915 | 0.744 | 0.806 | 0.488 | 0.914 |
| 72 | 0.755 | 0.735 | 0.743 | 0.594 | 0.739 | 0.736 | 0.678 | 0.600 | 0.793 | 0.710 | 0.649 | 0.650 |
| 73 | 0.456 | 0.804 | 0.883 | 0.702 | 0.586 | 0.694 | 0.689 | 0.607 | 0.706 | 0.637 | 0.699 | 0.790 |
| 74 | 0.519 | 0.551 | 0.507 | 0.569 | 0.739 | 0.543 | 0.560 | 0.599 | 0.573 | 0.579 | 0.566 | 0.616 |
| 75 | 0.595 | 0.623 | 0.604 | 0.707 | 0.586 | 0.790 | 0.795 | 0.802 | 0.695 | 0.717 | 0.803 | 0.727 |
| 76 | 0.323 | 0.521 | 0.300 | 0.553 | 0.566 | 0.511 | 0.486 | 0.482 | 0.577 | 0.538 | 0.482 | 0.512 |
| 77 | 0.685 | 0.749 | 0.723 | 0.651 | 0.720 | 0.760 | 0.709 | 0.703 | 0.704 | 0.670 | 0.693 | 0.733 |
| 78 | 1.000 | 1.000 | 1.000 | 1.000 | 1.000 | 0.734 | 0.970 | 0.860 | 1.000 | 1.000 | 0.977 | 1.000 |
| 79 | 0.645 | 0.762 | 0.742 | 0.673 | 0.760 | 0.738 | 0.593 | 0.592 | 0.625 | 0.641 | 0.586 | 0.628 |
| 80 | 0.732 | 0.927 | 0.683 | 0.782 | 1.000 | 0.826 | 0.536 | 0.385 | 0.753 | 0.468 | 0.488 | 0.414 |
| 81 | 0.849 | 0.897 | 0.939 | 0.677 | 0.560 | 0.909 | 0.689 | 0.555 | 0.720 | 0.731 | 0.691 | 0.685 |
| 82 | 0.401 | 0.390 | 0.406 | 0.403 | 0.425 | 0.412 | 0.412 | 0.432 | 0.429 | 0.399 | 0.403 | 0.358 |
| 83 | 0.348 | 0.407 | 0.501 | 0.641 | 0.615 | 0.515 | 0.676 | 0.654 | 0.608 | 0.597 | 0.690 | 0.527 |
| 84 | 0.381 | 0.479 | 0.238 | 0.341 | 0.328 | 0.390 | 0.488 | 0.461 | 0.544 | 0.447 | 0.402 | 0.642 |
| 85 | 0.458 | 0.543 | 0.636 | 0.642 | 0.642 | 0.504 | 0.638 | 0.652 | 0.579 | 0.613 | 0.643 | 0.650 |
| 86 | 0.426 | 0.490 | 0.799 | 0.352 | 0.384 | 0.740 | 0.390 | 0.368 | 0.523 | 0.278 | 0.389 | 0.453 |
| 87 | 0.297 | 0.408 | 0.371 | 0.351 | 0.378 | 0.395 | 0.374 | 0.459 | 0.414 | 0.376 | 0.374 | 0.369 |
| 88 | 0.373 | 0.281 | 0.372 | 0.315 | 0.302 | 0.394 | 0.417 | 0.325 | 0.301 | 0.304 | 0.417 | 0.303 |
| 89 | 0.542 | 0.469 | 0.574 | 0.314 | 0.307 | 0.260 | 0.427 | 0.357 | 0.488 | 0.401 | 0.418 | 0.322 |
| 90 | 0.252 | 0.205 | 0.301 | 0.200 | 0.217 | 0.237 | 0.234 | 0.213 | 0.210 | 0.211 | 0.228 | 0.187 |
| 91 | 0.331 | 0.382 | 0.302 | 0.315 | 0.368 | 0.394 | 0.205 | 0.355 | 0.380 | 0.354 | 0.205 | 0.370 |
| 92 | 0.443 | 0.482 | 0.575 | 0.615 | 0.638 | 0.626 | 0.795 | 0.732 | 0.601 | 0.809 | 0.886 | 0.887 |
| 93 | 0.310 | 0.434 | 0.383 | 0.340 | 0.344 | 0.426 | 0.366 | 0.373 | 0.369 | 0.328 | 0.357 | 0.332 |
| 94 | 0.738 | 0.812 | 0.531 | 0.702 | 0.856 | 0.846 | 0.696 | 0.778 | 0.734 | 0.796 | 0.661 | 0.711 |
| 95 | 0.780 | 0.514 | 0.265 | 0.227 | 0.233 | 0.266 | 0.302 | 0.263 | 0.401 | 0.289 | 0.301 | 0.210 |
| 96 | 0.519 | 0.691 | 0.616 | 0.529 | 0.547 | 0.585 | 0.581 | 0.556 | 0.613 | 0.640 | 0.601 | 0.676 |
| 97 | 0.515 | 0.546 | 0.362 | 0.256 | 0.242 | 0.300 | 0.317 | 0.389 | 0.529 | 0.358 | 0.307 | 0.210 |
| 98 | 0.520 | 0.631 | 0.566 | 0.567 | 0.533 | 0.586 | 0.490 | 0.429 | 0.539 | 0.467 | 0.500 | 0.445 |
| 99 | 0.347 | 0.375 | 0.506 | 0.412 | 0.392 | 0.406 | 0.402 | 0.424 | 0.388 | 0.397 | 0.400 | 0.427 |
| 100 | 0.761 | 0.443 | 0.703 | 0.396 | 0.454 | 0.507 | 0.625 | 0.589 | 0.636 | 0.553 | 0.586 | 0.358 |
| 101 | 0.490 | 0.578 | 0.502 | 0.458 | 0.377 | 0.446 | 0.364 | 0.292 | 0.565 | 0.360 | 0.366 | 0.417 |
| 102 | 0.538 | 0.521 | 0.510 | 0.513 | 0.544 | 0.519 | 0.519 | 0.508 | 0.516 | 0.529 | 0.514 | 0.520 |
| 103 | 0.071 | 0.073 | 0.069 | 0.068 | 0.070 | 0.071 | 0.069 | 0.075 | 0.074 | 0.074 | 0.069 | 0.072 |
| 104 | 0.942 | 0.949 | 0.792 | 0.586 | 0.474 | 0.606 | 0.778 | 0.616 | 0.841 | 0.795 | 0.790 | 0.735 |
| 105 | 0.790 | 0.888 | 0.804 | 0.613 | 0.467 | 0.756 | 0.249 | 0.452 | 0.780 | 0.491 | 0.226 | 0.456 |
| 106 | 0.613 | 0.591 | 0.585 | 0.437 | 0.501 | 0.542 | 0.509 | 0.516 | 0.508 | 0.509 | 0.489 | 0.492 |
| 107 | 0.655 | 0.677 | 0.649 | 0.635 | 0.549 | 0.598 | 0.423 | 0.627 | 0.673 | 0.552 | 0.421 | 0.601 |
| 108 | 0.197 | 0.202 | 0.252 | 0.167 | 0.169 | 0.244 | 0.187 | 0.218 | 0.209 | 0.210 | 0.191 | 0.197 |
| 109 | 0.865 | 0.792 | 0.835 | 0.749 | 0.554 | 0.731 | 0.502 | 0.624 | 0.806 | 0.509 | 0.611 | 0.546 |
| 110 | 0.700 | 0.692 | 0.748 | 0.614 | 0.597 | 0.528 | 0.677 | 0.404 | 0.628 | 0.574 | 0.647 | 0.617 |
| 111 | 0.842 | 0.680 | 0.916 | 0.749 | 0.726 | 0.618 | 0.867 | 0.556 | 0.810 | 0.742 | 0.865 | 0.756 |
| 112 | 0.509 | 0.515 | 0.535 | 0.522 | 0.522 | 0.520 | 0.515 | 0.521 | 0.546 | 0.531 | 0.515 | 0.526 |
| 113 | 0.615 | 0.496 | 0.612 | 0.175 | 0.188 | 0.394 | 0.308 | 0.244 | 0.453 | 0.265 | 0.326 | 0.217 |
| 114 | 0.249 | 0.181 | 0.180 | 0.155 | 0.155 | 0.181 | 0.163 | 0.175 | 0.169 | 0.163 | 0.164 | 0.176 |
| 115 | 0.565 | 0.362 | 0.596 | 0.574 | 0.541 | 0.536 | 0.495 | 0.540 | 0.555 | 0.531 | 0.513 | 0.600 |
| 116 | 0.344 | 0.396 | 0.382 | 0.483 | 0.600 | 0.586 | 0.410 | 0.593 | 0.567 | 0.526 | 0.442 | 0.546 |
| 117 | 0.452 | 0.619 | 0.668 | 0.590 | 0.556 | 0.651 | 0.698 | 0.727 | 0.769 | 0.710 | 0.718 | 0.752 |
| 118 | 0.691 | 0.462 | 0.823 | 0.391 | 0.403 | 0.455 | 0.478 | 0.320 | 0.438 | 0.424 | 0.480 | 0.502 |
| 119 | 0.744 | 0.613 | 0.384 | 0.226 | 0.198 | 0.565 | 0.331 | 0.254 | 0.596 | 0.331 | 0.329 | 0.232 |
| 120 | 0.506 | 0.580 | 0.501 | 0.587 | 0.570 | 0.561 | 0.580 | 0.647 | 0.688 | 0.576 | 0.586 | 0.539 |
| 121 | 0.691 | 0.405 | 0.729 | 0.498 | 0.403 | 0.753 | 0.717 | 0.606 | 0.451 | 0.559 | 0.653 | 0.468 |
| 122 | 0.511 | 0.548 | 0.526 | 0.519 | 0.549 | 0.525 | 0.529 | 0.531 | 0.536 | 0.540 | 0.526 | 0.519 |
| 123 | 0.861 | 0.619 | 0.820 | 0.894 | 0.938 | 0.856 | 0.643 | 0.757 | 0.792 | 0.729 | 0.666 | 0.588 |
| 124 | 0.606 | 0.509 | 0.573 | 0.470 | 0.389 | 0.445 | 0.327 | 0.415 | 0.577 | 0.369 | 0.310 | 0.265 |
| 125 | 0.505 | 0.734 | 0.510 | 0.898 | 0.959 | 0.525 | 0.885 | 0.878 | 0.977 | 0.890 | 0.906 | 0.820 |
| 126 | 0.700 | 0.566 | 0.433 | 0.696 | 0.710 | 0.655 | 0.576 | 0.748 | 0.883 | 0.726 | 0.537 | 0.496 |
| 127 | 0.282 | 0.364 | 0.334 | 0.101 | 0.133 | 0.225 | 0.123 | 0.132 | 0.204 | 0.129 | 0.100 | 0.129 |
| 128 | 0.733 | 0.230 | 0.993 | 0.579 | 0.592 | 0.910 | 0.804 | 0.830 | 0.904 | 0.695 | 0.804 | 0.338 |
| 129 | 0.261 | 0.434 | 0.394 | 0.184 | 0.580 | 0.429 | 0.425 | 0.233 | 0.386 | 0.260 | 0.413 | 0.127 |
| 130 | 0.511 | 0.589 | 0.522 | 0.631 | 0.617 | 0.551 | 0.529 | 0.538 | 0.554 | 0.610 | 0.529 | 0.664 |
| 131 | 0.833 | 0.540 | 0.594 | 0.222 | 0.217 | 0.818 | 0.299 | 0.359 | 0.648 | 0.332 | 0.299 | 0.172 |

Table 15: The ACC on datasets [67]-[131] (Part-2).

| Dataset Index | KMeans | AggClu | DBSCAN | BIRCH | GMM | OPTICS | SpeClu | MeanShift | k-PC | Affinity | SSC |
|---|---|---|---|---|---|---|---|---|---|---|---|
| 1 | 0.388 | 0.598 | 0.449 | 0.640 | 0.423 | 0.255 | 0.640 | 0.309 | 0.012 | 0.284 | 0.026 |
| 2 | 0.143 | 0.134 | 0.118 | 0.128 | 0.131 | 0.117 | 0.128 | 0.075 | 0.036 | 0.134 | 0.040 |
| 3 | 0.731 | 0.814 | 0.764 | 0.784 | 0.733 | 0.191 | 0.802 | 0.695 | 0.020 | 0.334 | 0.190 |
| 4 | 0.906 | 0.925 | 0.882 | 0.939 | 0.939 | 0.758 | 0.939 | 0.763 | 0.018 | 0.303 | 0.060 |
| 5 | 0.305 | 0.392 | 0.454 | 0.458 | 0.361 | 0.309 | 0.347 | 0.479 | 0.151 | 0.390 | 0.233 |
| 6 | 0.603 | 0.459 | 0.558 | 0.623 | 0.612 | 0.374 | 0.566 | 0.526 | 0.398 | 0.566 | 0.642 |
| 7 | 0.003 | 0.022 | 0.011 | 0.001 | 0.027 | 0.026 | 0.042 | 0.000 | 0.010 | 0.024 | 0.001 |
| 8 | 0.021 | 0.022 | 0.042 | 0.024 | 0.023 | 0.049 | 0.020 | 0.021 | 0.009 | 0.074 | 0.006 |
| 9 | 0.063 | 0.042 | 0.068 | 0.090 | 0.131 | 0.002 | 0.133 | 0.066 | 0.001 | 0.081 | 0.002 |
| 10 | 0.008 | 0.027 | 0.092 | 0.008 | 0.008 | 0.077 | 0.009 | 0.000 | 0.004 | 0.095 | 0.004 |
| 11 | 0.010 | 0.322 | 0.100 | 0.010 | 0.179 | 0.150 | 0.563 | 0.053 | 0.014 | 0.147 | 0.002 |
| 12 | 0.857 | 0.816 | 0.673 | 0.795 | 0.879 | 0.076 | 0.844 | 0.846 | 0.001 | 0.241 | 0.023 |
| 13 | 0.327 | 0.270 | 0.182 | 0.282 | 0.149 | 0.185 | 0.357 | 0.109 | 0.010 | 0.172 | 0.007 |
| 14 | 0.215 | 0.373 | 0.293 | 0.314 | 0.425 | 0.214 | 0.310 | 0.103 | 0.033 | 0.323 | 0.040 |
| 15 | 0.773 | 0.719 | 0.678 | 0.809 | 0.766 | 0.576 | 0.842 | 0.720 | 0.041 | 0.697 | 0.421 |
| 16 | 0.279 | 0.335 | 0.157 | 0.388 | 0.428 | 0.113 | 0.294 | 0.126 | 0.000 | 0.159 | 0.023 |
| 17 | 0.657 | 0.763 | 0.734 | 0.738 | 0.900 | 0.536 | 0.673 | 0.734 | 0.089 | 0.547 | 0.280 |
| 18 | 0.879 | 0.787 | 0.608 | 0.808 | 0.855 | 0.253 | 0.928 | 0.502 | 0.099 | 0.504 | 0.462 |
| 19 | 0.376 | 0.406 | 0.487 | 0.442 | 0.470 | 0.434 | 0.720 | 0.000 | 0.343 | 0.578 | 0.380 |
| 20 | 0.293 | 0.247 | 0.215 | 0.002 | 0.217 | 0.068 | 0.285 | 0.237 | 0.025 | 0.002 | 0.004 |
| 21 | 0.531 | 0.610 | 0.515 | 0.536 | 0.545 | 0.446 | 0.532 | 0.464 | 0.242 | 0.502 | 0.372 |
| 22 | 0.160 | 0.223 | 0.091 | 0.229 | 0.227 | 0.201 | 0.242 | 0.207 | 0.015 | 0.162 | 0.015 |
| 23 | 0.397 | 0.577 | 0.329 | 0.642 | 0.257 | 0.089 | 0.475 | 0.529 | 0.236 | 0.087 | 0.208 |
| 24 | 0.546 | 0.682 | 0.534 | 0.629 | 0.661 | 0.136 | 0.708 | 0.277 | 0.023 | 0.267 | 0.139 |
| 25 | 0.804 | 0.740 | 0.533 | 0.866 | 0.924 | 0.215 | 0.908 | 0.784 | 0.120 | 0.472 | 0.264 |
| 26 | 0.702 | 0.683 | 0.621 | 0.751 | 0.749 | 0.260 | 0.791 | 0.576 | 0.113 | 0.562 | 0.647 |
| 27 | 0.847 | 0.909 | 0.847 | 0.857 | 0.845 | 0.765 | 0.824 | 0.869 | 0.320 | 0.627 | 0.630 |
| 28 | 0.406 | 0.368 | 0.294 | 0.600 | 0.256 | 0.074 | 0.586 | 0.400 | 0.286 | 0.084 | 0.334 |
| 29 | 0.125 | 0.100 | 0.592 | 0.151 | 0.418 | 0.076 | 0.126 | 0.296 | 0.242 | 0.234 | 0.018 |
| 30 | 0.315 | 0.296 | 0.256 | 0.343 | 0.302 | 0.261 | 0.454 | 0.254 | 0.085 | 0.356 | 0.230 |
| 31 | 0.101 | 0.186 | 0.268 | 0.044 | 0.156 | 0.174 | 0.448 | 0.158 | 0.020 | 0.121 | 0.076 |
| 32 | 0.569 | 0.488 | 0.434 | 0.520 | 0.562 | 0.086 | 0.594 | 0.553 | 0.060 | 0.193 | 0.019 |
| 33 | 0.604 | 0.613 | 0.595 | 0.604 | 0.605 | 0.546 | 0.624 | 0.566 | 0.202 | 0.617 | 0.466 |
| 34 | 0.011 | 0.259 | 0.698 | 0.213 | 0.093 | 0.022 | 0.190 | 0.100 | 0.036 | 0.301 | 0.048 |
| 35 | 0.051 | 0.085 | 0.125 | 0.069 | 0.053 | 0.124 | 0.050 | 0.045 | 0.028 | 0.108 | 0.006 |
| 36 | 0.027 | 0.013 | 0.032 | 0.029 | 0.034 | 0.015 | 0.019 | 0.039 | 0.000 | 0.026 | 0.010 |
| 37 | 0.011 | 0.004 | 0.030 | 0.006 | 0.006 | 0.037 | 0.005 | 0.011 | 0.003 | 0.046 | 0.001 |
| 38 | 0.470 | 0.842 | 0.614 | 0.700 | 0.571 | 0.447 | 0.822 | 0.384 | 0.119 | 0.473 | 0.227 |
| 39 | 0.003 | 0.001 | 0.000 | 0.001 | 0.012 | 0.000 | 0.001 | 0.007 | 0.003 | 0.029 | 0.010 |
| 40 | 0.714 | 0.586 | 0.525 | 0.709 | 0.739 | 0.235 | 0.724 | 0.433 | 0.246 | 0.459 | 0.399 |
| 41 | 0.004 | 0.000 | 0.016 | 0.000 | 0.000 | 0.020 | 0.001 | 0.004 | 0.001 | 0.001 | 0.000 |
| 42 | 0.111 | 0.154 | 0.352 | 0.177 | 0.272 | 0.244 | 0.140 | 0.236 | 0.081 | 0.176 | 0.140 |
| 43 | 0.239 | 0.159 | 0.148 | 0.111 | 0.245 | 0.119 | 0.203 | 0.044 | 0.008 | 0.209 | 0.102 |
| 44 | 0.648 | 0.745 | 0.599 | 0.717 | 0.638 | 0.302 | 0.832 | 0.046 | 0.475 | 0.580 | 0.812 |
| 45 | 0.097 | 0.087 | 0.095 | 0.088 | 0.097 | 0.143 | 0.076 | 0.056 | 0.050 | 0.156 | 0.044 |
| 46 | 0.084 | 0.017 | 0.044 | 0.107 | 0.106 | 0.065 | 0.006 | 0.000 | 0.004 | 0.084 | 0.010 |
| 47 | 0.002 | 0.120 | 0.185 | 0.131 | 0.027 | 0.018 | 0.001 | 0.101 | 0.016 | 0.077 | 0.029 |
| 48 | 0.339 | 0.367 | 0.294 | 0.377 | 0.374 | 0.107 | 0.392 | 0.138 | 0.048 | 0.173 | 0.002 |
| 49 | 0.030 | 0.021 | 0.059 | 0.060 | 0.038 | 0.054 | 0.003 | 0.002 | 0.002 | 0.054 | 0.010 |
| 50 | 0.202 | 0.041 | 0.154 | 0.234 | 0.201 | 0.159 | 0.263 | 0.171 | 0.032 | 0.159 | 0.133 |
| 51 | 0.864 | 0.905 | 0.791 | 0.869 | 0.903 | 0.487 | 0.927 | 0.812 | 0.518 | 0.664 | 0.857 |
| 52 | 0.007 | 0.003 | 0.026 | 0.003 | 0.010 | 0.011 | 0.016 | 0.053 | 0.001 | 0.029 | 0.000 |
| 53 | 0.000 | 0.000 | 0.032 | 0.009 | 0.070 | 0.039 | 0.000 | 0.000 | 0.011 | 0.080 | 0.000 |
| 54 | 0.728 | 0.746 | 0.534 | 0.750 | 0.739 | 0.293 | 0.763 | 0.624 | 0.077 | 0.519 | 0.320 |
| 55 | 0.876 | 0.787 | 0.608 | 0.808 | 0.858 | 0.253 | 0.928 | 0.502 | 0.090 | 0.504 | 0.462 |
| 56 | 0.245 | 0.265 | 0.270 | 0.353 | 0.325 | 0.096 | 0.446 | 0.000 | 0.071 | 0.214 | 0.199 |
| 57 | 0.521 | 0.479 | 0.469 | 0.514 | 0.520 | 0.227 | 0.537 | 0.491 | 0.085 | 0.317 | 0.320 |
| 58 | 0.604 | 0.582 | 0.550 | 0.374 | 0.586 | 0.229 | 0.658 | 0.531 | 0.004 | 0.320 | 0.020 |
| 59 | 0.281 | 0.262 | 0.381 | 0.254 | 0.417 | 0.294 | 0.445 | 0.302 | 0.163 | 0.337 | 0.211 |
| 60 | 0.070 | 0.046 | 0.338 | 0.078 | 0.145 | 0.336 | 0.259 | 0.054 | 0.030 | 0.259 | 0.015 |
| 61 | 0.654 | 0.566 | 0.629 | 0.608 | 0.607 | 0.612 | 0.709 | 0.531 | 0.386 | 0.661 | 0.603 |
| 62 | 0.001 | 0.002 | 0.011 | 0.001 | 0.012 | 0.001 | 0.001 | 0.005 | 0.002 | 0.036 | 0.005 |
| 63 | 0.013 | 0.002 | 0.016 | 0.014 | 0.005 | 0.000 | 0.003 | 0.027 | 0.009 | 0.019 | 0.000 |
| 64 | 0.001 | 0.025 | 0.067 | 0.040 | 0.070 | 0.001 | 0.015 | 0.060 | 0.010 | 0.008 | 0.006 |
| 65 | 0.292 | 0.230 | 0.136 | 0.297 | 0.290 | 0.125 | 0.284 | 0.098 | 0.071 | 0.248 | 0.132 |
| 66 | 0.213 | 0.234 | 0.229 | 0.164 | 0.144 | 0.144 | 0.326 | 0.127 | 0.023 | 0.210 | 0.033 |

Table 16: The NMI on datasets [1]-[66] (Part-1).

| Dataset Index | S³COMP-C | k-FSC | AutoSC | DEC | IDEC | DSCN | PICA | ConClu | EDESC | DMICC | DIVC | LFSS |
|---|---|---|---|---|---|---|---|---|---|---|---|---|
| 1 | 0.013 | 0.203 | 0.368 | 0.645 | 0.639 | 0.193 | 0.442 | 0.414 | 0.403 | 0.415 | 0.312 | 0.640 |
| 2 | 0.027 | 0.059 | 0.073 | 0.180 | 0.133 | 0.095 | 0.121 | 0.175 | 0.151 | 0.161 | 0.067 | 0.109 |
| 3 | 0.025 | 0.123 | 0.010 | 0.789 | 0.617 | 0.653 | 0.798 | 0.834 | 0.832 | 0.804 | 0.803 | 0.815 |
| 4 | 0.024 | 0.038 | 0.924 | 0.921 | 0.916 | 0.677 | 0.811 | 0.932 | 0.796 | 0.838 | 0.669 | 0.841 |
| 5 | 0.141 | 0.298 | 0.360 | 0.347 | 0.338 | 0.340 | 0.330 | 0.417 | 0.326 | 0.347 | 0.341 | 0.396 |
| 6 | 0.417 | 0.524 | 0.623 | 0.603 | 0.617 | 0.414 | 0.622 | 0.519 | 0.555 | 0.565 | 0.609 | 0.559 |
| 7 | 0.005 | 0.003 | 0.008 | 0.001 | 0.002 | 0.030 | 0.001 | 0.002 | 0.003 | 0.004 | 0.002 | 0.013 |
| 8 | 0.006 | 0.006 | 0.008 | 0.018 | 0.017 | 0.017 | 0.023 | 0.031 | 0.026 | 0.031 | 0.022 | 0.028 |
| 9 | 0.003 | 0.013 | 0.001 | 0.098 | 0.101 | 0.086 | 0.084 | 0.135 | 0.080 | 0.096 | 0.093 | 0.102 |
| 10 | 0.004 | 0.005 | 0.004 | 0.006 | 0.034 | 0.020 | 0.025 | 0.040 | 0.015 | 0.041 | 0.023 | 0.042 |
| 11 | 0.022 | 0.063 | 0.021 | 0.010 | 0.001 | 0.009 | 0.000 | 0.145 | 0.471 | 0.133 | 0.000 | 0.041 |
| 12 | 0.011 | 0.186 | 0.891 | 0.843 | 0.547 | 0.842 | 0.858 | 0.887 | 0.906 | 0.801 | 0.854 | 0.882 |
| 13 | 0.014 | 0.043 | 0.197 | 0.154 | 0.162 | 0.213 | 0.197 | 0.329 | 0.197 | 0.260 | 0.221 | 0.189 |
| 14 | 0.035 | 0.073 | 0.293 | 0.197 | 0.302 | 0.275 | 0.264 | 0.190 | 0.189 | 0.215 | 0.192 | 0.336 |
| 15 | 0.325 | 0.530 | 0.626 | 0.715 | 0.714 | 0.535 | 0.758 | 0.772 | 0.660 | 0.751 | 0.750 | 0.832 |
| 16 | 0.018 | 0.117 | 0.081 | 0.112 | 0.547 | 0.330 | 0.277 | 0.415 | 0.301 | 0.289 | 0.303 | 0.432 |
| 17 | 0.063 | 0.220 | 0.620 | 0.797 | 0.736 | 0.413 | 0.624 | 0.732 | 0.635 | 0.718 | 0.630 | 0.716 |
| 18 | 0.214 | 0.533 | 0.861 | 0.752 | 0.737 | 0.568 | 0.791 | 0.839 | 0.804 | 0.724 | 0.805 | 0.716 |
| 19 | 0.245 | 0.331 | 0.449 | 0.274 | 0.027 | 0.485 | 0.329 | 0.238 | 0.391 | 0.264 | 0.341 | 0.385 |
| 20 | 0.014 | 0.021 | 0.005 | 0.269 | 0.287 | 0.188 | 0.252 | 0.290 | 0.267 | 0.296 | 0.263 | 0.290 |
| 21 | 0.304 | 0.347 | 0.526 | 0.552 | 0.535 | 0.330 | 0.523 | 0.529 | 0.579 | 0.546 | 0.543 | 0.576 |
| 22 | 0.015 | 0.111 | 0.140 | 0.120 | 0.120 | 0.123 | 0.148 | 0.223 | 0.197 | 0.149 | 0.206 | 0.141 |
| 23 | 0.015 | 0.259 | 0.246 | 0.205 | 0.426 | 0.108 | 0.099 | 0.213 | 0.250 | 0.258 | 0.144 | 0.146 |
| 24 | 0.005 | 0.050 | 0.554 | 0.491 | 0.426 | 0.639 | 0.528 | 0.570 | 0.634 | 0.573 | 0.456 | 0.471 |
| 25 | 0.162 | 0.438 | 0.658 | 0.631 | 0.598 | 0.739 | 0.879 | 0.887 | 0.700 | 0.767 | 0.873 | 0.764 |
| 26 | 0.448 | 0.620 | 0.640 | 0.632 | 0.615 | 0.593 | 0.621 | 0.625 | 0.669 | 0.640 | 0.604 | 0.656 |
| 27 | 0.434 | 0.674 | 0.654 | 0.670 | 0.728 | 0.455 | 0.746 | 0.671 | 0.824 | 0.825 | 0.702 | 0.702 |
| 28 | 0.013 | 0.251 | 0.210 | 0.200 | 0.405 | 0.108 | 0.082 | 0.176 | 0.283 | 0.266 | 0.112 | 0.161 |
| 29 | 0.091 | 0.465 | 0.113 | 0.529 | 0.508 | 0.040 | 0.171 | 0.250 | 0.251 | 0.160 | 0.190 | 0.254 |
| 30 | 0.178 | 0.229 | 0.164 | 0.291 | 0.330 | 0.285 | 0.271 | 0.260 | 0.274 | 0.277 | 0.271 | 0.276 |
| 31 | 0.015 | 0.067 | 0.242 | 0.072 | 0.088 | 0.081 | 0.229 | 0.300 | 0.159 | 0.225 | 0.203 | 0.220 |
| 32 | 0.007 | 0.032 | 0.555 | 0.377 | 0.330 | 0.494 | 0.523 | 0.556 | 0.586 | 0.531 | 0.550 | 0.565 |
| 33 | 0.464 | 0.495 | 0.622 | 0.599 | 0.608 | 0.559 | 0.635 | 0.638 | 0.568 | 0.613 | 0.634 | 0.647 |
| 34 | 0.005 | 0.071 | 0.026 | 0.072 | 0.032 | 0.235 | 0.288 | 0.256 | 0.173 | 0.120 | 0.121 | 0.503 |
| 35 | 0.005 | 0.002 | 0.009 | 0.049 | 0.012 | 0.032 | 0.014 | 0.061 | 0.057 | 0.053 | 0.035 | 0.064 |
| 36 | 0.000 | 0.000 | 0.007 | 0.007 | 0.017 | 0.007 | 0.008 | 0.030 | 0.016 | 0.022 | 0.017 | 0.030 |
| 37 | 0.006 | 0.004 | 0.000 | 0.000 | 0.001 | 0.006 | 0.027 | 0.021 | 0.020 | 0.014 | 0.009 | 0.019 |
| 38 | 0.148 | 0.373 | 0.859 | 0.455 | 0.430 | 0.361 | 0.545 | 0.559 | 0.672 | 0.563 | 0.582 | 0.622 |
| 39 | 0.004 | 0.005 | 0.004 | 0.001 | 0.001 | 0.003 | 0.012 | 0.036 | 0.020 | 0.007 | 0.012 | 0.004 |
| 40 | 0.261 | 0.344 | 0.469 | 0.583 | 0.501 | 0.637 | 0.583 | 0.610 | 0.550 | 0.616 | 0.538 | 0.661 |
| 41 | 0.000 | 0.000 | 0.000 | 0.003 | 0.003 | 0.000 | 0.002 | 0.004 | 0.003 | 0.003 | 0.005 | 0.006 |
| 42 | 0.110 | 0.181 | 0.220 | 0.266 | 0.278 | 0.103 | 0.021 | 0.228 | 0.097 | 0.244 | 0.014 | 0.247 |
| 43 | 0.013 | 0.048 | 0.018 | 0.220 | 0.202 | 0.222 | 0.228 | 0.279 | 0.243 | 0.269 | 0.224 | 0.210 |
| 44 | 0.840 | 0.793 | 0.777 | 0.539 | 0.464 | 0.396 | 0.618 | 0.399 | 0.697 | 0.525 | 0.565 | 0.542 |
| 45 | 0.026 | 0.009 | 0.070 | 0.076 | 0.083 | 0.057 | 0.079 | 0.069 | 0.058 | 0.082 | 0.029 | 0.090 |
| 46 | 0.001 | 0.011 | 0.064 | 0.149 | 0.136 | 0.088 | 0.060 | 0.118 | 0.054 | 0.086 | 0.068 | 0.081 |
| 47 | 0.013 | 0.010 | 0.001 | 0.012 | 0.131 | 0.013 | 0.037 | 0.008 | 0.039 | 0.068 | 0.062 | 0.054 |
| 48 | 0.003 | 0.018 | 0.393 | 0.335 | 0.256 | 0.295 | 0.423 | 0.463 | 0.439 | 0.399 | 0.436 | 0.436 |
| 49 | 0.001 | 0.000 | 0.002 | 0.036 | 0.039 | 0.021 | 0.031 | 0.028 | 0.024 | 0.029 | 0.040 | 0.090 |
| 50 | 0.114 | 0.130 | 0.011 | 0.123 | 0.167 | 0.069 | 0.099 | 0.135 | 0.143 | 0.166 | 0.066 | 0.161 |
| 51 | 0.752 | 0.817 | 0.910 | 0.845 | 0.847 | 0.705 | 0.847 | 0.675 | 0.892 | 0.788 | 0.737 | 0.703 |
| 52 | 0.001 | 0.001 | 0.001 | 0.004 | 0.011 | 0.005 | 0.003 | 0.000 | 0.001 | 0.009 | 0.004 | 0.003 |
| 53 | 0.028 | 0.001 | 0.138 | 0.094 | 0.064 | 0.147 | 0.034 | 0.092 | 0.035 | 0.000 | 0.021 | 0.036 |
| 54 | 0.090 | 0.285 | 0.679 | 0.640 | 0.600 | 0.450 | 0.726 | 0.657 | 0.709 | 0.709 | 0.714 | 0.767 |
| 55 | 0.207 | 0.533 | 0.861 | 0.710 | 0.746 | 0.652 | 0.775 | 0.897 | 0.809 | 0.719 | 0.780 | 0.807 |
| 56 | 0.010 | 0.179 | 0.252 | 0.186 | 0.064 | 0.195 | 0.162 | 0.243 | 0.319 | 0.213 | 0.182 | 0.184 |
| 57 | 0.015 | 0.233 | 0.010 | 0.527 | 0.464 | 0.469 | 0.513 | 0.473 | 0.454 | 0.525 | 0.451 | 0.511 |
| 58 | 0.002 | 0.015 | 0.554 | 0.613 | 0.631 | 0.400 | 0.327 | 0.562 | 0.590 | 0.608 | 0.351 | 0.582 |
| 59 | 0.243 | 0.166 | 0.328 | 0.257 | 0.261 | 0.271 | 0.261 | 0.289 | 0.290 | 0.271 | 0.269 | 0.288 |
| 60 | 0.020 | 0.019 | 0.043 | 0.085 | 0.464 | 0.077 | 0.077 | 0.077 | 0.079 | 0.087 | 0.081 | 0.084 |
| 61 | 0.394 | 0.583 | 0.568 | 0.588 | 0.609 | 0.490 | 0.617 | 0.551 | 0.594 | 0.595 | 0.604 | 0.603 |
| 62 | 0.001 | 0.011 | 0.001 | 0.001 | 0.003 | 0.008 | 0.000 | 0.001 | 0.005 | 0.004 | 0.002 | 0.011 |
| 63 | 0.001 | 0.002 | 0.011 | 0.007 | 0.007 | 0.007 | 0.003 | 0.002 | 0.001 | 0.000 | 0.005 | 0.004 |
| 64 | 0.005 | 0.007 | 0.000 | 0.032 | 0.033 | 0.036 | 0.022 | 0.073 | 0.045 | 0.028 | 0.007 | 0.017 |
| 65 | 0.090 | 0.152 | 0.217 | 0.241 | 0.260 | 0.202 | 0.219 | 0.189 | 0.221 | 0.237 | 0.224 | 0.237 |
| 66 | 0.023 | 0.033 | 0.166 | 0.176 | 0.154 | 0.182 | 0.199 | 0.231 | 0.235 | 0.196 | 0.209 | 0.164 |

Table 17: The NMI on datasets [1]-[66] (Part-2).

| Dataset Index | KMeans | AggClu | DBSCAN | BIRCH | GMM | OPTICS | SpeClu | MeanShift | k-PC | Affinity | SSC |
|---|---|---|---|---|---|---|---|---|---|---|---|
| 67 | 0.675 | 0.725 | 0.634 | 0.707 | 0.697 | 0.354 | 0.696 | 0.702 | 0.125 | 0.533 | 0.330 |
| 68 | 0.424 | 0.750 | 0.561 | 0.805 | 0.409 | 0.402 | 0.820 | 0.000 | 0.700 | 0.564 | 0.860 |
| 69 | 0.019 | 0.019 | 0.089 | 0.027 | 0.107 | 0.122 | 0.092 | 0.038 | 0.045 | 0.065 | 0.041 |
| 70 | 0.220 | 0.093 | 0.040 | 0.203 | 0.200 | 0.056 | 0.290 | 0.075 | 0.021 | 0.124 | 0.017 |
| 71 | 0.700 | 0.725 | 0.494 | 0.712 | 0.710 | 0.477 | 0.918 | 0.619 | 0.017 | 0.475 | 0.148 |
| 72 | 0.000 | 0.000 | 0.110 | 0.000 | 0.061 | 0.068 | 0.033 | 0.120 | 0.000 | 0.097 | 0.008 |
| 73 | 0.705 | 0.672 | 0.733 | 0.782 | 0.708 | 0.325 | 0.817 | 0.252 | 0.573 | 0.614 | 0.760 |
| 74 | 0.007 | 0.007 | 0.032 | 0.006 | 0.048 | 0.027 | 0.010 | 0.028 | 0.003 | 0.047 | 0.013 |
| 75 | 0.247 | 0.378 | 0.090 | 0.309 | 0.118 | 0.034 | 0.335 | 0.008 | 0.002 | 0.147 | 0.011 |
| 76 | 0.318 | 0.338 | 0.168 | 0.360 | 0.306 | 0.200 | 0.429 | 0.115 | 0.055 | 0.277 | 0.124 |
| 77 | 0.108 | 0.260 | 0.225 | 0.242 | 0.115 | 0.285 | 0.293 | 0.031 | 0.014 | 0.024 | 0.051 |
| 78 | 1.000 | 1.000 | 1.000 | 0.971 | 1.000 | 0.648 | 1.000 | 0.966 | 0.236 | 0.831 | 1.000 |
| 79 | 0.009 | 0.021 | 0.024 | 0.021 | 0.002 | 0.026 | 0.012 | 0.032 | 0.003 | 0.074 | 0.002 |
| 80 | 0.665 | 0.675 | 0.504 | 0.732 | 0.680 | 0.296 | 0.802 | 0.000 | 0.532 | 0.578 | 0.810 |
| 81 | 0.714 | 0.711 | 0.651 | 0.789 | 0.727 | 0.325 | 0.874 | 0.050 | 0.244 | 0.637 | 0.874 |
| 82 | 0.095 | 0.155 | 0.174 | 0.118 | 0.109 | 0.152 | 0.128 | 0.004 | 0.063 | 0.227 | 0.054 |
| 83 | 0.365 | 0.324 | 0.213 | 0.391 | 0.511 | 0.056 | 0.370 | 0.000 | 0.000 | 0.223 | 0.000 |
| 84 | 0.213 | 0.329 | 0.352 | 0.292 | 0.360 | 0.365 | 0.486 | 0.259 | 0.165 | 0.456 | 0.199 |
| 85 | 0.683 | 0.649 | 0.614 | 0.582 | 0.642 | 0.370 | 0.847 | 0.627 | 0.383 | 0.611 | 0.393 |
| 86 | 0.373 | 0.424 | 0.489 | 0.534 | 0.621 | 0.360 | 0.807 | 0.000 | 0.325 | 0.494 | 0.420 |
| 87 | 0.034 | 0.046 | 0.031 | 0.046 | 0.051 | 0.104 | 0.051 | 0.015 | 0.008 | 0.063 | 0.020 |
| 88 | 0.039 | 0.008 | 0.054 | 0.022 | 0.054 | 0.160 | 0.016 | 0.042 | 0.021 | 0.164 | 0.040 |
| 89 | 0.437 | 0.548 | 0.342 | 0.567 | 0.448 | 0.270 | 0.562 | 0.155 | 0.280 | 0.480 | 0.530 |
| 90 | 0.041 | 0.015 | 0.177 | 0.015 | 0.018 | 0.191 | 0.026 | 0.104 | 0.036 | 0.289 | 0.026 |
| 91 | 0.082 | 0.158 | 0.155 | 0.145 | 0.166 | 0.222 | 0.382 | 0.003 | 0.139 | 0.327 | 0.195 |
| 92 | 0.768 | 0.801 | 0.852 | 0.801 | 0.788 | 0.625 | 0.797 | 0.678 | 0.346 | 0.684 | 0.266 |
| 93 | 0.010 | 0.006 | 0.009 | 0.010 | 0.009 | 0.007 | 0.006 | 0.012 | 0.004 | 0.050 | 0.002 |
| 94 | 0.187 | 0.543 | 0.483 | 0.000 | 0.158 | 0.050 | 0.373 | 0.292 | 0.000 | 0.182 | 0.020 |
| 95 | 0.039 | 0.238 | 0.185 | 0.047 | 0.076 | 0.243 | 0.259 | 0.306 | 0.054 | 0.205 | 0.282 |
| 96 | 0.087 | 0.002 | 0.014 | 0.001 | 0.092 | 0.000 | 0.035 | 0.001 | 0.014 | 0.214 | 0.089 |
| 97 | 0.108 | 0.231 | 0.139 | 0.073 | 0.139 | 0.241 | 0.354 | 0.216 | 0.115 | 0.335 | 0.016 |
| 98 | 0.526 | 0.488 | 0.376 | 0.550 | 0.547 | 0.295 | 0.645 | 0.045 | 0.384 | 0.509 | 0.486 |
| 99 | 0.112 | 0.182 | 0.257 | 0.152 | 0.221 | 0.184 | 0.210 | 0.200 | 0.053 | 0.255 | 0.137 |
| 100 | 0.609 | 0.745 | 0.645 | 0.675 | 0.628 | 0.518 | 0.783 | 0.251 | 0.130 | 0.618 | 0.703 |
| 101 | 0.709 | 0.636 | 0.210 | 0.744 | 0.713 | 0.195 | 0.748 | 0.030 | 0.603 | 0.607 | 0.664 |
| 102 | 0.000 | 0.000 | 0.003 | 0.001 | 0.000 | 0.000 | 0.000 | 0.000 | 0.000 | 0.011 | 0.001 |
| 103 | 0.006 | 0.007 | 0.195 | 0.003 | 0.006 | 0.256 | 0.007 | 0.002 | 0.006 | 0.290 | 0.002 |
| 104 | 0.767 | 0.736 | 0.702 | 0.933 | 0.749 | 0.454 | 0.915 | 0.041 | 0.334 | 0.591 | 0.787 |
| 105 | 0.522 | 0.511 | 0.329 | 0.574 | 0.506 | 0.170 | 0.787 | 0.012 | 0.326 | 0.506 | 0.631 |
| 106 | 0.518 | 0.531 | 0.270 | 0.518 | 0.514 | 0.197 | 0.582 | 0.200 | 0.409 | 0.478 | 0.557 |
| 107 | 0.667 | 0.618 | 0.469 | 0.659 | 0.666 | 0.140 | 0.717 | 0.005 | 0.456 | 0.462 | 0.598 |
| 108 | 0.072 | 0.061 | 0.063 | 0.068 | 0.094 | 0.023 | 0.068 | 0.114 | 0.012 | 0.115 | 0.081 |
| 109 | 0.662 | 0.613 | 0.109 | 0.647 | 0.677 | 0.101 | 0.707 | 0.000 | 0.509 | 0.457 | 0.625 |
| 110 | 0.772 | 0.688 | 0.917 | 0.805 | 0.784 | 0.538 | 0.869 | 0.098 | 0.358 | 0.745 | 0.795 |
| 111 | 0.934 | 0.966 | 0.949 | 0.933 | 0.921 | 0.523 | 0.954 | 0.000 | 0.698 | 0.845 | 0.658 |
| 112 | 0.002 | 0.002 | 0.008 | 0.006 | 0.003 | 0.055 | 0.002 | 0.054 | 0.000 | 0.021 | 0.000 |
| 113 | 0.235 | 0.245 | 0.075 | 0.264 | 0.249 | 0.088 | 0.735 | 0.163 | 0.056 | 0.436 | 0.385 |
| 114 | 0.032 | 0.036 | 0.009 | 0.027 | 0.029 | 0.050 | 0.022 | 0.163 | 0.029 | 0.095 | 0.076 |
| 115 | 0.539 | 0.503 | 0.533 | 0.615 | 0.536 | 0.421 | 0.691 | 0.523 | 0.061 | 0.520 | 0.413 |
| 116 | 0.472 | 0.565 | 0.612 | 0.456 | 0.473 | 0.469 | 0.600 | 0.479 | 0.208 | 0.270 | 0.001 |
| 117 | 0.613 | 0.458 | 0.494 | 0.624 | 0.607 | 0.190 | 0.638 | 0.610 | 0.310 | 0.458 | 0.398 |
| 118 | 0.787 | 0.768 | 0.676 | 0.818 | 0.769 | 0.753 | 0.885 | 0.174 | 0.499 | 0.821 | 0.739 |
| 119 | 0.222 | 0.620 | 0.361 | 0.014 | 0.272 | 0.393 | 0.585 | 0.295 | 0.231 | 0.439 | 0.669 |
| 120 | 0.054 | 0.007 | 0.002 | 0.000 | 0.077 | 0.002 | 0.000 | 0.000 | 0.001 | 0.016 | 0.046 |
| 121 | 0.018 | 0.044 | 0.044 | 0.001 | 0.006 | 0.120 | 0.042 | 0.001 | 0.009 | 0.079 | 0.046 |
| 122 | 0.007 | 0.005 | 0.005 | 0.001 | 0.010 | 0.040 | 0.036 | 0.010 | 0.002 | 0.062 | 0.001 |
| 123 | 0.088 | 0.491 | 0.280 | 0.215 | 0.043 | 0.106 | 0.056 | 0.224 | 0.019 | 0.216 | 0.002 |
| 124 | 0.314 | 0.354 | 0.171 | 0.352 | 0.314 | 0.096 | 0.391 | 0.000 | 0.173 | 0.277 | 0.351 |
| 125 | 0.784 | 0.723 | 0.175 | 0.773 | 0.784 | 0.107 | 0.657 | 0.023 | 0.093 | 0.170 | 0.007 |
| 126 | 0.284 | 0.568 | 0.140 | 0.374 | 0.479 | 0.180 | 0.539 | 0.001 | 0.094 | 0.264 | 0.002 |
| 127 | 0.136 | 0.128 | 0.030 | 0.148 | 0.136 | 0.056 | 0.525 | 0.059 | 0.297 | 0.265 | 0.296 |
| 128 | 0.664 | 0.538 | 0.521 | 0.685 | 0.618 | 0.473 | 0.283 | 0.434 | 0.042 | 0.306 | 0.105 |
| 129 | 0.280 | 0.154 | 0.022 | 0.294 | 0.071 | 0.022 | 0.440 | 0.260 | 0.424 | 0.222 | 0.307 |
| 130 | 0.010 | 0.033 | 0.081 | 0.070 | 0.015 | 0.019 | 0.063 | 0.108 | 0.001 | 0.062 | 0.018 |
| 131 | 0.294 | 0.523 | 0.095 | 0.430 | 0.151 | 0.044 | 0.449 | 0.242 | 0.510 | 0.300 | 0.813 |

Table 18: The NMI on datasets [67]-[131] (Part-1).

| Dataset Index | S³COMP-C | k-FSC | AutoSC | DEC | IDEC | DSCN | PICA | ConClu | EDESC | DMICC | DIVC | LFSS |
|---|---|---|---|---|---|---|---|---|---|---|---|---|
| 67 | 0.254 | 0.306 | 0.526 | 0.649 | 0.644 | 0.532 | 0.558 | 0.560 | 0.653 | 0.562 | 0.570 | 0.586 |
| 68 | 0.837 | 0.776 | 0.927 | 0.581 | 0.609 | 0.824 | 0.300 | 0.183 | 0.555 | 0.245 | 0.281 | 0.244 |
| 69 | 0.071 | 0.053 | 0.007 | 0.067 | 0.069 | 0.076 | 0.026 | 0.081 | 0.046 | 0.040 | 0.023 | 0.122 |
| 70 | 0.005 | 0.033 | 0.016 | 0.051 | 0.037 | 0.088 | 0.213 | 0.271 | 0.174 | 0.306 | 0.160 | 0.163 |
| 71 | 0.090 | 0.060 | 0.399 | 0.662 | 0.678 | 0.571 | 0.291 | 0.791 | 0.625 | 0.685 | 0.142 | 0.765 |
| 72 | 0.047 | 0.110 | 0.017 | 0.030 | 0.067 | 0.008 | 0.075 | 0.026 | 0.169 | 0.061 | 0.072 | 0.046 |
| 73 | 0.389 | 0.731 | 0.830 | 0.678 | 0.588 | 0.691 | 0.644 | 0.577 | 0.716 | 0.588 | 0.653 | 0.713 |
| 74 | 0.000 | 0.019 | 0.004 | 0.013 | 0.067 | 0.009 | 0.012 | 0.032 | 0.017 | 0.022 | 0.014 | 0.040 |
| 75 | 0.018 | 0.029 | 0.016 | 0.166 | 0.588 | 0.299 | 0.278 | 0.292 | 0.144 | 0.176 | 0.308 | 0.151 |
| 76 | 0.046 | 0.188 | 0.034 | 0.253 | 0.012 | 0.253 | 0.189 | 0.206 | 0.310 | 0.245 | 0.213 | 0.184 |
| 77 | 0.003 | 0.003 | 0.016 | 0.069 | 0.054 | 0.087 | 0.232 | 0.260 | 0.238 | 0.210 | 0.219 | 0.145 |
| 78 | 1.000 | 1.000 | 1.000 | 1.000 | 1.000 | 0.724 | 0.959 | 0.921 | 1.000 | 1.000 | 0.969 | 1.000 |
| 79 | 0.009 | 0.003 | 0.005 | 0.003 | 0.007 | 0.015 | 0.034 | 0.073 | 0.053 | 0.035 | 0.035 | 0.038 |
| 80 | 0.768 | 0.850 | 0.781 | 0.719 | 1.000 | 0.793 | 0.447 | 0.285 | 0.732 | 0.364 | 0.412 | 0.284 |
| 81 | 0.795 | 0.840 | 0.889 | 0.671 | 0.573 | 0.829 | 0.608 | 0.515 | 0.758 | 0.641 | 0.618 | 0.621 |
| 82 | 0.004 | 0.066 | 0.017 | 0.096 | 0.087 | 0.069 | 0.111 | 0.110 | 0.098 | 0.092 | 0.101 | 0.091 |
| 83 | 0.001 | 0.061 | 0.370 | 0.337 | 0.253 | 0.183 | 0.407 | 0.403 | 0.376 | 0.336 | 0.366 | 0.334 |
| 84 | 0.165 | 0.279 | 0.041 | 0.161 | 0.147 | 0.271 | 0.337 | 0.280 | 0.406 | 0.285 | 0.206 | 0.472 |
| 85 | 0.376 | 0.541 | 0.653 | 0.662 | 0.652 | 0.580 | 0.646 | 0.625 | 0.609 | 0.642 | 0.652 | 0.675 |
| 86 | 0.294 | 0.393 | 0.818 | 0.287 | 0.342 | 0.652 | 0.325 | 0.298 | 0.526 | 0.185 | 0.347 | 0.352 |
| 87 | 0.017 | 0.010 | 0.026 | 0.040 | 0.043 | 0.039 | 0.045 | 0.043 | 0.037 | 0.055 | 0.040 | 0.051 |
| 88 | 0.026 | 0.042 | 0.038 | 0.033 | 0.045 | 0.033 | 0.000 | 0.038 | 0.033 | 0.034 | 0.000 | 0.060 |
| 89 | 0.546 | 0.416 | 0.570 | 0.227 | 0.206 | 0.267 | 0.356 | 0.271 | 0.446 | 0.327 | 0.350 | 0.175 |
| 90 | 0.102 | 0.040 | 0.074 | 0.019 | 0.016 | 0.013 | 0.000 | 0.011 | 0.044 | 0.012 | 0.005 | 0.007 |
| 91 | 0.188 | 0.175 | 0.124 | 0.069 | 0.149 | 0.164 | 0.000 | 0.179 | 0.166 | 0.102 | 0.000 | 0.120 |
| 92 | 0.340 | 0.581 | 0.728 | 0.711 | 0.680 | 0.614 | 0.853 | 0.728 | 0.715 | 0.782 | 0.830 | 0.813 |
| 93 | 0.001 | 0.002 | 0.005 | 0.006 | 0.003 | 0.007 | 0.004 | 0.012 | 0.008 | 0.006 | 0.003 | 0.009 |
| 94 | 0.029 | 0.112 | 0.000 | 0.088 | 0.169 | 0.209 | 0.239 | 0.277 | 0.246 | 0.222 | 0.195 | 0.243 |
| 95 | 0.645 | 0.408 | 0.110 | 0.061 | 0.064 | 0.117 | 0.197 | 0.182 | 0.289 | 0.158 | 0.198 | 0.064 |
| 96 | 0.104 | 0.228 | 0.025 | 0.005 | 0.018 | 0.038 | 0.068 | 0.030 | 0.052 | 0.078 | 0.034 | 0.097 |
| 97 | 0.467 | 0.496 | 0.317 | 0.102 | 0.071 | 0.149 | 0.287 | 0.212 | 0.480 | 0.303 | 0.269 | 0.063 |
| 98 | 0.523 | 0.536 | 0.606 | 0.541 | 0.495 | 0.572 | 0.432 | 0.375 | 0.505 | 0.410 | 0.432 | 0.329 |
| 99 | 0.049 | 0.088 | 0.300 | 0.131 | 0.119 | 0.185 | 0.136 | 0.142 | 0.118 | 0.124 | 0.162 |
| 100 | 0.782 | 0.423 | 0.758 | 0.424 | 0.482 | 0.553 | 0.691 | 0.637 | 0.747 | 0.620 | 0.663 | 0.292 |
| 101 | 0.678 | 0.677 | 0.708 | 0.623 | 0.543 | 0.657 | 0.463 | 0.430 | 0.705 | 0.484 | 0.461 | 0.444 |
| 102 | 0.000 | 0.001 | 0.000 | 0.000 | 0.001 | 0.000 | 0.000 | 0.000 | 0.000 | 0.000 | 0.000 | 0.000 |
| 103 | 0.006 | 0.006 | 0.005 | 0.006 | 0.006 | 0.006 | 0.006 | 0.005 | 0.006 | 0.007 | 0.007 | 0.006 |
| 104 | 0.868 | 0.894 | 0.791 | 0.534 | 0.343 | 0.646 | 0.679 | 0.490 | 0.774 | 0.666 | 0.694 | 0.542 |
| 105 | 0.779 | 0.779 | 0.807 | 0.568 | 0.379 | 0.706 | 0.157 | 0.362 | 0.740 | 0.395 | 0.121 | 0.331 |
| 106 | 0.609 | 0.535 | 0.627 | 0.431 | 0.477 | 0.521 | 0.464 | 0.439 | 0.554 | 0.471 | 0.447 | 0.458 |
| 107 | 0.671 | 0.653 | 0.683 | 0.661 | 0.563 | 0.599 | 0.353 | 0.590 | 0.673 | 0.541 | 0.358 | 0.479 |
| 108 | 0.070 | 0.078 | 0.114 | 0.037 | 0.042 | 0.095 | 0.057 | 0.066 | 0.071 | 0.067 | 0.058 | 0.057 |
| 109 | 0.753 | 0.669 | 0.724 | 0.710 | 0.526 | 0.637 | 0.413 | 0.560 | 0.756 | 0.460 | 0.507 | 0.354 |
| 110 | 0.800 | 0.812 | 0.864 | 0.738 | 0.731 | 0.684 | 0.754 | 0.565 | 0.756 | 0.687 | 0.726 | 0.677 |
| 111 | 0.919 | 0.755 | 0.964 | 0.900 | 0.894 | 0.789 | 0.911 | 0.682 | 0.928 | 0.837 | 0.901 | 0.849 |
| 112 | 0.001 | 0.001 | 0.004 | 0.002 | 0.002 | 0.004 | 0.001 | 0.001 | 0.007 | 0.003 | 0.001 | 0.002 |
| 113 | 0.537 | 0.366 | 0.505 | 0.050 | 0.072 | 0.323 | 0.233 | 0.165 | 0.404 | 0.154 | 0.244 | 0.072 |
| 114 | 0.115 | 0.056 | 0.053 | 0.030 | 0.028 | 0.039 | 0.036 | 0.026 | 0.038 | 0.034 | 0.038 | 0.039 |
| 115 | 0.524 | 0.279 | 0.610 | 0.580 | 0.554 | 0.520 | 0.438 | 0.496 | 0.583 | 0.512 | 0.465 | 0.504 |
| 116 | 0.102 | 0.207 | 0.339 | 0.404 | 0.441 | 0.447 | 0.335 | 0.475 | 0.466 | 0.449 | 0.290 | 0.433 |
| 117 | 0.263 | 0.441 | 0.615 | 0.443 | 0.402 | 0.488 | 0.560 | 0.584 | 0.668 | 0.590 | 0.582 | 0.619 |
| 118 | 0.826 | 0.671 | 0.913 | 0.630 | 0.647 | 0.678 | 0.693 | 0.583 | 0.702 | 0.661 | 0.691 | 0.703 |
| 119 | 0.688 | 0.491 | 0.252 | 0.127 | 0.101 | 0.493 | 0.188 | 0.126 | 0.469 | 0.198 | 0.221 | 0.082 |
| 120 | 0.001 | 0.021 | 0.001 | 0.033 | 0.020 | 0.020 | 0.021 | 0.063 | 0.107 | 0.022 | 0.026 | 0.004 |
| 121 | 0.024 | 0.035 | 0.046 | 0.005 | 0.007 | 0.057 | 0.002 | 0.031 | 0.005 | 0.026 | 0.010 | 0.004 |
| 122 | 0.005 | 0.008 | 0.004 | 0.001 | 0.010 | 0.011 | 0.005 | 0.003 | 0.005 | 0.009 | 0.005 | 0.001 |
| 123 | 0.002 | 0.009 | 0.009 | 0.337 | 0.465 | 0.026 | 0.136 | 0.229 | 0.177 | 0.094 | 0.089 | 0.104 |
| 124 | 0.410 | 0.362 | 0.430 | 0.388 | 0.284 | 0.337 | 0.040 | 0.310 | 0.421 | 0.282 | 0.000 | 0.082 |
| 125 | 0.007 | 0.168 | 0.012 | 0.630 | 0.788 | 0.029 | 0.569 | 0.466 | 0.860 | 0.593 | 0.602 | 0.334 |
| 126 | 0.376 | 0.133 | 0.001 | 0.405 | 0.391 | 0.283 | 0.215 | 0.409 | 0.674 | 0.391 | 0.153 | 0.084 |
| 127 | 0.282 | 0.320 | 0.344 | 0.062 | 0.109 | 0.215 | 0.086 | 0.101 | 0.206 | 0.096 | 0.057 | 0.083 |
| 128 | 0.257 | 0.018 | 0.940 | 0.394 | 0.331 | 0.706 | 0.000 | 0.567 | 0.712 | 0.440 | 0.000 | 0.220 |
| 129 | 0.128 | 0.401 | 0.442 | 0.039 | 0.042 | 0.363 | 0.054 | 0.324 | 0.388 | 0.304 | 0.041 | 0.014 |
| 130 | 0.008 | 0.025 | 0.001 | 0.065 | 0.071 | 0.009 | 0.003 | 0.004 | 0.014 | 0.047 | 0.003 | 0.079 |
| 131 | 0.697 | 0.520 | 0.669 | 0.067 | 0.031 | 0.742 | 0.000 | 0.462 | 0.600 | 0.330 | 0.000 | 0.023 |

Table 19: The NMI on datasets [67]-[131] (Part-2).

| Dataset Index | KMeans | AggClu | DBSCAN | BIRCH | GMM | OPTICS | SpeClu | MeanShift | k-PC | Affinity | SSC |
|---|---|---|---|---|---|---|---|---|---|---|---|
| 1 | 0.526 | 0.636 | 0.589 | 0.690 | 0.398 | 0.281 | 0.690 | 0.330 | -0.001 | 0.147 | -0.003 |
| 2 | 0.079 | 0.088 | 0.046 | 0.098 | 0.082 | 0.006 | 0.074 | 0.002 | 0.013 | 0.006 | 0.010 |
| 3 | 0.833 | 0.891 | 0.831 | 0.874 | 0.799 | -0.021 | 0.885 | 0.815 | 0.043 | 0.200 | 0.270 |
| 4 | 0.953 | 0.965 | 0.941 | 0.972 | 0.972 | 0.868 | 0.972 | 0.851 | 0.034 | 0.053 | 0.030 |
| 5 | 0.163 | 0.199 | 0.302 | 0.282 | 0.190 | 0.084 | 0.220 | 0.288 | 0.098 | 0.142 | 0.115 |
| 6 | 0.469 | 0.362 | 0.278 | 0.473 | 0.492 | 0.024 | 0.369 | 0.357 | 0.327 | 0.192 | 0.550 |
| 7 | 0.004 | 0.016 | 0.045 | 0.002 | 0.084 | 0.017 | 0.033 | 0.000 | 0.005 | -0.004 | -0.012 |
| 8 | 0.015 | 0.008 | 0.007 | 0.018 | 0.020 | 0.000 | 0.015 | -0.002 | 0.005 | 0.005 | 0.002 |
| 9 | 0.115 | 0.074 | 0.129 | 0.122 | 0.117 | 0.006 | 0.184 | 0.082 | 0.002 | 0.015 | 0.013 |
| 10 | 0.006 | 0.012 | 0.002 | 0.006 | 0.006 | 0.000 | 0.007 | 0.000 | 0.002 | 0.005 | 0.002 |
| 11 | -0.005 | 0.414 | 0.076 | -0.005 | 0.185 | 0.062 | 0.574 | -0.006 | -0.017 | 0.009 | 0.001 |
| 12 | 0.910 | 0.884 | 0.731 | 0.873 | 0.927 | -0.001 | 0.897 | 0.902 | 0.003 | 0.020 | 0.021 |
| 13 | 0.393 | 0.339 | 0.186 | 0.331 | 0.190 | 0.025 | 0.421 | 0.051 | 0.008 | 0.040 | 0.012 |
| 14 | 0.157 | 0.302 | 0.189 | 0.283 | 0.364 | 0.034 | 0.256 | 0.037 | 0.013 | 0.089 | 0.030 |
| 15 | 0.756 | 0.698 | 0.654 | 0.814 | 0.745 | 0.461 | 0.850 | 0.683 | 0.000 | 0.588 | 0.340 |
| 16 | 0.354 | 0.408 | 0.121 | 0.482 | 0.532 | 0.004 | 0.379 | 0.060 | 0.000 | 0.017 | 0.028 |
| 17 | 0.616 | 0.718 | 0.568 | 0.672 | 0.904 | 0.341 | 0.641 | 0.568 | 0.039 | 0.353 | 0.249 |
| 18 | 0.901 | 0.785 | 0.556 | 0.819 | 0.879 | 0.088 | 0.947 | 0.401 | 0.063 | 0.245 | 0.501 |
| 19 | 0.151 | 0.137 | 0.027 | 0.188 | 0.179 | 0.003 | 0.502 | 0.000 | 0.146 | 0.097 | 0.140 |
| 20 | 0.353 | 0.309 | 0.283 | 0.000 | 0.267 | -0.000 | 0.345 | 0.293 | 0.032 | 0.000 | 0.001 |
| 21 | 0.300 | 0.388 | 0.286 | 0.284 | 0.355 | 0.228 | 0.410 | 0.159 | 0.132 | 0.223 | 0.170 |
| 22 | 0.241 | 0.352 | 0.100 | 0.312 | 0.328 | 0.192 | 0.382 | 0.270 | 0.015 | 0.051 | -0.051 |
| 23 | 0.598 | 0.709 | 0.553 | 0.797 | 0.362 | 0.097 | 0.675 | 0.682 | 0.457 | 0.003 | 0.406 |
| 24 | 0.664 | 0.780 | 0.585 | 0.714 | 0.774 | 0.048 | 0.779 | 0.316 | 0.014 | 0.053 | 0.169 |
| 25 | 0.818 | 0.693 | 0.341 | 0.867 | 0.944 | 0.007 | 0.914 | 0.802 | 0.084 | 0.093 | 0.240 |
| 26 | 0.686 | 0.686 | 0.560 | 0.742 | 0.738 | 0.018 | 0.739 | 0.493 | 0.068 | 0.247 | 0.535 |
| 27 | 0.804 | 0.950 | 0.918 | 0.925 | 0.745 | 0.553 | 0.784 | 0.864 | 0.135 | 0.556 | 0.420 |
| 28 | 0.610 | 0.559 | 0.510 | 0.778 | 0.358 | 0.080 | 0.724 | 0.542 | 0.513 | 0.002 | 0.564 |
| 29 | 0.168 | 0.140 | 0.713 | 0.207 | 0.504 | -0.009 | 0.159 | 0.341 | 0.322 | 0.118 | 0.018 |
| 30 | 0.176 | 0.165 | 0.142 | 0.195 | 0.177 | 0.006 | 0.301 | 0.091 | 0.046 | 0.107 | 0.120 |
| 31 | 0.015 | 0.115 | 0.322 | 0.001 | 0.124 | 0.133 | 0.419 | 0.042 | 0.009 | 0.019 | 0.079 |
| 32 | 0.682 | 0.601 | 0.578 | 0.630 | 0.673 | 0.005 | 0.703 | 0.677 | 0.086 | 0.019 | 0.032 |
| 33 | 0.388 | 0.382 | 0.307 | 0.371 | 0.368 | 0.130 | 0.418 | 0.304 | 0.083 | 0.402 | 0.284 |
| 34 | 0.013 | 0.341 | 0.800 | 0.132 | 0.083 | -0.005 | 0.128 | 0.120 | 0.049 | 0.051 | 0.074 |
| 35 | 0.038 | 0.059 | 0.004 | 0.057 | 0.032 | 0.004 | 0.007 | 0.005 | 0.016 | 0.004 | 0.003 |
| 36 | 0.078 | 0.052 | 0.116 | 0.076 | 0.132 | 0.046 | 0.082 | 0.105 | 0.002 | 0.001 | 0.040 |
| 37 | 0.039 | 0.003 | 0.063 | 0.028 | -0.011 | 0.001 | 0.020 | 0.010 | 0.022 | 0.002 | 0.011 |
| 38 | 0.424 | 0.883 | 0.563 | 0.637 | 0.528 | 0.254 | 0.867 | 0.265 | 0.085 | 0.210 | 0.206 |
| 39 | -0.016 | -0.003 | 0.000 | -0.003 | 0.039 | 0.000 | -0.003 | -0.008 | 0.014 | 0.004 | -0.050 |
| 40 | 0.668 | 0.363 | 0.324 | 0.639 | 0.690 | 0.011 | 0.631 | 0.267 | 0.165 | 0.060 | 0.329 |
| 41 | -0.018 | 0.004 | 0.043 | -0.009 | -0.009 | 0.028 | -0.002 | 0.005 | 0.009 | 0.002 | -0.002 |
| 42 | 0.023 | 0.027 | 0.194 | 0.030 | 0.179 | 0.079 | 0.087 | 0.078 | 0.058 | 0.008 | 0.120 |
| 43 | 0.329 | 0.181 | 0.135 | 0.149 | 0.336 | -0.036 | 0.292 | -0.000 | 0.006 | 0.020 | 0.006 |
| 44 | 0.510 | 0.621 | 0.379 | 0.554 | 0.502 | 0.003 | 0.721 | 0.000 | 0.358 | 0.103 | 0.697 |
| 45 | 0.038 | 0.059 | 0.022 | 0.021 | 0.033 | 0.010 | 0.028 | 0.013 | 0.017 | 0.009 | 0.009 |
| 46 | 0.011 | 0.041 | 0.067 | 0.108 | 0.050 | 0.074 | -0.024 | -0.001 | 0.002 | 0.002 | -0.010 |
| 47 | 0.008 | 0.100 | 0.261 | 0.111 | 0.060 | -0.005 | 0.001 | 0.148 | 0.046 | 0.003 | 0.077 |
| 48 | 0.288 | 0.432 | 0.278 | 0.474 | 0.331 | 0.013 | 0.392 | 0.055 | 0.059 | 0.029 | 0.001 |
| 49 | 0.047 | 0.028 | 0.023 | 0.096 | 0.065 | 0.017 | 0.003 | 0.001 | 0.002 | 0.003 | 0.000 |
| 50 | 0.188 | 0.041 | 0.188 | 0.266 | 0.231 | 0.153 | 0.392 | 0.181 | -0.039 | 0.006 | 0.316 |
| 51 | 0.704 | 0.844 | 0.643 | 0.711 | 0.875 | 0.236 | 0.934 | 0.658 | 0.390 | 0.330 | 0.861 |
| 52 | 0.033 | 0.013 | 0.079 | 0.013 | 0.067 | 0.048 | 0.069 | 0.085 | 0.009 | 0.000 | 0.040 |
| 53 | -0.000 | -0.000 | 0.024 | 0.013 | 0.023 | 0.042 | -0.001 | 0.000 | 0.007 | 0.001 | -0.010 |
| 54 | 0.773 | 0.789 | 0.480 | 0.758 | 0.779 | 0.062 | 0.811 | 0.621 | 0.056 | 0.299 | 0.360 |
| 55 | 0.897 | 0.785 | 0.556 | 0.819 | 0.877 | 0.088 | 0.947 | 0.401 | 0.056 | 0.245 | 0.501 |
| 56 | 0.131 | 0.227 | 0.235 | 0.198 | 0.152 | 0.077 | 0.305 | 0.000 | 0.024 | 0.004 | 0.307 |
| 57 | 0.355 | 0.363 | 0.346 | 0.347 | 0.383 | 0.024 | 0.404 | 0.331 | 0.099 | 0.015 | 0.304 |
| 58 | 0.678 | 0.649 | 0.619 | 0.448 | 0.655 | 0.239 | 0.773 | 0.624 | 0.017 | 0.204 | 0.070 |
| 59 | 0.242 | 0.214 | 0.310 | 0.208 | 0.394 | 0.036 | 0.321 | 0.183 | 0.155 | 0.163 | 0.224 |
| 60 | 0.026 | 0.012 | 0.100 | 0.033 | 0.081 | 0.098 | 0.180 | 0.001 | 0.012 | 0.011 | 0.004 |
| 61 | 0.469 | 0.421 | 0.381 | 0.401 | 0.416 | 0.331 | 0.604 | 0.238 | 0.273 | 0.443 | 0.471 |
| 62 | 0.001 | 0.007 | 0.028 | 0.005 | 0.026 | 0.000 | 0.000 | 0.002 | 0.002 | 0.002 | 0.006 |
| 63 | 0.025 | 0.004 | 0.071 | 0.061 | 0.048 | 0.000 | 0.019 | 0.044 | 0.061 | 0.000 | 0.010 |
| 64 | -0.001 | 0.069 | 0.150 | 0.105 | 0.104 | 0.006 | 0.012 | 0.174 | 0.016 | 0.030 | 0.011 |
| 65 | 0.211 | 0.134 | 0.046 | 0.231 | 0.209 | 0.024 | 0.178 | 0.029 | 0.026 | 0.029 | 0.065 |
| 66 | 0.166 | 0.331 | 0.282 | 0.147 | 0.115 | 0.020 | 0.423 | 0.126 | 0.008 | 0.039 | 0.012 |

Table 20: The ARI on datasets [1]-[66] (Part-1).

| Dataset Index | S³COMP-C | k-FSC | AutoSC | DEC | IDEC | DSCN | PICA | ConClu | EDESC | DMICC | DIVC | LFSS |
|---|---|---|---|---|---|---|---|---|---|---|---|---|
| 1 | -0.038 | -0.009 | 0.441 | 0.723 | 0.713 | 0.237 | 0.446 | 0.530 | 0.445 | 0.521 | 0.361 | 0.690 |
| 2 | 0.019 | 0.030 | 0.038 | 0.108 | 0.087 | 0.041 | 0.070 | 0.123 | 0.093 | 0.104 | 0.038 | 0.079 |
| 3 | 0.026 | 0.165 | -0.012 | 0.879 | 0.504 | 0.730 | 0.882 | 0.897 | 0.901 | 0.883 | 0.885 | 0.880 |
| 4 | 0.019 | 0.051 | 0.965 | 0.962 | 0.958 | 0.727 | 0.875 | 0.969 | 0.842 | 0.900 | 0.716 | 0.903 |
| 5 | 0.089 | 0.205 | 0.232 | 0.188 | 0.189 | 0.198 | 0.192 | 0.259 | 0.180 | 0.213 | 0.212 | 0.239 |
| 6 | 0.349 | 0.442 | 0.530 | 0.480 | 0.504 | 0.215 | 0.519 | 0.385 | 0.422 | 0.459 | 0.508 | 0.468 |
| 7 | 0.028 | 0.012 | 0.045 | 0.001 | 0.009 | 0.032 | -0.003 | 0.003 | 0.003 | 0.002 | -0.005 | 0.018 |
| 8 | 0.003 | 0.004 | 0.001 | 0.013 | 0.010 | 0.014 | 0.018 | 0.026 | 0.023 | 0.026 | 0.019 | 0.024 |
| 9 | 0.001 | 0.036 | 0.008 | 0.151 | 0.175 | 0.026 | 0.106 | 0.160 | 0.090 | 0.141 | 0.107 | 0.124 |
| 10 | 0.002 | 0.003 | 0.002 | 0.003 | 0.027 | 0.012 | 0.023 | 0.034 | 0.011 | 0.036 | 0.020 | 0.037 |
| 11 | -0.018 | 0.101 | -0.016 | 0.010 | -0.002 | -0.003 | 0.000 | 0.194 | 0.565 | 0.179 | 0.000 | 0.040 |
| 12 | 0.016 | 0.221 | 0.940 | 0.900 | 0.586 | 0.903 | 0.918 | 0.940 | 0.951 | 0.872 | 0.915 | 0.936 |
| 13 | 0.011 | 0.059 | 0.167 | 0.201 | 0.207 | 0.241 | 0.253 | 0.421 | 0.260 | 0.337 | 0.289 | 0.252 |
| 14 | 0.022 | 0.039 | 0.215 | 0.127 | 0.211 | 0.209 | 0.186 | 0.103 | 0.155 | 0.167 | 0.144 | 0.222 |
| 15 | 0.291 | 0.527 | 0.602 | 0.688 | 0.696 | 0.477 | 0.749 | 0.791 | 0.638 | 0.746 | 0.744 | 0.827 |
| 16 | 0.022 | 0.157 | 0.055 | 0.141 | 0.586 | 0.419 | 0.351 | 0.518 | 0.380 | 0.368 | 0.387 | 0.537 |
| 17 | 0.052 | 0.175 | 0.584 | 0.789 | 0.703 | 0.318 | 0.592 | 0.662 | 0.595 | 0.683 | 0.600 | 0.701 |
| 18 | 0.193 | 0.564 | 0.884 | 0.765 | 0.754 | 0.550 | 0.806 | 0.867 | 0.801 | 0.719 | 0.802 | 0.712 |
| 19 | 0.082 | 0.154 | 0.165 | 0.087 | 0.030 | 0.079 | 0.132 | 0.086 | 0.172 | 0.093 | 0.139 | 0.176 |
| 20 | 0.018 | 0.014 | 0.000 | 0.336 | 0.355 | 0.214 | 0.326 | 0.374 | 0.331 | 0.364 | 0.328 | 0.371 |
| 21 | 0.177 | 0.198 | 0.392 | 0.285 | 0.265 | 0.173 | 0.397 | 0.376 | 0.403 | 0.424 | 0.396 | 0.443 |
| 22 | -0.048 | 0.103 | 0.198 | 0.157 | 0.192 | 0.197 | 0.086 | 0.227 | 0.193 | 0.185 | 0.140 | 0.069 |
| 23 | 0.007 | 0.482 | 0.425 | 0.259 | 0.615 | 0.104 | 0.072 | 0.250 | 0.372 | 0.366 | 0.132 | 0.126 |
| 24 | -0.006 | 0.045 | 0.672 | 0.570 | 0.615 | 0.752 | 0.641 | 0.690 | 0.736 | 0.690 | 0.567 | 0.562 |
| 25 | 0.161 | 0.434 | 0.640 | 0.594 | 0.567 | 0.655 | 0.904 | 0.913 | 0.678 | 0.784 | 0.891 | 0.772 |
| 26 | 0.374 | 0.638 | 0.567 | 0.575 | 0.562 | 0.564 | 0.571 | 0.612 | 0.626 | 0.647 | 0.534 | 0.614 |
| 27 | 0.142 | 0.533 | 0.423 | 0.566 | 0.676 | 0.371 | 0.694 | 0.639 | 0.838 | 0.761 | 0.669 | 0.549 |
| 28 | 0.036 | 0.475 | 0.353 | 0.244 | 0.595 | 0.149 | 0.062 | 0.177 | 0.395 | 0.395 | 0.078 | 0.187 |
| 29 | -0.041 | 0.591 | -0.033 | 0.647 | 0.613 | 0.057 | 0.201 | 0.229 | 0.345 | 0.192 | 0.215 | 0.288 |
| 30 | 0.119 | 0.163 | 0.077 | 0.172 | 0.285 | 0.170 | 0.144 | 0.153 | 0.168 | 0.175 | 0.153 | 0.157 |
| 31 | 0.008 | 0.078 | 0.242 | 0.073 | 0.086 | 0.029 | 0.287 | 0.352 | 0.176 | 0.260 | 0.242 | 0.267 |
| 32 | 0.010 | 0.043 | 0.668 | 0.400 | 0.285 | 0.582 | 0.618 | 0.670 | 0.689 | 0.639 | 0.644 | 0.677 |
| 33 | 0.277 | 0.302 | 0.412 | 0.382 | 0.391 | 0.321 | 0.446 | 0.433 | 0.336 | 0.407 | 0.440 | 0.450 |
| 34 | 0.008 | 0.075 | 0.013 | 0.094 | 0.043 | 0.128 | 0.345 | 0.334 | 0.204 | 0.161 | 0.150 | 0.576 |
| 35 | 0.004 | -0.001 | 0.005 | 0.031 | 0.045 | 0.017 | 0.009 | 0.048 | 0.043 | 0.033 | 0.030 | 0.041 |
| 36 | -0.008 | 0.003 | 0.050 | 0.018 | 0.051 | 0.029 | 0.016 | 0.053 | 0.039 | 0.094 | 0.024 | 0.062 |
| 37 | 0.025 | 0.013 | 0.001 | -0.000 | 0.004 | 0.018 | 0.035 | 0.045 | 0.034 | 0.032 | 0.009 | 0.030 |
| 38 | 0.110 | 0.343 | 0.904 | 0.468 | 0.430 | 0.330 | 0.521 | 0.553 | 0.689 | 0.547 | 0.555 | 0.594 |
| 39 | 0.000 | -0.025 | -0.023 | -0.003 | -0.003 | -0.011 | 0.002 | 0.043 | 0.050 | 0.019 | 0.007 | -0.021 |
| 40 | 0.198 | 0.245 | 0.299 | 0.481 | 0.423 | 0.581 | 0.459 | 0.507 | 0.479 | 0.519 | 0.420 | 0.557 |
| 41 | 0.000 | -0.001 | -0.009 | -0.005 | -0.016 | -0.005 | -0.002 | 0.023 | 0.013 | -0.001 | 0.006 | 0.038 |
| 42 | 0.084 | 0.143 | 0.164 | 0.150 | 0.128 | 0.013 | 0.017 | 0.193 | 0.059 | 0.206 | 0.010 | 0.208 |
| 43 | 0.001 | 0.028 | -0.014 | 0.303 | 0.278 | 0.271 | 0.245 | 0.382 | 0.335 | 0.327 | 0.269 | 0.275 |
| 44 | 0.833 | 0.752 | 0.671 | 0.403 | 0.317 | 0.145 | 0.537 | 0.278 | 0.573 | 0.414 | 0.470 | 0.454 |
| 45 | 0.003 | -0.001 | 0.018 | 0.020 | 0.022 | 0.003 | 0.024 | 0.036 | 0.028 | 0.025 | 0.010 | 0.028 |
| 46 | -0.004 | -0.046 | 0.088 | 0.106 | 0.231 | 0.036 | 0.022 | 0.044 | -0.020 | 0.049 | 0.030 | 0.022 |
| 47 | -0.000 | 0.027 | 0.004 | 0.019 | 0.153 | -0.012 | 0.041 | 0.020 | 0.056 | 0.094 | 0.088 | 0.058 |
| 48 | 0.003 | 0.020 | 0.474 | 0.339 | 0.235 | 0.288 | 0.522 | 0.567 | 0.539 | 0.452 | 0.539 | 0.534 |
| 49 | 0.003 | 0.000 | 0.001 | 0.058 | 0.072 | 0.040 | 0.041 | 0.042 | 0.037 | 0.054 | 0.053 | 0.124 |
| 50 | 0.212 | 0.152 | 0.006 | 0.072 | 0.175 | 0.103 | 0.078 | 0.053 | 0.135 | 0.178 | 0.057 | 0.119 |
| 51 | 0.785 | 0.824 | 0.921 | 0.778 | 0.747 | 0.677 | 0.753 | 0.670 | 0.830 | 0.768 | 0.621 | 0.701 |
| 52 | 0.001 | 0.000 | -0.006 | 0.039 | 0.071 | 0.028 | -0.000 | -0.005 | 0.001 | 0.055 | 0.003 | 0.013 |
| 53 | 0.119 | 0.014 | 0.068 | 0.032 | 0.187 | 0.067 | 0.043 | 0.015 | 0.024 | -0.001 | 0.017 | 0.038 |
| 54 | 0.090 | 0.285 | 0.720 | 0.646 | 0.592 | 0.465 | 0.762 | 0.681 | 0.678 | 0.721 | 0.756 | 0.813 |
| 55 | 0.191 | 0.564 | 0.884 | 0.723 | 0.787 | 0.646 | 0.778 | 0.917 | 0.819 | 0.727 | 0.791 | 0.848 |
| 56 | 0.013 | 0.159 | 0.131 | 0.111 | 0.187 | 0.166 | 0.117 | 0.198 | 0.247 | 0.146 | 0.125 | 0.141 |
| 57 | 0.001 | 0.225 | 0.000 | 0.479 | 0.372 | 0.358 | 0.542 | 0.488 | 0.400 | 0.528 | 0.467 | 0.543 |
| 58 | -0.016 | 0.062 | 0.649 | 0.714 | 0.724 | 0.495 | 0.360 | 0.624 | 0.666 | 0.687 | 0.392 | 0.649 |
| 59 | 0.249 | 0.159 | 0.252 | 0.239 | 0.224 | 0.230 | 0.252 | 0.287 | 0.289 | 0.276 | 0.242 | 0.282 |
| 60 | 0.009 | 0.004 | 0.007 | 0.035 | 0.372 | 0.033 | 0.032 | 0.025 | 0.040 | 0.036 | 0.033 | 0.037 |
| 61 | 0.258 | 0.475 | 0.412 | 0.396 | 0.423 | 0.324 | 0.488 | 0.362 | 0.429 | 0.440 | 0.482 | 0.478 |
| 62 | 0.002 | 0.017 | 0.002 | 0.003 | 0.008 | 0.018 | 0.002 | 0.003 | 0.008 | 0.005 | 0.008 | 0.019 |
| 63 | 0.005 | 0.015 | 0.063 | 0.008 | 0.043 | 0.042 | 0.006 | -0.025 | -0.007 | 0.003 | 0.009 | 0.007 |
| 64 | 0.002 | 0.000 | -0.002 | 0.075 | 0.086 | 0.066 | 0.043 | 0.059 | 0.078 | 0.055 | 0.008 | 0.021 |
| 65 | 0.044 | 0.083 | 0.148 | 0.144 | 0.164 | 0.119 | 0.124 | 0.099 | 0.146 | 0.130 | 0.124 | 0.127 |
| 66 | 0.012 | 0.018 | 0.059 | 0.145 | 0.155 | 0.145 | 0.131 | 0.271 | 0.222 | 0.190 | 0.138 | 0.102 |

Table 21: The ARI on datasets [1]-[66] (Part-2).

| Dataset Index | KMeans | AggClu | DBSCAN | BIRCH | GMM | OPTICS | SpeClu | MeanShift | k-PC | Affinity | SSC |
|---|---|---|---|---|---|---|---|---|---|---|---|
| 67 | 0.724 | 0.766 | 0.644 | 0.741 | 0.739 | 0.150 | 0.748 | 0.723 | 0.057 | 0.242 | 0.210 |
| 68 | 0.197 | 0.657 | 0.359 | 0.793 | 0.168 | 0.114 | 0.781 | 0.000 | 0.658 | 0.181 | 0.870 |
| 69 | 0.006 | 0.004 | 0.088 | -0.006 | 0.066 | 0.074 | 0.016 | -0.008 | 0.042 | 0.003 | 0.058 |
| 70 | 0.327 | 0.175 | 0.095 | 0.086 | 0.296 | 0.014 | 0.246 | 0.063 | -0.018 | 0.018 | 0.053 |
| 71 | 0.620 | 0.610 | 0.416 | 0.634 | 0.603 | 0.135 | 0.934 | 0.438 | 0.010 | 0.139 | 0.105 |
| 72 | 0.000 | 0.000 | 0.215 | 0.000 | 0.151 | 0.048 | 0.049 | 0.165 | -0.002 | 0.001 | 0.016 |
| 73 | 0.582 | 0.481 | 0.603 | 0.645 | 0.566 | 0.002 | 0.686 | 0.091 | 0.465 | 0.125 | 0.670 |
| 74 | 0.007 | 0.001 | 0.039 | 0.003 | 0.037 | 0.003 | 0.013 | -0.002 | 0.005 | 0.005 | 0.021 |
| 75 | 0.269 | 0.479 | 0.004 | 0.368 | 0.140 | 0.012 | 0.334 | -0.003 | 0.004 | 0.013 | 0.020 |
| 76 | 0.286 | 0.322 | 0.156 | 0.331 | 0.264 | 0.009 | 0.403 | 0.003 | 0.047 | 0.063 | 0.093 |
| 77 | -0.095 | 0.160 | 0.323 | 0.127 | 0.065 | 0.398 | 0.408 | -0.048 | -0.028 | -0.038 | 0.127 |
| 78 | 1.000 | 1.000 | 1.000 | 0.971 | 1.000 | 0.314 | 1.000 | 0.957 | 0.153 | 0.606 | 1.000 |
| 79 | 0.051 | 0.031 | 0.022 | 0.031 | 0.020 | -0.060 | 0.021 | 0.056 | 0.000 | 0.011 | -0.022 |
| 80 | 0.564 | 0.553 | 0.244 | 0.632 | 0.582 | 0.004 | 0.687 | 0.000 | 0.399 | 0.115 | 0.670 |
| 81 | 0.617 | 0.546 | 0.445 | 0.685 | 0.630 | 0.004 | 0.858 | 0.001 | 0.189 | 0.193 | 0.865 |
| 82 | 0.052 | 0.074 | 0.061 | 0.065 | 0.071 | 0.021 | 0.076 | 0.001 | 0.027 | 0.012 | -0.011 |
| 83 | 0.254 | 0.312 | 0.115 | 0.337 | 0.575 | -0.000 | 0.252 | 0.000 | -0.000 | 0.014 | -0.000 |
| 84 | 0.078 | 0.148 | 0.074 | 0.114 | 0.270 | 0.023 | 0.200 | 0.013 | 0.113 | 0.061 | 0.072 |
| 85 | 0.550 | 0.434 | 0.418 | 0.398 | 0.506 | 0.010 | 0.765 | 0.457 | 0.271 | 0.445 | 0.276 |
| 86 | 0.214 | 0.208 | 0.227 | 0.313 | 0.440 | 0.005 | 0.689 | 0.000 | 0.220 | 0.044 | 0.270 |
| 87 | 0.033 | 0.062 | 0.029 | 0.046 | 0.048 | -0.003 | 0.051 | 0.015 | -0.001 | 0.040 | 0.010 |
| 88 | 0.009 | 0.002 | 0.022 | 0.016 | 0.021 | -0.018 | 0.003 | 0.017 | 0.033 | 0.050 | 0.010 |
| 89 | 0.315 | 0.400 | 0.057 | 0.429 | 0.322 | 0.001 | 0.314 | 0.003 | 0.175 | 0.063 | 0.350 |
| 90 | 0.003 | 0.005 | 0.011 | -0.003 | 0.005 | 0.016 | 0.002 | 0.001 | 0.030 | 0.017 | -0.006 |
| 91 | 0.052 | 0.083 | 0.100 | 0.090 | 0.097 | 0.014 | 0.278 | 0.000 | 0.128 | 0.019 | 0.174 |
| 92 | 0.620 | 0.616 | 0.682 | 0.616 | 0.662 | 0.300 | 0.696 | 0.416 | 0.163 | 0.387 | 0.184 |
| 93 | 0.008 | 0.002 | 0.008 | 0.023 | 0.020 | 0.003 | 0.000 | 0.005 | 0.004 | 0.002 | 0.001 |
| 94 | 0.368 | 0.676 | 0.624 | 0.001 | 0.305 | 0.007 | 0.367 | 0.480 | -0.002 | 0.012 | 0.026 |
| 95 | -0.004 | 0.057 | 0.012 | -0.009 | -0.001 | 0.005 | 0.131 | -0.002 | 0.017 | -0.012 | 0.173 |
| 96 | -0.038 | 0.001 | 0.018 | -0.000 | 0.054 | 0.000 | -0.005 | -0.000 | 0.032 | 0.023 | -0.053 |
| 97 | -0.003 | 0.018 | 0.086 | 0.002 | 0.032 | 0.006 | 0.177 | -0.017 | 0.057 | 0.067 | 0.004 |
| 98 | 0.394 | 0.346 | 0.074 | 0.415 | 0.401 | 0.004 | 0.423 | 0.000 | 0.282 | 0.100 | 0.296 |
| 99 | 0.075 | 0.111 | 0.076 | 0.122 | 0.135 | 0.023 | 0.116 | 0.097 | 0.040 | 0.066 | 0.098 |
| 100 | 0.316 | 0.511 | 0.408 | 0.399 | 0.350 | 0.101 | 0.624 | 0.002 | 0.042 | 0.245 | 0.540 |
| 101 | 0.473 | 0.363 | 0.007 | 0.502 | 0.476 | 0.001 | 0.454 | 0.000 | 0.367 | 0.120 | 0.459 |
| 102 | -0.000 | 0.000 | 0.005 | 0.003 | -0.000 | 0.000 | -0.000 | 0.000 | -0.000 | 0.000 | 0.004 |
| 103 | -0.000 | -0.000 | -0.000 | -0.000 | 0.000 | -0.000 | 0.000 | 0.000 | -0.000 | -0.000 | -0.000 |
| 104 | 0.727 | 0.602 | 0.553 | 0.924 | 0.677 | 0.162 | 0.909 | 0.000 | 0.276 | 0.160 | 0.756 |
| 105 | 0.400 | 0.339 | 0.103 | 0.376 | 0.345 | 0.000 | 0.678 | 0.000 | 0.248 | 0.050 | 0.472 |
| 106 | 0.341 | 0.332 | 0.035 | 0.303 | 0.367 | 0.001 | 0.397 | 0.011 | 0.284 | 0.099 | 0.398 |
| 107 | 0.525 | 0.429 | 0.172 | 0.497 | 0.519 | 0.000 | 0.554 | 0.000 | 0.341 | 0.049 | 0.451 |
| 108 | 0.034 | 0.030 | 0.021 | 0.032 | 0.049 | 0.000 | 0.033 | 0.001 | 0.003 | 0.014 | 0.039 |
| 109 | 0.553 | 0.499 | 0.001 | 0.491 | 0.541 | 0.000 | 0.487 | 0.000 | 0.418 | 0.038 | 0.535 |
| 110 | 0.580 | 0.389 | 0.788 | 0.619 | 0.592 | 0.069 | 0.710 | 0.002 | 0.157 | 0.408 | 0.537 |
| 111 | 0.780 | 0.847 | 0.827 | 0.766 | 0.793 | 0.056 | 0.905 | 0.000 | 0.464 | 0.555 | 0.371 |
| 112 | 0.002 | 0.001 | 0.001 | 0.007 | 0.004 | 0.000 | 0.002 | -0.000 | 0.000 | 0.001 | -0.000 |
| 113 | 0.116 | 0.074 | 0.022 | 0.119 | 0.123 | 0.001 | 0.640 | -0.000 | 0.023 | 0.139 | 0.251 |
| 114 | 0.001 | 0.000 | 0.003 | -0.002 | -0.001 | 0.000 | -0.003 | -0.009 | 0.003 | 0.004 | 0.028 |
| 115 | 0.395 | 0.325 | 0.327 | 0.477 | 0.387 | 0.259 | 0.492 | 0.317 | 0.047 | 0.254 | 0.267 |
| 116 | 0.302 | 0.417 | 0.464 | 0.260 | 0.303 | 0.320 | 0.489 | 0.341 | 0.114 | 0.021 | -0.008 |
| 117 | 0.529 | 0.387 | 0.346 | 0.551 | 0.514 | 0.012 | 0.521 | 0.533 | 0.290 | 0.100 | 0.328 |
| 118 | 0.462 | 0.388 | 0.072 | 0.501 | 0.434 | 0.129 | 0.639 | 0.001 | 0.090 | 0.492 | 0.405 |
| 119 | 0.041 | 0.470 | 0.060 | -0.000 | 0.067 | 0.012 | 0.281 | 0.004 | 0.160 | 0.101 | 0.581 |
| 120 | 0.073 | 0.010 | 0.002 | -0.000 | 0.099 | 0.000 | 0.000 | 0.000 | 0.001 | 0.004 | 0.063 |
| 121 | -0.037 | 0.111 | 0.111 | -0.001 | 0.002 | 0.073 | 0.111 | 0.004 | 0.006 | 0.005 | 0.037 |
| 122 | 0.000 | 0.002 | 0.002 | 0.009 | 0.009 | 0.001 | 0.005 | 0.000 | 0.003 | 0.004 | 0.000 |
| 123 | 0.110 | 0.686 | 0.478 | 0.259 | 0.068 | 0.024 | 0.064 | 0.361 | 0.045 | 0.301 | -0.008 |
| 124 | 0.210 | 0.258 | 0.019 | 0.293 | 0.212 | -0.006 | 0.314 | 0.000 | 0.122 | 0.009 | 0.283 |
| 125 | 0.834 | 0.775 | 0.028 | 0.854 | 0.834 | 0.002 | 0.688 | 0.000 | 0.120 | 0.002 | 0.000 |
| 126 | 0.197 | 0.634 | 0.137 | 0.194 | 0.488 | 0.057 | 0.484 | 0.000 | 0.103 | 0.011 | -0.000 |
| 127 | 0.049 | 0.040 | 0.002 | 0.052 | 0.048 | 0.000 | 0.320 | -0.000 | 0.182 | 0.019 | 0.158 |
| 128 | 0.754 | 0.774 | 0.765 | 0.713 | 0.769 | 0.721 | 0.348 | 0.758 | 0.158 | 0.143 | 0.038 |
| 129 | 0.347 | 0.170 | 0.058 | 0.278 | 0.055 | 0.058 | 0.322 | 0.118 | 0.209 | 0.024 | 0.102 |
| 130 | 0.012 | 0.024 | 0.041 | 0.047 | 0.018 | -0.000 | 0.052 | 0.013 | 0.002 | 0.023 | 0.001 |
| 131 | 0.127 | 0.321 | 0.050 | 0.234 | 0.057 | 0.036 | 0.269 | 0.084 | 0.574 | 0.163 | 0.865 |

Table 22: The ARI on datasets [67]-[131] (Part-1).

| Dataset Index | S³COMP-C | k-FSC | AutoSC | DEC | IDEC | DSCN | PICA | ConClu | EDESC | DMICC | DIVC | LFSS |
|---|---|---|---|---|---|---|---|---|---|---|---|---|
| 67 | 0.139 | 0.243 | 0.446 | 0.697 | 0.696 | 0.573 | 0.481 | 0.580 | 0.695 | 0.588 | 0.470 | 0.491 |
| 68 | 0.820 | 0.784 | 0.930 | 0.530 | 0.548 | 0.802 | 0.220 | 0.107 | 0.462 | 0.152 | 0.200 | 0.196 |
| 69 | 0.094 | 0.048 | 0.002 | -0.004 | -0.004 | 0.074 | 0.025 | 0.078 | 0.032 | 0.022 | 0.019 | 0.164 |
| 70 | 0.019 | 0.051 | 0.045 | 0.102 | 0.083 | 0.136 | 0.232 | 0.289 | 0.206 | 0.351 | 0.193 | 0.152 |
| 71 | 0.084 | 0.025 | 0.299 | 0.573 | 0.583 | 0.496 | 0.279 | 0.780 | 0.535 | 0.619 | 0.133 | 0.765 |
| 72 | 0.078 | 0.204 | 0.030 | 0.036 | 0.163 | 0.013 | 0.091 | 0.038 | 0.304 | 0.122 | 0.097 | 0.083 |
| 73 | 0.281 | 0.680 | 0.770 | 0.560 | 0.429 | 0.563 | 0.525 | 0.423 | 0.586 | 0.465 | 0.540 | 0.631 |
| 74 | -0.001 | 0.013 | -0.002 | 0.019 | 0.163 | 0.005 | 0.016 | 0.038 | 0.021 | 0.026 | 0.018 | 0.053 |
| 75 | 0.034 | 0.055 | 0.034 | 0.189 | 0.429 | 0.345 | 0.349 | 0.364 | 0.179 | 0.220 | 0.378 | 0.206 |
| 76 | 0.015 | 0.188 | 0.005 | 0.232 | 0.017 | 0.223 | 0.163 | 0.174 | 0.289 | 0.213 | 0.178 | 0.186 |
| 77 | 0.025 | -0.007 | -0.036 | -0.082 | 0.002 | 0.139 | 0.171 | 0.160 | 0.162 | 0.109 | 0.146 | 0.205 |
| 78 | 1.000 | 1.000 | 1.000 | 1.000 | 1.000 | 0.572 | 0.931 | 0.834 | 1.000 | 1.000 | 0.953 | 1.000 |
| 79 | 0.033 | 0.000 | -0.021 | 0.029 | 0.020 | 0.052 | 0.037 | 0.030 | 0.057 | 0.058 | 0.033 | 0.060 |
| 80 | 0.684 | 0.845 | 0.678 | 0.650 | 1.000 | 0.733 | 0.327 | 0.162 | 0.645 | 0.251 | 0.296 | 0.200 |
| 81 | 0.738 | 0.815 | 0.874 | 0.561 | 0.445 | 0.808 | 0.505 | 0.376 | 0.647 | 0.550 | 0.515 | 0.534 |
| 82 | -0.003 | 0.052 | -0.001 | 0.060 | 0.061 | 0.049 | 0.069 | 0.082 | 0.053 | 0.053 | 0.060 | 0.049 |
| 83 | 0.001 | 0.041 | 0.253 | 0.342 | 0.265 | 0.180 | 0.351 | 0.325 | 0.342 | 0.312 | 0.344 | 0.248 |
| 84 | 0.108 | 0.160 | 0.004 | 0.103 | 0.091 | 0.113 | 0.253 | 0.204 | 0.281 | 0.203 | 0.173 | 0.409 |
| 85 | 0.286 | 0.420 | 0.538 | 0.529 | 0.506 | 0.358 | 0.511 | 0.491 | 0.453 | 0.501 | 0.513 | 0.550 |
| 86 | 0.208 | 0.276 | 0.764 | 0.166 | 0.203 | 0.574 | 0.212 | 0.175 | 0.394 | 0.105 | 0.229 | 0.246 |
| 87 | 0.005 | 0.012 | 0.031 | 0.034 | 0.046 | 0.033 | 0.032 | 0.064 | 0.043 | 0.047 | 0.032 | 0.033 |
| 88 | 0.019 | 0.028 | -0.003 | 0.039 | 0.040 | 0.005 | 0.000 | 0.055 | 0.036 | 0.029 | 0.000 | 0.046 |
| 89 | 0.423 | 0.299 | 0.443 | 0.141 | 0.129 | 0.072 | 0.244 | 0.177 | 0.332 | 0.220 | 0.239 | 0.137 |
| 90 | 0.055 | 0.025 | 0.030 | 0.009 | 0.013 | 0.002 | 0.000 | 0.009 | 0.030 | 0.009 | 0.003 | 0.005 |
| 91 | 0.159 | 0.134 | 0.064 | 0.048 | 0.109 | 0.096 | 0.000 | 0.124 | 0.152 | 0.078 | 0.000 | 0.099 |
| 92 | 0.281 | 0.414 | 0.610 | 0.542 | 0.565 | 0.474 | 0.749 | 0.588 | 0.561 | 0.694 | 0.785 | 0.770 |
| 93 | 0.001 | -0.000 | 0.008 | 0.003 | 0.002 | 0.006 | 0.006 | 0.023 | 0.012 | 0.008 | 0.002 | 0.014 |
| 94 | -0.083 | 0.266 | 0.002 | 0.117 | 0.269 | 0.268 | 0.153 | 0.287 | 0.235 | 0.293 | 0.112 | 0.173 |
| 95 | 0.640 | 0.317 | 0.007 | -0.007 | 0.002 | 0.025 | 0.107 | 0.080 | 0.220 | 0.087 | 0.102 | 0.024 |
| 96 | -0.038 | 0.152 | -0.029 | -0.000 | 0.003 | -0.021 | 0.023 | 0.007 | 0.059 | 0.078 | 0.040 | 0.124 |
| 97 | 0.348 | 0.363 | 0.190 | 0.054 | 0.026 | 0.059 | 0.179 | 0.146 | 0.380 | 0.182 | 0.153 | 0.039 |
| 98 | 0.362 | 0.445 | 0.439 | 0.412 | 0.372 | 0.402 | 0.321 | 0.267 | 0.371 | 0.280 | 0.328 | 0.235 |
| 99 | 0.038 | 0.072 | 0.216 | 0.106 | 0.085 | 0.116 | 0.105 | 0.108 | 0.108 | 0.097 | 0.095 | 0.122 |
| 100 | 0.633 | 0.256 | 0.563 | 0.202 | 0.279 | 0.350 | 0.490 | 0.438 | 0.526 | 0.400 | 0.449 | 0.135 |
| 101 | 0.429 | 0.487 | 0.421 | 0.360 | 0.260 | 0.325 | 0.232 | 0.185 | 0.475 | 0.242 | 0.235 | 0.256 |
| 102 | 0.001 | 0.002 | 0.000 | 0.001 | 0.003 | -0.001 | 0.000 | 0.000 | 0.001 | 0.000 | 0.000 | 0.000 |
| 103 | -0.000 | -0.000 | 0.000 | -0.000 | 0.000 | 0.000 | 0.000 | 0.000 | -0.000 | -0.000 | 0.000 | 0.000 |
| 104 | 0.870 | 0.888 | 0.742 | 0.445 | 0.259 | 0.500 | 0.625 | 0.412 | 0.724 | 0.625 | 0.639 | 0.504 |
| 105 | 0.712 | 0.776 | 0.737 | 0.459 | 0.275 | 0.633 | 0.110 | 0.248 | 0.673 | 0.301 | 0.080 | 0.260 |
| 106 | 0.478 | 0.418 | 0.466 | 0.296 | 0.335 | 0.357 | 0.316 | 0.300 | 0.393 | 0.320 | 0.301 | 0.332 |
| 107 | 0.540 | 0.560 | 0.550 | 0.528 | 0.401 | 0.432 | 0.292 | 0.470 | 0.563 | 0.419 | 0.287 | 0.402 |
| 108 | 0.042 | 0.039 | 0.061 | 0.015 | 0.018 | 0.051 | 0.028 | 0.035 | 0.038 | 0.033 | 0.029 | 0.028 |
| 109 | 0.724 | 0.624 | 0.659 | 0.615 | 0.382 | 0.550 | 0.339 | 0.461 | 0.696 | 0.321 | 0.419 | 0.289 |
| 110 | 0.602 | 0.649 | 0.720 | 0.535 | 0.518 | 0.374 | 0.592 | 0.301 | 0.561 | 0.477 | 0.551 | 0.502 |
| 111 | 0.807 | 0.580 | 0.909 | 0.715 | 0.710 | 0.508 | 0.824 | 0.436 | 0.780 | 0.670 | 0.806 | 0.697 |
| 112 | 0.000 | 0.001 | 0.005 | 0.002 | 0.002 | 0.002 | 0.001 | 0.001 | 0.009 | 0.004 | 0.001 | 0.002 |
| 113 | 0.426 | 0.274 | 0.378 | 0.015 | 0.022 | 0.176 | 0.123 | 0.072 | 0.296 | 0.080 | 0.131 | 0.031 |
| 114 | 0.057 | 0.018 | 0.015 | -0.000 | -0.001 | 0.002 | 0.006 | 0.008 | 0.005 | 0.003 | 0.006 | 0.008 |
| 115 | 0.411 | 0.187 | 0.481 | 0.490 | 0.448 | 0.353 | 0.274 | 0.351 | 0.434 | 0.368 | 0.306 | 0.410 |
| 116 | 0.040 | 0.092 | 0.192 | 0.223 | 0.309 | 0.398 | 0.177 | 0.392 | 0.347 | 0.301 | 0.166 | 0.268 |
| 117 | 0.204 | 0.433 | 0.556 | 0.385 | 0.333 | 0.419 | 0.480 | 0.516 | 0.658 | 0.512 | 0.511 | 0.588 |
| 118 | 0.564 | 0.288 | 0.761 | 0.210 | 0.236 | 0.253 | 0.318 | 0.155 | 0.323 | 0.271 | 0.320 | 0.335 |
| 119 | 0.609 | 0.417 | 0.118 | 0.055 | 0.053 | 0.368 | 0.115 | 0.077 | 0.403 | 0.119 | 0.135 | 0.037 |
| 120 | 0.000 | 0.028 | 0.000 | 0.045 | 0.024 | 0.020 | 0.028 | 0.086 | 0.144 | 0.029 | 0.035 | 0.006 |
| 121 | 0.023 | 0.021 | 0.037 | 0.002 | -0.005 | 0.088 | 0.006 | 0.033 | -0.010 | 0.051 | 0.010 | 0.004 |
| 122 | 0.000 | 0.010 | 0.002 | 0.002 | 0.012 | 0.002 | 0.004 | 0.003 | 0.006 | 0.009 | 0.002 | 0.001 |
| 123 | -0.008 | -0.033 | -0.048 | 0.511 | 0.645 | 0.015 | 0.081 | 0.233 | 0.276 | 0.046 | 0.068 | 0.029 |
| 124 | 0.345 | 0.273 | 0.333 | 0.267 | 0.189 | 0.251 | 0.034 | 0.209 | 0.330 | 0.176 | 0.000 | 0.067 |
| 125 | 0.000 | 0.222 | 0.000 | 0.677 | 0.842 | 0.011 | 0.630 | 0.573 | 0.909 | 0.626 | 0.666 | 0.410 |
| 126 | 0.425 | 0.142 | 0.000 | 0.353 | 0.372 | 0.320 | 0.227 | 0.434 | 0.689 | 0.415 | 0.155 | 0.103 |
| 127 | 0.140 | 0.202 | 0.180 | 0.016 | 0.034 | 0.044 | 0.029 | 0.032 | 0.094 | 0.031 | 0.020 | 0.029 |
| 128 | 0.391 | 0.013 | 0.974 | 0.384 | 0.300 | 0.832 | 0.000 | 0.711 | 0.780 | 0.543 | 0.000 | 0.038 |
| 129 | -0.057 | 0.281 | 0.198 | 0.061 | 0.042 | 0.285 | 0.049 | 0.149 | 0.269 | 0.281 | 0.016 | 0.012 |
| 130 | 0.000 | 0.031 | 0.002 | 0.072 | 0.054 | 0.011 | 0.004 | 0.005 | 0.018 | 0.061 | 0.003 | 0.108 |
| 131 | 0.735 | 0.422 | 0.528 | 0.037 | 0.019 | 0.710 | 0.000 | 0.265 | 0.480 | 0.174 | 0.000 | 0.012 |

Table 23: The ARI on datasets [67]-[131] (Part-2).

## E IMPLEMENTATION TIME COST

The average time cost (over five runs) of each method on each of the 131 datasets is provided from Table 24 to Table 27.

| Dataset Index | KMeans | AggClu | DBSCAN | BIRCH | GMM | OPTICS | SpeClu | MeanShift | k-PC | Affinity | SSC |
|---|---|---|---|---|---|---|---|---|---|---|---|
| 1 | 0.026 | 0.011 | 0.002 | 0.093 | 0.031 | 0.063 | 0.022 | 0.146 | 0.053 | 0.008 | 0.193 |
| 2 | 0.128 | 0.548 | 0.103 | 0.755 | 13.324 | 26.359 | 7.273 | 2.138 | 0.619 | 21.220 | 467.539 |
| 3 | 0.015 | 0.009 | 0.010 | 0.054 | 0.087 | 0.373 | 0.087 | 0.589 | 0.091 | 0.866 | 14.132 |
| 4 | 0.019 | 0.033 | 0.016 | 0.091 | 0.067 | 1.051 | 0.131 | 1.556 | 0.100 | 0.817 | 23.797 |
| 5 | 0.028 | 0.002 | 0.003 | 0.015 | 0.087 | 0.130 | 0.042 | 0.825 | 0.039 | 0.092 | 2.933 |
| 6 | 0.105 | 0.195 | 0.071 | 0.247 | 1.532 | 10.033 | 0.820 | 1.247 | 0.483 | 17.918 | 52.426 |
| 7 | 0.019 | 0.002 | 0.003 | 0.019 | 0.071 | 0.012 | 0.006 | 0.275 | 0.071 | 0.080 | 0.111 |
| 8 | 0.039 | 0.047 | 0.014 | 0.106 | 0.143 | 1.028 | 0.019 | 1.349 | 0.084 | 0.941 | 11.236 |
| 9 | 0.031 | 0.018 | 0.009 | 0.100 | 0.182 | 0.663 | 0.014 | 1.167 | 0.034 | 0.434 | 15.410 |
| 10 | 0.079 | 0.227 | 0.044 | 0.480 | 1.118 | 7.323 | 0.052 | 1.298 | 0.197 | 5.001 | 165.389 |
| 11 | 0.063 | 0.992 | 0.361 | 1.371 | 3.394 | 68.108 | 0.303 | 5.031 | 0.207 | 113.844 | 576.251 |
| 12 | 0.035 | 2.734 | 1.346 | 0.696 | 0.580 | 489.641 | 5.750 | 24.701 | 0.357 | 182.770 | 4341.151 |
| 13 | 0.012 | 0.004 | 0.004 | 0.021 | 0.118 | 0.234 | 0.083 | 0.664 | 0.023 | 0.203 | 0.211 |
| 14 | 0.021 | 0.003 | 0.003 | 0.009 | 0.185 | 0.179 | 0.011 | 0.404 | 0.075 | 0.243 | 2.764 |
| 15 | 0.011 | 0.002 | 0.002 | 0.006 | 0.050 | 0.093 | 0.046 | 0.591 | 0.103 | 0.014 | 1.577 |
| 16 | 0.031 | 0.151 | 0.065 | 0.252 | 1.350 | 3.978 | 1.407 | 2.664 | 0.183 | 11.275 | 60.462 |
| 17 | 0.008 | 0.002 | 0.003 | 0.006 | 0.106 | 0.095 | 0.026 | 0.249 | 0.050 | 0.036 | 0.114 |
| 18 | 0.010 | 0.001 | 0.003 | 0.009 | 0.082 | 0.023 | 0.030 | 0.602 | 0.019 | 0.017 | 1.830 |
| 19 | 0.391 | 3.370 | 1.174 | 3.901 | 36.224 | 642.053 | 2.040 | 13.488 | 3.194 | 398.349 | 4072.630 |
| 20 | 0.011 | 0.011 | 0.010 | 0.036 | 0.031 | 0.084 | 0.141 | 1.135 | 0.022 | 0.244 | 14.013 |
| 21 | 0.016 | 0.001 | 0.002 | 0.006 | 0.061 | 0.075 | 0.060 | 0.480 | 0.120 | 0.007 | 1.086 |
| 22 | 0.010 | 0.001 | 0.002 | 0.009 | 0.123 | 0.052 | 0.074 | 0.089 | 0.015 | 0.014 | 0.111 |
| 23 | 0.053 | 3.759 | 1.376 | 0.378 | 1.325 | 508.937 | 1.716 | 19.430 | 0.515 | 130.535 | 6549.695 |
| 24 | 0.019 | 0.008 | 0.007 | 0.059 | 0.266 | 0.470 | 0.066 | 0.357 | 0.030 | 0.383 | 1.167 |
| 25 | 0.030 | 0.062 | 0.047 | 0.161 | 1.423 | 5.119 | 0.379 | 2.035 | 0.083 | 4.050 | 103.367 |
| 26 | 0.040 | 0.018 | 0.010 | 0.117 | 1.467 | 0.730 | 0.017 | 0.572 | 0.389 | 0.506 | 3.652 |
| 27 | 0.018 | 0.001 | 0.002 | 0.007 | 0.136 | 0.064 | 0.006 | 0.099 | 0.108 | 0.014 | 1.025 |
| 28 | 0.058 | 4.038 | 1.592 | 0.389 | 1.296 | 629.835 | 6.513 | 22.252 | 0.411 | 148.610 | 8128.247 |
| 29 | 0.013 | 0.006 | 0.004 | 0.030 | 0.299 | 0.063 | 0.011 | 0.361 | 0.017 | 0.091 | 0.361 |
| 30 | 0.039 | 0.034 | 0.012 | 0.168 | 1.354 | 1.094 | 0.021 | 0.711 | 0.540 | 1.312 | 45.365 |
| 31 | 0.009 | 0.001 | 0.002 | 0.007 | 0.023 | 0.050 | 0.004 | 0.088 | 0.026 | 0.016 | 1.009 |
| 32 | 0.029 | 0.438 | 0.200 | 0.205 | 0.155 | 23.553 | 0.217 | 3.749 | 0.070 | 21.456 | 509.126 |
| 33 | 0.041 | 0.007 | 0.008 | 0.014 | 0.204 | 0.556 | 0.113 | 0.814 | 0.190 | 0.319 | 1.324 |
| 34 | 0.021 | 0.027 | 0.023 | 0.051 | 0.099 | 0.076 | 0.028 | 1.528 | 0.027 | 2.350 | 32.637 |
| 35 | 0.130 | 1.169 | 0.234 | 0.598 | 3.679 | 10.116 | 7.639 | 8.538 | 0.494 | 59.312 | 1020.823 |
| 36 | 0.052 | 0.283 | 0.104 | 0.279 | 6.197 | 19.390 | 1.818 | 3.917 | 0.154 | 20.445 | 265.634 |
| 37 | 0.021 | 0.020 | 0.014 | 0.114 | 0.657 | 0.079 | 0.349 | 0.735 | 0.076 | 1.558 | 8.716 |
| 38 | 0.013 | 0.001 | 0.003 | 0.018 | 0.147 | 0.105 | 0.034 | 0.463 | 0.031 | 0.017 | 1.569 |
| 39 | 0.013 | 0.018 | 0.006 | 0.064 | 0.209 | 0.014 | 0.011 | 0.460 | 0.027 | 0.570 | 16.019 |
| 40 | 0.107 | 4.453 | 1.105 | 1.702 | 9.063 | 373.295 | 6.126 | 13.241 | 1.532 | 251.236 | 4448.016 |
| 41 | 0.019 | 0.019 | 0.011 | 0.117 | 0.106 | 0.091 | 0.019 | 1.044 | 0.032 | 0.349 | 10.822 |
| 42 | 0.158 | 3.913 | 0.465 | 4.141 | 15.153 | 107.774 | 0.587 | 7.764 | 4.652 | 138.519 | 15475.563 |
| 43 | 0.035 | 0.060 | 0.025 | 0.090 | 0.169 | 0.305 | 0.028 | 1.207 | 0.046 | 2.642 | 23.565 |
| 44 | 0.267 | 0.602 | 0.192 | 1.304 | 10.275 | 39.518 | 1.486 | 3.887 | 0.907 | 36.292 | 1879.024 |
| 45 | 0.052 | 0.066 | 0.017 | 0.178 | 0.624 | 2.055 | 0.035 | 1.815 | 0.140 | 2.786 | 31.266 |
| 46 | 0.043 | 0.111 | 0.023 | 0.229 | 0.177 | 3.982 | 0.370 | 1.489 | 0.107 | 3.583 | 82.994 |
| 47 | 0.108 | 3.878 | 1.488 | 0.605 | 0.620 | 631.343 | 4.877 | 29.290 | 0.190 | 243.122 | 4868.109 |
| 48 | 0.020 | 0.012 | 0.010 | 0.041 | 0.138 | 0.338 | 0.175 | 0.844 | 0.020 | 1.156 | 17.422 |
| 49 | 0.072 | 0.635 | 0.140 | 1.347 | 2.286 | 3.259 | 5.416 | 5.645 | 0.134 | 90.330 | 455.408 |
| 50 | 0.059 | 0.092 | 0.046 | 0.256 | 1.636 | 0.732 | 0.910 | 1.682 | 0.130 | 11.499 | 183.978 |
| 51 | 0.024 | 0.006 | 0.004 | 0.048 | 0.193 | 0.240 | 0.077 | 0.517 | 0.199 | 0.089 | 7.473 |
| 52 | 0.044 | 0.494 | 0.024 | 0.924 | 1.086 | 2.643 | 1.048 | 2.411 | 0.473 | 12.398 | 4015.081 |
| 53 | 0.056 | 1.453 | 1.031 | 4.824 | 1.582 | 629.845 | 7.789 | 9.724 | 0.360 | 330.008 | 3435.475 |
| 54 | 0.019 | 0.003 | 0.003 | 0.026 | 0.048 | 0.157 | 0.102 | 0.442 | 0.023 | 0.032 | 0.189 |
| 55 | 0.017 | 0.001 | 0.003 | 0.016 | 0.072 | 0.023 | 0.041 | 0.484 | 0.020 | 0.017 | 2.068 |
| 56 | 0.160 | 4.172 | 1.033 | 3.834 | 4.818 | 14.912 | 2.017 | 14.128 | 0.948 | 200.819 | 9547.905 |
| 57 | 0.156 | 7.469 | 1.033 | 8.093 | 22.380 | 162.250 | 24.964 | 26.707 | 8.233 | 153.907 | 2498.089 |
| 58 | 0.019 | 0.015 | 0.013 | 0.021 | 0.058 | 1.074 | 0.286 | 0.923 | 0.067 | 0.952 | 19.051 |
| 59 | 0.048 | 0.004 | 0.004 | 0.026 | 0.247 | 0.288 | 0.016 | 1.056 | 0.091 | 0.159 | 0.575 |
| 60 | 0.414 | 4.085 | 1.114 | 1.188 | 12.821 | 192.929 | 2.040 | 29.161 | 1.300 | 377.470 | 4314.663 |
| 61 | 0.029 | 0.002 | 0.003 | 0.026 | 0.086 | 0.163 | 0.056 | 0.152 | 0.139 | 0.088 | 0.189 |
| 62 | 0.059 | 0.096 | 0.047 | 0.258 | 2.997 | 5.806 | 0.051 | 1.355 | 0.127 | 8.352 | 108.407 |
| 63 | 0.070 | 1.947 | 0.556 | 1.915 | 4.774 | 1.073 | 37.385 | 8.884 | 0.472 | 110.597 | 6596.127 |
| 64 | 0.019 | 0.002 | 0.004 | 0.008 | 0.180 | 0.034 | 0.011 | 0.549 | 0.054 | 0.046 | 0.241 |
| 65 | 0.075 | 0.078 | 0.027 | 0.209 | 1.405 | 1.609 | 0.283 | 1.464 | 0.204 | 3.778 | 45.165 |
| 66 | 0.024 | 0.004 | 0.004 | 0.037 | 0.122 | 0.198 | 0.011 | 0.361 | 0.040 | 0.052 | 4.501 |

Table 24: The Time cost (s) on datasets [1]-[66] (Part-1).

| Dataset Index | S³COMP-C | k-FSC | AutoSC | DEC | IDEC | DSCN | PICA | ConClu | EDESC | DMICC | DIVC | LFSS |
|---|---|---|---|---|---|---|---|---|---|---|---|---|
| 1 | 0.367 | 0.127 | 0.000 | 22.177 | 6.709 | 1.942 | 2.601 | 3.154 | 3.755 | 4.611 | 2.888 | 26.842 |
| 2 | 36.022 | 5.766 | 59.031 | 89.233 | 26.961 | 99.336 | 87.344 | 218.419 | 32.250 | 137.717 | 82.611 | 92.551 |
| 3 | 1.705 | 0.485 | 0.000 | 41.495 | 23.769 | 13.412 | 17.754 | 55.481 | 4.979 | 24.462 | 19.133 | 31.945 |
| 4 | 2.886 | 0.593 | 7.640 | 30.110 | 40.996 | 12.272 | 27.595 | 72.871 | 3.362 | 38.400 | 34.201 | 47.565 |
| 5 | 1.056 | 0.425 | 0.000 | 24.931 | 11.952 | 3.257 | 7.890 | 61.508 | 3.256 | 11.328 | 9.709 | 41.427 |
| 6 | 20.139 | 3.327 | 0.000 | 102.749 | 23.769 | 93.360 | 64.472 | 133.714 | 10.885 | 85.429 | 70.231 | 45.329 |
| 7 | 0.442 | 0.801 | 0.000 | 21.179 | 9.735 | 5.961 | 5.664 | 64.553 | 3.680 | 8.305 | 6.547 | 28.484 |
| 8 | 2.749 | 1.491 | 6.932 | 42.129 | 8.580 | 14.923 | 26.042 | 67.615 | 6.456 | 43.076 | 30.541 | 55.918 |
| 9 | 5.147 | 0.691 | 4.822 | 41.764 | 29.081 | 10.924 | 20.481 | 60.115 | 5.248 | 32.969 | 25.205 | 27.873 |
| 10 | 7.105 | 1.951 | 24.944 | 84.919 | 19.259 | 34.877 | 50.715 | 117.728 | 4.706 | 74.650 | 60.013 | 84.665 |
| 11 | 22.985 | 1.187 | 0.000 | 134.521 | 43.532 | 192.212 | 118.124 | 332.505 | 25.502 | 194.839 | 139.713 | 38.302 |
| 12 | 314.263 | 3.658 | 413.826 | 337.317 | 94.745 | 806.534 | 258.272 | 605.154 | 30.459 | 385.828 | 309.376 | 101.702 |
| 13 | 0.780 | 0.237 | 0.000 | 35.343 | 15.665 | 4.280 | 9.906 | 43.955 | 4.378 | 15.872 | 12.437 | 25.574 |
| 14 | 0.660 | 0.111 | 1.234 | 23.900 | 14.207 | 4.329 | 7.535 | 25.687 | 4.888 | 11.643 | 9.067 | 46.283 |
| 15 | 0.784 | 0.586 | 0.761 | 7.266 | 9.517 | 12.739 | 4.974 | 54.773 | 2.735 | 11.669 | 6.140 | 29.135 |
| 16 | 8.294 | 1.568 | 38.162 | 100.509 | 94.745 | 44.916 | 59.900 | 155.041 | 14.982 | 82.549 | 74.572 | 36.754 |
| 17 | 0.610 | 0.029 | 0.858 | 23.720 | 9.355 | 4.077 | 5.882 | 60.768 | 2.390 | 8.119 | 6.078 | 30.096 |
| 18 | 0.864 | 0.800 | 0.000 | 21.057 | 9.980 | 3.302 | 5.023 | 57.742 | 3.473 | 8.165 | 6.772 | 29.656 |
| 19 | 252.828 | 34.940 | 549.459 | 463.212 | 24.071 | 757.877 | 247.269 | 515.042 | 21.284 | 376.377 | 296.739 | 276.373 |
| 20 | 6.466 | 0.314 | 0.000 | 50.620 | 32.563 | 9.043 | 22.610 | 69.564 | 5.681 | 32.532 | 20.699 | 23.841 |
| 21 | 0.696 | 0.280 | 0.000 | 6.432 | 9.513 | 2.662 | 5.015 | 59.240 | 3.901 | 8.070 | 4.620 | 50.895 |
| 22 | 0.387 | 0.432 | 0.000 | 15.939 | 6.995 | 2.581 | 2.701 | 2.351 | 3.327 | 4.914 | 2.681 | 32.057 |
| 23 | 378.348 | 2.457 | 415.643 | 296.164 | 89.515 | 382.810 | 220.766 | 540.077 | 13.213 | 402.254 | 219.317 | 115.389 |
| 24 | 2.957 | 1.041 | 3.328 | 17.986 | 89.515 | 4.047 | 14.434 | 55.397 | 4.607 | 22.623 | 13.303 | 40.694 |
| 25 | 18.368 | 3.496 | 0.000 | 78.955 | 19.649 | 21.672 | 49.787 | 224.092 | 4.478 | 83.604 | 45.233 | 45.520 |
| 26 | 2.525 | 3.498 | 6.963 | 50.358 | 8.012 | 8.353 | 21.388 | 61.567 | 4.096 | 35.031 | 22.293 | 83.714 |
| 27 | 0.565 | 0.782 | 0.000 | 6.404 | 8.800 | 9.388 | 5.215 | 57.857 | 4.297 | 8.359 | 4.816 | 32.826 |
| 28 | 298.023 | 2.604 | 0.000 | 270.228 | 94.296 | 491.816 | 245.479 | 635.500 | 14.070 | 389.778 | 234.521 | 83.165 |
| 29 | 1.034 | 0.415 | 0.000 | 30.202 | 15.586 | 2.540 | 10.100 | 69.230 | 4.210 | 15.393 | 9.929 | 33.310 |
| 30 | 2.859 | 5.318 | 0.000 | 47.941 | 37.609 | 8.590 | 26.645 | 60.323 | 4.501 | 41.011 | 23.950 | 79.303 |
| 31 | 0.256 | 0.475 | 0.763 | 22.379 | 6.210 | 1.331 | 2.820 | 2.193 | 3.366 | 4.642 | 2.483 | 33.572 |
| 32 | 18.540 | 1.902 | 83.622 | 102.193 | 37.609 | 77.150 | 98.984 | 262.225 | 15.462 | 150.039 | 94.117 | 59.418 |
| 33 | 3.531 | 1.329 | 4.629 | 46.033 | 25.168 | 11.536 | 18.237 | 103.280 | 6.349 | 28.630 | 16.239 | 71.626 |
| 34 | 3.923 | 1.235 | 16.729 | 67.063 | 49.040 | 15.287 | 36.580 | 67.344 | 3.900 | 52.769 | 35.032 | 37.689 |
| 35 | 68.764 | 6.513 | 0.000 | 147.177 | 23.459 | 130.153 | 122.127 | 382.424 | 58.164 | 190.428 | 114.081 | 111.224 |
| 36 | 29.145 | 1.680 | 54.172 | 117.570 | 106.458 | 61.441 | 75.989 | 169.893 | 11.213 | 110.043 | 70.680 | 76.580 |
| 37 | 5.992 | 0.680 | 0.000 | 29.730 | 36.596 | 10.077 | 23.994 | 70.594 | 5.635 | 38.082 | 24.343 | 88.350 |
| 38 | 0.821 | 0.139 | 0.000 | 18.561 | 9.041 | 2.066 | 5.177 | 50.414 | 3.542 | 7.104 | 4.906 | 42.715 |
| 39 | 3.561 | 1.081 | 0.000 | 21.140 | 26.752 | 9.334 | 17.142 | 56.475 | 2.884 | 24.420 | 17.841 | 34.056 |
| 40 | 156.119 | 8.283 | 0.000 | 381.182 | 94.384 | 1048.799 | 250.300 | 1124.609 | 13.583 | 376.773 | 236.117 | 185.872 |
| 41 | 5.800 | 1.049 | 6.934 | 47.662 | 36.254 | 11.496 | 24.156 | 71.313 | 5.723 | 42.144 | 23.097 | 61.196 |
| 42 | 43.075 | 13.857 | 166.433 | 276.379 | 45.900 | 162.018 | 146.364 | 296.794 | 49.686 | 190.255 | 140.324 | 115.899 |
| 43 | 9.093 | 1.608 | 14.692 | 71.720 | 51.802 | 13.662 | 34.494 | 62.442 | 8.578 | 52.410 | 33.305 | 37.213 |
| 44 | 48.188 | 7.073 | 95.101 | 184.010 | 30.489 | 175.656 | 97.735 | 528.705 | 8.534 | 150.237 | 92.428 | 161.115 |
| 45 | 9.691 | 5.703 | 0.000 | 70.665 | 53.472 | 20.003 | 36.389 | 71.834 | 7.975 | 56.496 | 34.973 | 82.507 |
| 46 | 12.953 | 1.875 | 0.000 | 67.109 | 64.878 | 21.156 | 43.289 | 96.027 | 11.048 | 70.985 | 44.937 | 64.303 |
| 47 | 246.482 | 9.542 | 0.000 | 333.316 | 93.768 | 550.350 | 241.384 | 497.397 | 14.562 | 414.421 | 230.219 | 137.642 |
| 48 | 2.532 | 0.525 | 5.584 | 25.833 | 33.251 | 4.528 | 22.315 | 55.214 | 5.472 | 36.955 | 21.299 | 45.701 |
| 49 | 57.488 | 2.119 | 110.257 | 138.928 | 43.087 | 100.902 | 110.162 | 304.012 | 31.522 | 170.259 | 108.025 | 69.671 |
| 50 | 46.074 | 1.454 | 28.504 | 89.857 | 23.426 | 36.768 | 52.186 | 94.776 | 14.459 | 81.031 | 52.458 | 51.438 |
| 51 | 6.104 | 0.492 | 1.637 | 29.084 | 15.772 | 4.936 | 9.682 | 60.871 | 3.520 | 15.938 | 9.780 | 46.302 |
| 52 | 20.475 | 2.247 | 0.000 | 57.585 | 18.067 | 16.974 | 39.055 | 100.286 | 4.264 | 63.467 | 37.927 | 31.079 |
| 53 | 250.064 | 3.888 | 0.000 | 319.244 | 77.042 | 720.102 | 237.725 | 610.427 | 14.823 | 346.115 | 238.762 | 77.500 |
| 54 | 0.538 | 0.135 | 0.000 | 22.498 | 11.471 | 2.757 | 7.312 | 60.383 | 2.954 | 11.880 | 8.125 | 25.588 |
| 55 | 3.547 | 0.785 | 0.000 | 22.228 | 9.299 | 2.927 | 5.006 | 51.633 | 2.957 | 7.203 | 4.903 | 28.464 |
| 56 | 244.532 | 8.117 | 487.303 | 315.048 | 77.042 | 873.339 | 239.411 | 553.498 | 48.430 | 387.511 | 242.214 | 192.701 |
| 57 | 414.237 | 8.693 | 0.000 | 309.447 | 86.277 | 713.473 | 223.035 | 1230.547 | 40.935 | 387.723 | 214.466 | 146.421 |
| 58 | 6.079 | 0.313 | 0.000 | 53.465 | 43.012 | 86.733 | 26.319 | 71.149 | 5.123 | 35.675 | 24.942 | 19.034 |
| 59 | 1.943 | 1.898 | 2.255 | 30.853 | 18.387 | 9.865 | 11.953 | 71.530 | 3.320 | 23.783 | 11.579 | 36.301 |
| 60 | 245.363 | 32.146 | 0.000 | 315.097 | 86.277 | 937.083 | 239.262 | 554.598 | 76.899 | 399.541 | 251.076 | 174.896 |
| 61 | 1.021 | 1.336 | 0.000 | 20.918 | 10.898 | 8.410 | 7.250 | 59.745 | 3.684 | 19.329 | 7.642 | 39.308 |
| 62 | 16.771 | 0.984 | 28.194 | 67.857 | 20.951 | 16.552 | 50.003 | 129.781 | 12.907 | 67.051 | 52.764 | 75.960 |
| 63 | 96.237 | 5.350 | 219.182 | 227.453 | 55.614 | 150.865 | 139.646 | 602.670 | 9.629 | 190.008 | 144.206 | 112.641 |
| 64 | 1.407 | 0.167 | 1.252 | 29.973 | 15.476 | 2.431 | 9.515 | 65.248 | 4.363 | 14.205 | 9.711 | 28.591 |
| 65 | 11.367 | 3.010 | 0.000 | 61.824 | 54.907 | 13.548 | 35.190 | 213.363 | 10.594 | 57.471 | 36.235 | 81.221 |
| 66 | 1.384 | 1.904 | 1.682 | 25.348 | 11.797 | 2.778 | 7.418 | 65.315 | 4.522 | 10.630 | 7.353 | 44.522 |

Table 25: The Time cost (s) on datasets [1]-[66] (Part-2).

| Dataset Index | KMeans | AggClu | DBSCAN | BIRCH | GMM | OPTICS | SpeClu | MeanShift | k-PC | Affinity | SSC |
|---|---|---|---|---|---|---|---|---|---|---|---|
| 67 | 0.028 | 0.004 | 0.003 | 0.012 | 0.055 | 0.178 | 0.048 | 0.604 | 0.040 | 0.061 | 4.168 |
| 68 | 0.031 | 0.005 | 0.004 | 0.042 | 0.115 | 0.210 | 0.087 | 0.216 | 0.072 | 0.054 | 6.431 |
| 69 | 0.008 | 0.001 | 0.002 | 0.013 | 0.069 | 0.013 | 0.005 | 0.171 | 0.051 | 0.009 | 0.143 |
| 70 | 0.015 | 0.006 | 0.004 | 0.048 | 0.077 | 0.285 | 0.127 | 0.276 | 0.034 | 0.104 | 9.370 |
| 71 | 0.023 | 0.066 | 0.045 | 0.058 | 0.140 | 6.204 | 0.462 | 1.971 | 0.196 | 7.016 | 87.508 |
| 72 | 0.118 | 9.083 | 1.543 | 5.370 | 1.668 | 40.818 | 71.671 | 25.334 | 1.250 | 558.461 | 8152.290 |
| 73 | 0.181 | 2.352 | 0.733 | 2.065 | 12.347 | 272.430 | 7.530 | 9.487 | 0.942 | 110.751 | 2486.083 |
| 74 | 0.021 | 0.014 | 0.020 | 0.084 | 1.065 | 0.112 | 0.344 | 0.704 | 0.029 | 2.562 | 6.611 |
| 75 | 0.041 | 0.191 | 0.051 | 0.238 | 0.553 | 8.369 | 0.058 | 1.951 | 0.132 | 4.011 | 35.074 |
| 76 | 0.037 | 0.008 | 0.004 | 0.048 | 0.187 | 0.292 | 0.015 | 0.306 | 0.122 | 0.283 | 13.764 |
| 77 | 0.021 | 0.002 | 0.003 | 0.021 | 0.158 | 0.121 | 0.045 | 0.130 | 0.018 | 0.016 | 2.652 |
| 78 | 0.032 | 0.011 | 0.005 | 0.066 | 0.065 | 0.342 | 0.084 | 0.322 | 1.038 | 0.109 | 74.916 |
| 79 | 0.023 | 0.013 | 0.011 | 0.042 | 0.086 | 0.162 | 0.014 | 0.563 | 0.116 | 1.299 | 12.491 |
| 80 | 0.105 | 0.150 | 0.038 | 0.472 | 5.379 | 5.849 | 0.337 | 1.522 | 0.487 | 8.760 | 519.821 |
| 81 | 0.161 | 0.404 | 0.041 | 0.972 | 2.611 | 5.834 | 0.341 | 2.910 | 5.263 | 6.941 | 2651.364 |
| 82 | 0.330 | 1.120 | 0.445 | 0.632 | 4.942 | 18.030 | 0.500 | 4.069 | 0.402 | 60.053 | 1662.161 |
| 83 | 0.109 | 1.459 | 0.300 | 2.019 | 2.942 | 83.585 | 11.759 | 4.479 | 0.204 | 79.862 | 485.065 |
| 84 | 0.242 | 7.517 | 1.117 | 2.070 | 37.574 | 532.763 | 14.782 | 27.039 | 6.357 | 310.988 | 3110.478 |
| 85 | 0.059 | 0.063 | 0.040 | 0.030 | 0.709 | 6.092 | 0.052 | 2.050 | 0.143 | 5.695 | 31.054 |
| 86 | 0.360 | 3.487 | 1.416 | 2.278 | 7.283 | 628.230 | 5.224 | 35.146 | 1.501 | 139.177 | 8020.986 |
| 87 | 0.031 | 0.024 | 0.014 | 0.018 | 0.117 | 0.420 | 0.021 | 1.061 | 0.096 | 1.553 | 10.484 |
| 88 | 0.126 | 2.078 | 0.588 | 0.951 | 5.378 | 56.638 | 9.741 | 8.353 | 0.518 | 158.136 | 1974.683 |
| 89 | 0.592 | 3.296 | 0.139 | 4.376 | 25.399 | 30.065 | 2.152 | 18.995 | 2.540 | 47.409 | 185.288 |
| 90 | 0.772 | 23.649 | 0.753 | 23.746 | 45.163 | 364.396 | 40.511 | 89.081 | 6.750 | 176.260 | 1998.194 |
| 91 | 5.069 | 81.334 | 2.067 | 72.643 | 106.137 | 641.858 | 10.599 | 51.214 | 12.349 | 394.553 | 2569.169 |
| 92 | 0.032 | 0.012 | 0.007 | 0.109 | 0.146 | 0.602 | 0.014 | 0.420 | 0.084 | 0.295 | 1.089 |
| 93 | 0.065 | 0.098 | 0.033 | 0.310 | 0.567 | 3.528 | 0.043 | 1.724 | 0.156 | 5.214 | 60.541 |
| 94 | 0.253 | 4.509 | 1.451 | 0.345 | 0.792 | 19.332 | 17.925 | 31.434 | 0.182 | 138.731 | 5811.314 |
| 95 | 0.534 | 1.511 | 0.011 | 3.203 | 1296.898 | 0.602 | 0.361 | 13.580 | 0.351 | 4.643 | 6.240 |
| 96 | 0.029 | 0.091 | 0.059 | 0.359 | 0.812 | 0.094 | 0.046 | 1.544 | 0.078 | 12.257 | 33.344 |
| 97 | 0.689 | 3.179 | 0.046 | 5.311 | 87.149 | 8.339 | 3.794 | 24.544 | 1.339 | 9.371 | 82.107 |
| 98 | 0.065 | 0.192 | 0.045 | 0.367 | 3.058 | 5.788 | 0.526 | 1.445 | 0.469 | 11.748 | 36.287 |
| 99 | 0.040 | 0.037 | 0.012 | 0.079 | 0.404 | 0.825 | 0.156 | 0.594 | 0.068 | 0.603 | 3.040 |
| 100 | 0.156 | 0.053 | 0.005 | 0.480 | 13.217 | 0.372 | 0.016 | 0.881 | 0.308 | 0.171 | 1.122 |
| 101 | 1.569 | 16.059 | 0.653 | 21.128 | 18.467 | 305.332 | 13.609 | 145.011 | 7.354 | 64.906 | 1142.617 |
| 102 | 0.099 | 1.213 | 1.151 | 4.895 | 0.673 | 6.995 | 2.751 | 25.970 | 0.395 | 212.739 | 2812.276 |
| 103 | 0.264 | 1.453 | 1.181 | 0.167 | 1.049 | 630.629 | 222.240 | 14.262 | 2.894 | 403.997 | 324.547 |
| 104 | 0.052 | 0.060 | 0.014 | 0.179 | 1.529 | 1.446 | 0.162 | 0.606 | 0.150 | 2.744 | 5.417 |
| 105 | 1.012 | 21.396 | 1.135 | 24.394 | 14.672 | 646.262 | 8.691 | 140.585 | 9.857 | 208.848 | 2768.614 |
| 106 | 0.397 | 2.543 | 0.112 | 3.672 | 6.386 | 20.028 | 0.806 | 41.887 | 2.496 | 18.336 | 169.391 |
| 107 | 0.682 | 20.226 | 1.154 | 23.673 | 63.514 | 635.603 | 3.549 | 254.321 | 8.872 | 108.639 | 2788.326 |
| 108 | 0.981 | 3.449 | 0.143 | 4.132 | 9.815 | 25.075 | 0.161 | 44.930 | 2.386 | 21.782 | 171.150 |
| 109 | 0.691 | 21.329 | 1.045 | 21.789 | 12.803 | 640.593 | 18.242 | 92.121 | 8.329 | 151.105 | 2724.846 |
| 110 | 0.157 | 0.263 | 0.023 | 0.566 | 4.865 | 2.096 | 0.715 | 3.189 | 19.503 | 4.450 | 14.278 |
| 111 | 0.202 | 0.312 | 0.023 | 0.717 | 6.508 | 2.414 | 0.462 | 2.523 | 2.652 | 1.723 | 14.680 |
| 112 | 1.308 | 11.284 | 0.054 | 10.713 | 15.048 | 9.399 | 2.412 | 10.257 | 1.295 | 34.247 | 41.574 |
| 113 | 0.506 | 0.417 | 0.013 | 0.810 | 39.716 | 1.024 | 0.021 | 4.692 | 0.727 | 2.360 | 4.368 |
| 114 | 0.470 | 0.203 | 0.008 | 0.378 | 15.078 | 0.617 | 0.015 | 3.810 | 0.358 | 1.186 | 3.257 |
| 115 | 0.965 | 25.350 | 0.010 | 24.479 | 20.450 | 0.634 | 23.234 | 1.048 | 0.407 | 1.010 | 1.938 |
| 116 | 0.286 | 8.016 | 1.256 | 9.716 | 29.924 | 451.806 | 1.556 | 39.075 | 8.301 | 97.094 | 1660.810 |
| 117 | 0.219 | 1.903 | 0.627 | 0.735 | 1.365 | 178.842 | 2.263 | 8.475 | 0.743 | 76.081 | 2286.553 |
| 118 | 0.786 | 0.200 | 0.007 | 0.423 | 1045.199 | 0.306 | 0.025 | 4.323 | 0.879 | 0.153 | 0.873 |
| 119 | 0.097 | 0.258 | 0.013 | 1.161 | 11.451 | 1.217 | 0.519 | 2.302 | 0.801 | 1.711 | 12.991 |
| 120 | 0.164 | 2.051 | 0.131 | 2.990 | 6.983 | 25.113 | 0.147 | 8.872 | 1.827 | 33.556 | 91.778 |
| 121 | 0.034 | 0.382 | 0.158 | 0.939 | 3.847 | 6.790 | 1.695 | 2.893 | 0.432 | 25.949 | 389.976 |
| 122 | 0.021 | 0.490 | 0.076 | 1.073 | 0.980 | 4.017 | 4.770 | 2.487 | 0.523 | 31.285 | 136.807 |
| 123 | 0.026 | 0.115 | 0.013 | 0.715 | 1.182 | 0.872 | 0.093 | 0.999 | 0.164 | 2.451 | 9.655 |
| 124 | 5.078 | 171.064 | 1.129 | 194.480 | 1024.369 | 635.580 | 164.548 | 46.693 | 11.838 | 538.897 | 2991.328 |
| 125 | 3.320 | 178.808 | 2.934 | 111.498 | 292.325 | 646.709 | 2.232 | 61.516 | 6.871 | 427.106 | 3716.442 |
| 126 | 1.617 | 73.614 | 0.545 | 84.873 | 195.329 | 184.604 | 16.516 | 25.666 | 7.535 | 179.785 | 1625.346 |
| 127 | 9.698 | 172.163 | 1.175 | 210.256 | 1755.463 | 630.454 | 2.156 | 79.402 | 11.280 | 670.174 | 3392.056 |
| 128 | 3.099 | 176.738 | 3.325 | 165.215 | 763.302 | 319.293 | 115.985 | 108.861 | 10.660 | 531.712 | 1896.369 |
| 129 | 163.057 | 1190.028 | 14.069 | 965.166 | 2059.787 | 567.507 | 2.502 | 119.592 | 6.783 | 2573.199 | 6479.839 |
| 130 | 6.545 | 186.351 | 2.205 | 198.558 | 6362.279 | 46.631 | 6.490 | 44.564 | 2.619 | 425.023 | 194.256 |
| 131 | 78.143 | 600.573 | 0.988 | 574.892 | 18787.872 | 346.752 | 68.886 | 65.962 | 6.863 | 111.995 | 1714.213 |

Table 26: The Time cost (s) on datasets [67]-[131] (Part-1).

| Dataset Index | S³COMP-C | k-FSC | AutoSC | DEC | IDEC | DSCN | PICA | ConClu | EDESC | DMICC | DIVC | LFSS |
|---|---|---|---|---|---|---|---|---|---|---|---|---|
| 67 | 6.095 | 2.390 | 1.713 | 13.033 | 16.182 | 4.704 | 9.612 | 65.390 | 3.236 | 15.737 | 9.817 | 52.855 |
| 68 | 0.899 | 0.754 | 1.455 | 31.460 | 15.729 | 2.586 | 9.578 | 67.219 | 4.896 | 14.164 | 9.761 | 69.202 |
| 69 | 0.736 | 0.135 | 0.691 | 21.942 | 8.547 | 2.540 | 4.916 | 60.014 | 2.418 | 7.063 | 4.992 | 50.620 |
| 70 | 1.130 | 0.802 | 1.598 | 28.911 | 13.961 | 2.354 | 9.740 | 62.549 | 4.583 | 18.149 | 9.764 | 53.517 |
| 71 | 49.596 | 1.044 | 28.739 | 80.754 | 74.034 | 48.738 | 51.575 | 279.547 | 9.723 | 78.473 | 52.057 | 62.405 |
| 72 | 224.471 | 5.314 | 0.000 | 351.963 | 94.820 | 571.097 | 238.337 | 551.744 | 16.219 | 377.805 | 234.808 | 77.654 |
| 73 | 150.134 | 17.109 | 289.949 | 308.578 | 70.758 | 478.766 | 185.164 | 366.898 | 11.930 | 289.247 | 178.357 | 247.280 |
| 74 | 7.625 | 0.581 | 0.000 | 36.537 | 94.820 | 9.937 | 28.712 | 76.357 | 3.415 | 39.375 | 28.792 | 41.870 |
| 75 | 8.369 | 1.963 | 0.000 | 87.545 | 70.758 | 33.450 | 56.308 | 138.544 | 14.410 | 92.362 | 54.193 | 92.363 |
| 76 | 1.032 | 1.675 | 0.000 | 29.006 | 11.212 | 2.690 | 10.121 | 65.098 | 4.648 | 19.919 | 9.854 | 61.823 |
| 77 | 0.502 | 0.816 | 0.000 | 8.813 | 8.323 | 1.949 | 5.321 | 61.787 | 3.817 | 6.713 | 4.693 | 30.904 |
| 78 | 5.894 | 0.892 | 2.538 | 13.212 | 5.030 | 9.305 | 10.188 | 58.736 | 3.280 | 17.046 | 9.375 | 29.884 |
| 79 | 3.933 | 0.231 | 0.000 | 22.816 | 30.433 | 6.517 | 19.041 | 67.998 | 4.828 | 26.860 | 18.149 | 31.899 |
| 80 | 17.391 | 5.578 | 0.000 | 82.856 | 5.030 | 30.769 | 49.501 | 282.547 | 5.778 | 88.101 | 45.574 | 156.487 |
| 81 | 19.567 | 9.606 | 28.008 | 62.262 | 19.161 | 30.817 | 49.912 | 123.368 | 5.885 | 86.719 | 46.486 | 84.899 |
| 82 | 85.051 | 5.793 | 0.000 | 176.123 | 51.030 | 233.320 | 133.423 | 510.666 | 41.041 | 205.304 | 125.380 | 79.874 |
| 83 | 66.167 | 3.649 | 130.225 | 156.066 | 39.534 | 112.088 | 119.628 | 279.523 | 22.464 | 167.113 | 122.487 | 108.287 |
| 84 | 268.529 | 17.316 | 0.000 | 314.355 | 77.356 | 702.740 | 247.208 | 555.431 | 15.177 | 372.923 | 237.461 | 181.722 |
| 85 | 17.838 | 4.603 | 0.000 | 77.940 | 15.875 | 82.301 | 47.523 | 279.895 | 5.934 | 67.723 | 46.105 | 95.161 |
| 86 | 444.890 | 28.734 | 482.052 | 434.524 | 93.463 | 497.616 | 239.080 | 603.763 | 15.082 | 374.421 | 231.593 | 295.440 |
| 87 | 3.246 | 2.297 | 0.000 | 50.579 | 11.630 | 9.322 | 26.064 | 65.514 | 5.957 | 42.024 | 24.612 | 54.184 |
| 88 | 145.100 | 12.562 | 0.000 | 227.377 | 46.821 | 398.489 | 147.066 | 636.304 | 10.888 | 238.700 | 131.656 | 99.241 |
| 89 | 94.929 | 25.558 | 84.225 | 163.467 | 28.974 | 57.450 | 85.803 | 238.476 | 8.557 | 142.289 | 76.593 | 93.839 |
| 90 | 747.463 | 22.577 | 384.139 | 366.632 | 79.425 | 757.833 | 201.796 | 415.066 | 13.671 | 283.880 | 188.653 | 429.988 |
| 91 | 920.422 | 23.579 | 0.000 | 386.530 | 101.593 | 442.883 | 250.263 | 499.519 | 111.128 | 395.012 | 229.945 | 338.776 |
| 92 | 4.160 | 2.436 | 3.988 | 18.847 | 6.926 | 4.663 | 14.544 | 61.723 | 3.658 | 23.104 | 13.685 | 79.602 |
| 93 | 17.812 | 3.311 | 0.000 | 75.340 | 18.289 | 9.597 | 43.068 | 123.335 | 11.872 | 66.684 | 38.473 | 48.807 |
| 94 | 791.076 | 1.463 | 475.280 | 268.404 | 89.843 | 472.955 | 246.187 | 541.834 | 14.380 | 338.915 | 220.275 | 85.856 |
| 95 | 9.191 | 7.152 | 5.493 | 48.622 | 11.462 | 5.702 | 19.809 | 14.124 | 5.190 | 35.372 | 18.530 | 114.968 |
| 96 | 7.001 | 0.882 | 0.000 | 52.852 | 18.620 | 13.676 | 47.371 | 124.314 | 4.502 | 76.607 | 44.228 | 32.113 |
| 97 | 10.543 | 16.406 | 0.000 | 94.291 | 24.658 | 30.290 | 55.091 | 139.147 | 13.386 | 104.469 | 51.316 | 107.498 |
| 98 | 21.763 | 3.474 | 0.000 | 84.626 | 18.851 | 58.076 | 49.193 | 281.117 | 5.891 | 79.624 | 44.392 | 233.757 |
| 99 | 2.439 | 0.767 | 5.233 | 23.975 | 7.954 | 3.565 | 21.032 | 93.231 | 5.988 | 39.990 | 19.771 | 42.061 |
| 100 | 6.281 | 8.071 | 2.853 | 12.360 | 5.320 | 3.184 | 10.187 | 63.810 | 5.476 | 16.539 | 9.287 | 60.190 |
| 101 | 324.162 | 76.093 | 0.000 | 221.483 | 64.253 | 1223.510 | 196.766 | 364.010 | 17.696 | 311.561 | 174.968 | 500.888 |
| 102 | 204.229 | 12.238 | 470.114 | 324.943 | 90.306 | 285.491 | 235.982 | 554.547 | 72.227 | 328.798 | 217.935 | 93.160 |
| 103 | 274.903 | 97.995 | 0.000 | 442.398 | 76.917 | 809.316 | 236.566 | 570.124 | 18.912 | 331.701 | 213.124 | 120.934 |
| 104 | 7.664 | 1.130 | 0.000 | 55.324 | 13.060 | 18.371 | 27.112 | 23.036 | 4.573 | 40.718 | 23.951 | 98.018 |
| 105 | 437.735 | 50.094 | 0.000 | 278.293 | 95.029 | 464.973 | 251.182 | 570.963 | 96.058 | 388.483 | 224.819 | 195.596 |
| 106 | 17.267 | 22.845 | 67.111 | 132.796 | 30.504 | 92.989 | 75.804 | 119.075 | 23.859 | 117.212 | 68.058 | 156.577 |
| 107 | 404.795 | 40.005 | 0.000 | 384.618 | 79.122 | 426.696 | 249.461 | 573.656 | 17.409 | 375.267 | 221.887 | 165.412 |
| 108 | 155.180 | 23.982 | 0.000 | 94.491 | 34.702 | 106.886 | 83.962 | 450.711 | 8.363 | 142.143 | 73.475 | 228.707 |
| 109 | 410.232 | 28.684 | 428.845 | 276.364 | 93.984 | 550.955 | 251.640 | 581.243 | 87.851 | 382.977 | 223.125 | 399.949 |
| 110 | 7.121 | 13.498 | 0.000 | 62.648 | 12.396 | 48.339 | 36.805 | 195.046 | 8.146 | 70.052 | 33.639 | 93.492 |
| 111 | 17.958 | 17.562 | 20.665 | 61.757 | 12.312 | 56.599 | 36.874 | 196.161 | 7.959 | 76.063 | 33.204 | 94.563 |
| 112 | 0.000 | 2.075 | 30.844 | 81.698 | 31.664 | 52.177 | 59.891 | 140.613 | 5.899 | 89.220 | 56.010 | 107.522 |
| 113 | 20.463 | 9.555 | 9.422 | 50.691 | 11.071 | 18.815 | 24.844 | 66.863 | 5.033 | 39.230 | 23.025 | 126.061 |
| 114 | 11.600 | 10.039 | 0.000 | 23.461 | 9.637 | 6.919 | 19.841 | 13.352 | 4.556 | 31.808 | 18.077 | 121.681 |
| 115 | 7.913 | 5.586 | 0.000 | 42.483 | 7.626 | 10.266 | 20.514 | 62.559 | 3.700 | 30.361 | 18.116 | 58.103 |
| 116 | 293.745 | 16.828 | 322.713 | 329.341 | 69.757 | 330.215 | 221.367 | 815.046 | 61.027 | 408.579 | 198.830 | 118.480 |
| 117 | 232.412 | 3.866 | 176.196 | 172.417 | 59.510 | 200.503 | 160.510 | 287.845 | 10.462 | 249.816 | 143.502 | 64.234 |
| 118 | 0.000 | 13.714 | 0.000 | 34.350 | 5.238 | 10.774 | 9.233 | 9.233 | 10.036 | 19.091 | 9.822 | 185.198 |
| 119 | 28.284 | 6.740 | 12.767 | 55.560 | 11.560 | 21.940 | 28.397 | 22.334 | 5.070 | 45.296 | 25.037 | 111.847 |
| 120 | 143.237 | 5.120 | 61.209 | 145.417 | 32.810 | 29.927 | 83.469 | 214.569 | 24.023 | 137.591 | 74.554 | 51.986 |
| 121 | 15.384 | 2.301 | 98.462 | 90.691 | 27.023 | 133.594 | 78.469 | 103.791 | 6.170 | 115.750 | 73.288 | 116.271 |
| 122 | 30.317 | 2.392 | 58.426 | 103.211 | 23.038 | 74.481 | 66.993 | 315.970 | 5.845 | 97.363 | 61.992 | 54.248 |
| 123 | 8.995 | 1.043 | 0.000 | 44.565 | 8.076 | 8.457 | 22.995 | 55.419 | 5.934 | 37.490 | 20.043 | 43.650 |
| 124 | 722.267 | 45.054 | 614.958 | 505.053 | 103.687 | 574.906 | 269.685 | 598.663 | 96.909 | 406.847 | 237.910 | 677.024 |
| 125 | 180.650 | 10.899 | 622.722 | 398.814 | 99.108 | 450.410 | 270.414 | 1109.282 | 42.162 | 389.016 | 235.541 | 258.810 |
| 126 | 502.201 | 15.848 | 383.468 | 270.644 | 65.986 | 367.189 | 186.832 | 1074.034 | 43.383 | 346.717 | 161.079 | 123.138 |
| 127 | 894.238 | 55.072 | 1446.632 | 346.666 | 119.516 | 668.339 | 266.735 | 516.649 | 23.319 | 368.011 | 239.869 | 403.312 |
| 128 | 139.133 | 31.994 | 0.000 | 328.224 | 397.632 | 345.830 | 236.922 | 431.006 | 20.772 | 299.296 | 218.689 | 274.354 |
| 129 | 236.394 | 76.186 | 1401.179 | 1247.324 | 436.283 | 750.081 | 413.133 | 610.585 | 89.354 | 560.993 | 423.782 | 639.779 |
| 130 | 23.578 | 5.235 | 150.481 | 572.408 | 121.890 | 111.331 | 145.200 | 314.410 | 30.899 | 197.983 | 137.241 | 250.686 |
| 131 | 139.198 | 41.039 | 1008.241 | 1052.915 | 205.081 | 403.822 | 319.091 | 420.152 | 124.930 | 494.253 | 290.539 | 874.117 |

Table 27: The Time cost (s) on datasets [67]-[131] (Part-2).