# OpenReview forum: "CLUBench: A Clustering Benchmark"
_ICLR.cc/2026/Conference — ICLR 2026 Conference Withdrawn Submission_

### Official Review · Reviewer_y2gF · 2025-10-28

**Soundness:** 3
**Presentation:** 2
**Contribution:** 3
**Rating:** 4
**Confidence:** 4

**Summary:**

This paper introduces CLUBench, a large-scale clustering benchmark tool that evaluates 23 different algorithms that include classical, subspace and deep methods across 131 datasets spanning tabular, text and image modalities while providing results using 4 metrics which are ACC, NMI, ARI and time. The authors also provide some interesting findings like deep learning methods do not consistently dominate classical ones and much more.

**Strengths:**

The paper clearly mentions all the algorithms and the datasets that are present in the benchmarking tool along with what the authors are planning to add in the future. This tool also allows the user to add their own datasets and methods.

The authors also showcase the strengths of the benchmarking tool by giving some insights on the methods such as how deep learning clustering techniques were not giving too much of a significant advantage compared to conventional methods in some cases.

The authors also clearly mentioned their search spaces of the hyper parameters allowing for greater reproducibility.

**Weaknesses:**

Some of the plots were too overcrowded and hence difficult to read. For example, Figure 2, 4 and 9.

The authors mentioned that the run time numbers were collected on different hardware. So, the run time comparisons are not too well comparable?

The authors in order to control the scale of the experiments undersampled the datasets with sample size n >= 10000 samples and also removed the extreme clusters which were also outliers. This may alter some class imbalance. This is not shown as a comparison to the original dataset.

The authors only give out 4 result metrics: ACC, NMI, ARI and time. All of these metrics except time uses ground truth labels. The authors did not mention why other metrics like SIL, CHI, DBI were not given.

The authors mention that for non image data, the CNN based methods were adapted to MLP backbones. However, they did not provide any form of comparisons showcasing that performance between both of them were the same.

Your benchmarks assume that we already know the number of clusters in advance. Something to think for the future.

**Questions:**

There is already a project called clustering benchmark (clustbench). See clustering-benchmarks.gagolewski.com Wouldn't be nicer to use another name a bit different rather than CLUbench?

Could you please explain why internal metrics like CHI, SIL were not used?

Could you please explain how much difference in the results are coming when there is no undersampling being done to the data and when there is?

Could you please show the comparison results when CNN based methods were adapted to MLP and the original CNN method?

What implementation of spectral clustering are you using? What about memory constraints of spectral clustering?

Looking at the SpeClu code this seems to be a new implementation why to cite a 2001 paper on this? Also why to import the scikit learn version but not use it or use it some times?

---

### Official Review · Reviewer_grFx · 2025-10-31

**Soundness:** 2
**Presentation:** 2
**Contribution:** 1
**Rating:** 2
**Confidence:** 4

**Summary:**

The authors introduce CLUBench, performing a big benchmarking study for existing clustering algorithms. They evaluate 23 clustering methods (conventional as well as deep clustering algorithms) across 131 datasets (incl. tabular, text, image, and bioinformatics data).
They compare the clustering results for various hyperparameter settings regarding NMI, ARI, and Accuracy and try to find groups of clustering methods that perform similar. They furthermore investigate hyperparameter robustness and whether dataset properties like dimensionality or imbalance of classes influences the performance of clustering methods in a predictale way. The latter analyses are mostly performed by visually analysing t-SNE embeddings of the performances accross datasets and hyperparameter settings.

**Strengths:**

S1) Neutral benchmarks are really important in nowadays research. The authors investigated a large number of datasets and clustering methods, including a range of hyperparameter settings.

S2) The paper is easy to follow and the authors derive several qualitative outcomes, e.g., that deep clustering is not necessarily better than traditional clustering.

S3) The results seem reproducible and a lot of tables resulting from the benchmark study are given in the appendix.

**Weaknesses:**

W1) The selection of ranges for hyperparameters is suboptimal and sometimes not clear. E.g., according to Table 9, k was not tuned for k-Means. While one can assume the authors used the ground truth number of clusters, this leads to a biased evaluation regarding the hyperparameter robustness in Figure 10.

While the authors tune DBSCAN parameters in a non-trivially chosen parameter range, they could have used existing literature [0] as a basis instead- especially as their method quite often leads to very poor clustering results for DBSCAN. I.e., regard k-distance values instead of arbitrarily chosen percentages of average pairwise distances.

[0] Schubert, E., Sander, J., Ester, M., Kriegel, H. P., & Xu, X. (2017). DBSCAN revisited, revisited: why and how you should (still) use DBSCAN. ACM Transactions on Database Systems (TODS), 42(3), 1-21.

W2) Evaluation of clusterings with noise is not clear. The authors define in 3.1 that each point belongs to a cluster. However, noise-detecting methods like DBSCAN leave points unassigned. It is not clear how the authors handled this when computing NMI and ARI.

W3) Inferring conclusions based on t-SNE embeddings is problematic and while they give nice ideas, I do not think they are scientifically reliable.

W4) The cleaning process of the data is problematic and not described prominently enough. It might yield significant biases in the evaluation.

W5) In Table 5 many results are not available. However, there is no reason given for that. A

W6) Quite some important information is only contained in the appendix

Minor: Section heading of 4.3.1 is at the bottom of the page without any text following.

**Questions:**

See weak points

Q1) How many of the results were not available and why?

Q2) How did you handle noise labels in the evaluation?

Q3) Why did you choose the search ranges the way you did? Are there guidelines? Did you test different values for k for k-Means?

---

### Official Review · Reviewer_jCJA · 2025-11-01

**Soundness:** 2
**Presentation:** 2
**Contribution:** 2
**Rating:** 2
**Confidence:** 4

**Summary:**

CLUBench attempts to benchmark 23 clustering algorithms (conventional and deep learning-based) on 131 datasets across tabular, text, and image modalities. The authors report 174,485 experiments with analysis of performance comparisons, similarity studies, and low-rank structure. The large scale benchmark should help practitioners in selecting which algorithm is  suitable for a given task.

**Strengths:**

1. The paper represents substantial work, collecting 131 datasets across multiple modalities and conducting extensive experiments. The scale of the evaluation is commendable.
2. The benchmark aggregates datasets from diverse domains and data types (tabular, text, image, bioinformatics), which could be valuable for the community if implementation issues are resolved.

**Weaknesses:**

1. Despite claims of comprehensiveness, only 23 algorithms are included and existing benchmarking efforts like ClustPy (https://github.com/collinleiber/ClustPy) [1] are neither discussed, nor compared against.
2. The benchmark is incomplete at submission. In Table 3., multiple methods show "NA" for original images, suggesting the authors couldn't run the implementations on original data for some deep clustering algorithms while it worked for others like ConClu or PICA.
3. The DEC implementation (https://anonymous.4open.science/r/CLUBench-ICLR2026/CLUBench/algorithms/DEC.py) is incomplete. Some implementation files are missing (e.g. utils.py for TwoLayerDAE). Overall, a comparison to the original results and/or implementation is missing to check if the reimplementations for the benchmark are correct.
3. The selection of algorithms for the benchmark is not well motivated. Out of the many existing (deep) clustering algorithms a what seems to be arbitrary subset was selected without clear motivation.

[1] Leiber, Collin, et al. "Benchmarking deep clustering algorithms with clustpy." 2023 IEEE International Conference on Data Mining Workshops (ICDMW). IEEE, 2023.

**Questions:**

1. How does CLUBench differ from existing benchmarking efforts like ClustPy, and what are the comparative advantages? A direct comparison would strengthen your contribution and justify the need for another benchmark.
2. Table 1 is incomplete. Can you provide the remaining results?
3. For each algorithm: are you using official implementations, faithful reimplementations, or modified versions? If these are reimplementations or modified versions, how do you validate that the implementations reproduce the results?
4. How was the selection of (deep) clustering algorithms done? Given that more recent deep clustering algorithms are missing it should be clearly motivated why the current methods were selected.

---

### Official Review · Reviewer_QqWN · 2025-11-02

**Soundness:** 2
**Presentation:** 2
**Contribution:** 1
**Rating:** 2
**Confidence:** 3

**Summary:**

This paper introduces a new clustering benchmark, CLUBench. Extensive empirical evaluation is performed comparing well known clustering methods across the different benchmark datasets. Deep clustering methods are also considered.  Low-rank characteristics in performance is observed and analyzed.

**Strengths:**

* **Empirical Breadth**: There is a lot of information in the experiments performed. The authors run a large number of algorithms on a large number of benchmarks.
* **Need for Standardization**: Indeed, I would agree that a clustering benchmark can be a meaningful contribution, even in 2025.

**Weaknesses:**

* **Research Question** - I am a bit confused about the insights gained from the proposed benchmark compared to previous benchmarks. What is it that we learn from this benchmark that we didn't already know in the scientific community? I ask this not to be rude, but because I feel this is the core question that needs to be answered by the paper. Otherwise, the great efforts of the authors will not yield the impact I expect that they want. Table 1 is not sufficient in my opinion.
* **Benchmark Configuration** - After reading, I am still left with the question about the design space of which datasets should be in the benchmark. There is a dearth of datasets with larger number of clusters. There is a dearth of more modern Generative AI world clustering datasets. There is a heavy focus on Tabular. I'm struggling still with the reasons listed for these datasets.
* **Algorithm Selection** - Similar question here, what about all distributed clustering techniques? What about methods that are much more scalable and therefore likely to be used by practitioners?
* **Deep Clustering Results** - I think I have missed something important, why do the deep clustering results look so different than a paper like https://ojs.aaai.org/index.php/AAAI/article/view/26032 ?
* **Venue Match** - I think that readers in KDD and similar more traditionally data mining conferences could be better suited for this paper.

**Questions:**

1. Please clarify Deep Clustering results?
2. What is the core observation that this benchmark provides that previous benchmarks did not provide?

---

### Note · Authors · 2025-11-13

**Comment:**

Dear AC and Reviewers

We appreciate the efforts from you and will continue to improve our work.

Sincerely,
All authors

**Withdrawal Confirmation:**

I have read and agree with the venue's withdrawal policy on behalf of myself and my co-authors.